# Uncovering the Hidden Dynamics of Video Self-supervised Learning under Distribution Shifts

**Pritam Sarkar**[1,2]  **Ahmad Beirami**[3]  **Ali Etemad**[1,3]

[1]Queen's University  [2]Vector Institute  [3]Google Research

{pritam.sarkar, ali.etemad}@queensu.ca  beirami@google.com

## Abstract

Video self-supervised learning (VSSL) has made significant progress in recent years. However, the exact behavior and dynamics of these models under different forms of distribution shift are not yet known. In this paper, we comprehensively study the behavior of six popular self-supervised methods ($v$-SimCLR, $v$-MoCo, $v$-BYOL, $v$-SimSiam, $v$-DINO, $v$-MAE) in response to various forms of natural distribution shift, i.e., (*i*) context shift, (*ii*) viewpoint shift, (*iii*) actor shift, (*iv*) source shift, (*v*) generalizability to unknown classes (zero-shot), and (*vi*) open-set recognition. To perform this extensive study, we carefully craft a test bed consisting of 17 in-distribution and out-of-distribution benchmark pairs using available public datasets and a series of evaluation protocols to stress-test the different methods under the intended shifts. Our study uncovers a series of intriguing findings and interesting behaviors of VSSL methods. For instance, we observe that while video models generally struggle with context shifts, $v$-MAE and supervised learning exhibit more robustness. Moreover, our study shows that $v$-MAE is a strong temporal learner, whereas contrastive methods, $v$-SimCLR and $v$-MoCo, exhibit strong performances against viewpoint shifts. When studying the notion of open-set recognition, we notice a trade-off between closed-set and open-set recognition performance if the pretrained VSSL encoders are used without finetuning. We hope that our work will contribute to the development of robust video representation learning frameworks for various real-world scenarios. The project page and code are available at: https://pritamqu.github.io/OOD-VSSL.

## 1  Introduction

Self-supervised learning has achieved tremendous success in learning strong and meaningful representations in various video domains, such as action recognition [1, 2, 3, 4, 5, 6, 7], action localization [8], video summarization [9], and video captioning [10, 11, 12]. Considering the diversity and complexity of the video domain, real-world deployment of video-based intelligent systems requires to understand the model performance under distribution shifts. Distribution shifts may occur due to differences in contextual information, viewpoint, geographical location, and the presence of unknown classes with respect to the training data, among others.

Despite the vast amount of work on VSSL [13, 14, 15, 16, 17, 18, 5, 1, 19, 8, 12], a number of fundamental questions regarding the out-of-distribution behavior and dynamics of VSSL methods remain unanswered. To date, there have been no comprehensive studies of these aspects, which we attempt to address in this paper. Specifically, we pose and answer the following questions:

**Q1.** How do the learned spatial and temporal representations vary based on different VSSL pretraining methodologies? How robust are these representations to different distribution shifts?

37th Conference on Neural Information Processing Systems (NeurIPS 2023).

**Q2.** Considering recent findings about the robustness of finetuning on the generalizability of large language models (LLMs) [20, 21], we pose the question: How does finetuning influence the out-of-distribution (OoD) generalization and zero-shot performance of VSSL?

**Q3.** How do VSSL methods perform on open-set problems (where test samples can be from classes previously unknown to the model)? And what is the relationship between performance in closed-set vs. open-set recognition?

**Q4.** Do different VSSL methods exhibit comparable decision-making patterns ('decision similarity') given the same training conditions? And how is this impacted by different distribution shifts?

To address these questions, we consider six different self-supervised learning algorithms, (*i*) SimCLR [22], (*ii*) MOCO-v3 [23], (*iii*) BYOL [24], (*iv*) SimSiam [25], (*v*) DINO [26], and (*vi*) MAE [27], along with fully supervised learning, and analyze their behaviors in various OoD settings. We select these methods to cover three key categories of self-supervision, namely *contrastive* (SimCLR, MoCo-v3), *non-contrastive* (BYOL, SimSiam, DINO), and *generative* (MAE), approaches. In particular, these methods represent fundamental approaches on which numerous variants have been built and therefore represent many existing nuanced solutions in the area [2, 3, 4, 5, 6, 7, 1, 19, 8, 12]. For distribution shifts, we study a series of different possibilities which occur in videos in real-world settings due to changes in context (e.g., real vs. mime actions) [28, 29, 30, 31, 32], viewpoint (e.g, third-person vs. ego-centric view) [33, 34, 35, 36], actor (e.g., real-world vs. synthetic) [37, 38], and data sources (e.g., different datasets) [39, 40]. Moreover, we evaluate the generalizability of the VSSL methods on unseen classes, i.e., zero-shot recognition [41, 42, 43] and open-set recognition [44, 45, 46, 47]. To

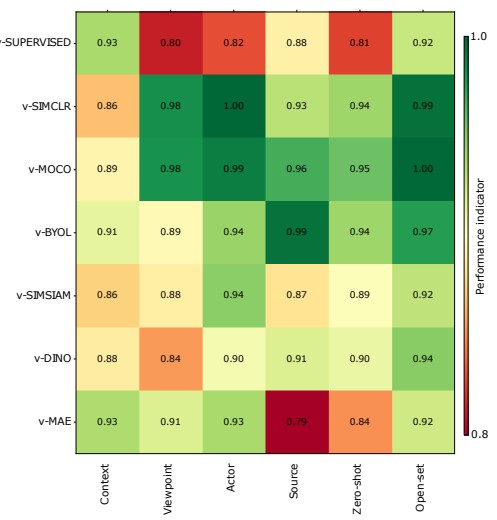

Figure 1: We present a **high-level overview** of different video learning methods depicting their performance under different distribution shifts. We normalized the eval. metric of each method to that of the highest performing method in each OoD scenario and averaged the results over multiple OoD datasets. See Appendix D.6 for more details.

perform this investigation, we design a comprehensive study consisting of 17 in-distribution and out-of-distribution (InD-OoD) dataset pairs and examine the dynamics of VSSL under different distribution shifts using a variety of evaluation protocols including linear evaluation, finetuning, unsupervised clustering, and zero-shot recognition. Moreover, for a fair comparison, the VSSL methods are pretrained in identical experimental setups. We provide a high-level overview comparing the performance of all methods across all shifts in Figure 1. To the best of our knowledge, this is the first work to study VSSL under distribution shift to this depth.

In summary, our contributions are as follows:

- We present the first comprehensive and large-scale systematic investigation of real-world distribution shifts on VSSL methods. Our study encompasses 2 large-scale pretraining datasets, 7 video learning algorithms, 17 InD-OoD dataset pairs, as well as 3 toy datasets. Our thorough evaluation involves a total of 269 experiments, covering various evaluation protocols including linear evaluation, finetuning, unsupervised clustering, zero-shot recognition, and open-set recognition.
- Our study uncovers a series of intriguing behaviors and relationships of various VSSL methods, shedding light on the strengths and weaknesses that often go unnoticed in InD validation. Moreover, our investigation provides a comprehensive and impartial perspective on the effectiveness of supervised vs. self-supervised pretraining.

## 2   Related work

The analysis of robustness is a well-established area in computer vision, with an extensive body of research dedicated to image-based models [48, 49, 50, 51, 52]. Despite the growing popularity of video models in different domains, detailed comparative studies on their robustness remains

under-explored. We come across two recent works [53, 54] that study the behavior of video models (supervised) against synthetic perturbations. An initial study [54] explores the performance of video models against spatial corruptions like noise, blur, color jittering and others. In a subsequent work, [53] extends the analysis of video models on temporal perturbations like frame reversal, jumbling, and freezing among others. In particular, our study focuses on real-world distribution shifts, which we find crucial as synthetic perturbations yield little or no consistency to the natural distribution shifts of real data [49, 48]. Moreover, none of the prior works attempts to understand the behavior of *video self-supervised* methods under various forms of distribution shift.

## 3   Preliminaries

**Contrastive methods.** We study two popular contrastive methods: $v$-SimCLR[1] and $v$-MoCo. These methods learn representations by maximizing the similarity of positive pairs while minimizing the similarity of negative pairs using the InfoNCE loss [22, 23, 55], where the similarity scores are calculated using L2-normalized cosine distance. The implementation of our $v$-MoCo is based on MoCo-v3 [23], and it uses a momentum target encoder to fetch the key embeddings for the corresponding query embeddings. The momentum encoder is progressively updated from the online encoder using an exponential moving average (EMA) technique [56]. In contrast, $v$-SimCLR directly uses the online encoder to compute the similarity scores from the embeddings of the given views. Additionally, $v$-MoCo employs a predictor head, unlike $v$-SimCLR.

**Non-contrastive methods.** We study three popular non-contrastive methods $v$-BYOL, $v$-SimSiam, and $v$-DINO. These methods learn meaningful representations by minimizing the distance between positive views of a given sample, without requiring any negative pairs. $v$-BYOL uses an architecture similar to $v$-MoCo, but instead of optimizing an InfoNCE loss, it minimizes the L2-normalized cosine distance of the positive views [24, 25]. $v$-SimSiam is an ablation variant of $v$-BYOL that does not use a target encoder and directly obtains the embeddings corresponding to the views from the online encoder. The setup of $v$-DINO is also similar to $v$-BYOL, but it introduces 'Centering' [26] to prevent against model collapse, instead of using a predictor head like $v$-BYOL and $v$-SimSiam.

**Generative method.** Finally, we also study $v$-MAE which aims to learn representations through reconstructions of heavily masked inputs. $v$-MAE employs an autoencoder architecture [27, 1, 57] where the encoder compresses the input information, which is then reconstructed by the decoder while minimizing the L2-reconstruction error [27].

We provide additional details and design specifics for each method in Appendix B.

## 4   Distribution Shifts

Let $p_y$ denote a probability distribution over labels. Let $p_{x|y}$ denote a class conditional density of input $x$ given labels $y$. We draw $y$ from $p_y$ and then $x$ from $p_{x|y}(x|y)$. In the case of InD, there is no distribution shift and the validation set is drawn from the exact same distribution as above. We consider two types of distribution shifts in this paper: input-based and output-based. For **Input shift**, we assume that the class conditional density shifts from $p_{x|y}$ to $q_{x|y}$. To measure the OoD performance, we choose $q_{x|y}$ such that there is no overlap between the distributions of the unlabelled pretraining data and OoD validation sets. For **Output shift**, we assume that the label distribution shifts from $p_y$ to $q_y$. Note that we are particularly interested in cases where the support set of $q_y$ might be different from that of $p_y$, i.e., novel classes appear at test time.

We study four different input shifts, (*i*) context shift, (*ii*) viewpoint shift, (*iii*) actor shift, and (*iv*) source shift. **Context shift** or out-of-context refers to when the scenic background or contextual information is misleading or absent, e.g., mime actions. Humans possess a deep understanding of actions and can thus recognize actions without much context. However, vision models tend to rely heavily on contextual cues and background objects to make predictions [28, 58, 59]. As a result, models may generalize poorly when contextual information is absent or different from what the model has been trained on. **Viewpoint shift** refers to the change in the viewpoint or perspective of the camera that captures the scene. Some of the popular viewpoints in the video domains include third-person view, egocentric view, bird's-eye view, top-down view, and surveillance camera view,

---

[1]$v$ denotes the video variant of the image-based self-supervised method.

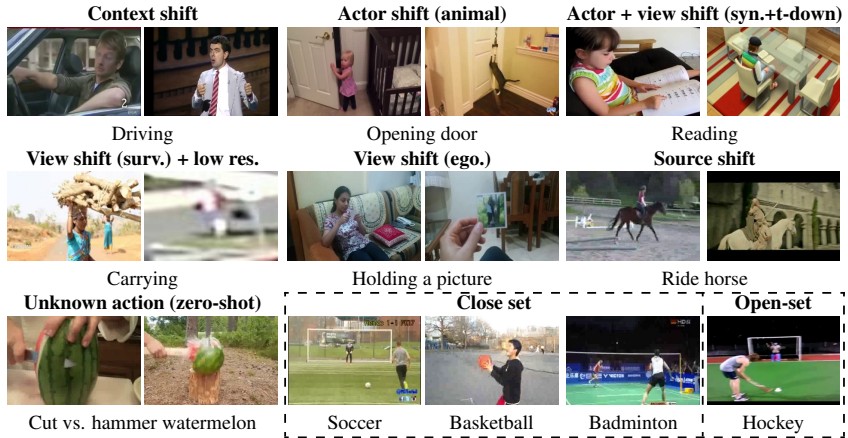

Figure 2: **Sample video frames** of distribution shifts. In these examples, the left frames of each category represent an InD sample and the right frames represent an OoD sample.

among a few others. In this work, we examine the generalizability of models trained on third-person views to other viewpoints including egocentric, surveillance camera, and top-down views. Further, we study generalizability under **actor shift**, which occurs when the 'type' of actors changes between training and test sets. These actor-type shifts can include human-to-animal shifts or synthetic-to-real shifts. We consider human actions in the real world as InD while animal actions and synthetic videos (e.g., video games, animation) are considered OoD. Lastly, we study the distribution shifts caused due to the changes in the data sources, referred to as **source shift**. As discussed in [39, 40], datasets of similar classes also exhibit distribution shifts due to the difference in curation strategies, annotation schemes, demographics, geographical locations, and other factors, even when none of the specific shifts mentioned earlier apply. To assess the generalizability of video models under real-world distribution shifts, we also investigate their performance when faced with multiple forms of distribution shifts occurring simultaneously. For example, we evaluate the model's ability to handle 'top-down synthetic videos' which causes both viewpoint and actor shifts concurrently. A few examples are presented in Figure 2.

We also investigate the dynamics of video models under output shifts. We perform **zero-shot** recognition on both regular actions and unusual or rare actions. An example of rare actions would be 'hammering watermelon' as opposed to 'cutting watermelon', as shown in Figure 2. Lastly, we evaluate performance in **open-set** problems where models are required to distinguish between known vs. unknown classes while correctly predicting the known ones [60, 45, 44, 46]. Deep learning models are known to struggle in such scenarios due to their tendency towards over-confident predictions. We note that existing literature has only evaluated VSSL methods under closed-set scenarios, neglecting the importance of their performance in real-world open-set settings.

## 5 Experiment setup

**Benchmarks.** We use two large-scale video action datasets, Kinetics400 [61] and Kinetics700 [62], for pretraining the video models. The results presented in the main paper use Kinetics400 for pre-training, while we provide additional results based on pretraining with Kinetics700 in Appendix D. To evaluate the video models under distribution shifts, we use a total of 12 real-world benchmarks, comprising: **Mimetics10** and **Mimetics50** [28] as the out-of-context validation sets; **CharadesEgo** [36], **TinyVirat-v2** [63], and **Sims4Action** [64] to investigate viewpoint shifts (egocentric, surveillance camera, and top-down views, respectively); **ActorShift** [65] and **Sims4action** [64], for actor shifts (animal and synthetic domains, respectively); **UCF101** [66] and **HMDB51** [67] for source shift; **UCF101** [66], **HMDB51** [67], and **RareAct** [68] for zero-shot recognition; **UCF101** and **HMDB51** for open-set recognition while using **Kinetics400** and **UCF101** as closed-set. For each OoD validation set, we create an InD training and validation set to measure the change in performance. We construct the InD splits using **Kinetics400** [61], **Kinetics700** [62], **MiT-v2** [69], and **CharadesEgo** [36]. Finally, we also use 3 toy datasets to conduct experiments in controlled

setups, including **ToyBox** [70], **COIL** [34], **STL-10** [71]. Additional details of the benchmarks can be found in Appendix C.

**Pretraining.** To ensure a fair comparison between the VSSL methods, we pretrain them in identical setups with necessary adjustments in hyperparameters. Although some of the methods studied in this work are already available in the literature [3, 5, 72, 1], they follow a variety of experiment setups including different architectures (e.g., R2+1D [73], R3D [72], TSM [74], ViT [75]), inputs, and others. Therefore, they could not be directly adopted for our experiments and are instead re-implemented. Specifically, we keep the encoder, inputs, batch sizes, and maximum number of pretraining epochs the same for all the methods. Furthermore, VSSL methods are tuned based on the InD validation split, with no exposure to OoD validation sets. We use the ViT-Base [75, 76] as the encoder, with a patch size of $4 \times 16^2$. Amongst the 6 VSSL methods studied in this paper, all of them use a Siamese [77] architecture other than $v$-MAE. Therefore, the contrastive and non-contrastive methods are fed with 2 random spatio-temporal augmented crops from the original videos, similar to [72]. For a fair comparison, the $v$-MAE which requires a single view as input, is fed with $3 \times 32 \times 112^2$ inputs, while the other VSSL methods are fed with $3 \times 16 \times 112^2$ inputs per view. Additional details for VSSL pretraining can be found in Appendix B.

**Evaluation.** To study input-based distribution shifts, we perform linear evaluation and finetuning using the InD training splits, followed by evaluating on both InD and OoD validation splits. We follow the standard protocol used in [72, 2, 4, 6, 78, 79, 7] for linear evaluation and finetuning. To evaluate the models on zero-shot recognition, we follow [41, 42] and jointly train video-text encoders using the Word2Vec [80] word embeddings. Lastly, we follow [45] for open-set recognition. We report top-1 accuracy for multi-class classification and mean average precision (meanAP) for multi-label multi-class classification. Moreover, for open-set problems, we report the area under the ROC curve (AUC) for distinguishing known vs. unknown, and accuracy for measuring closed-set performance. Please find more details in Appendix C.

## 6 Findings

**Q1: Dynamics of learned spatial and temporal representations under distribution shifts**

Our experiments on context shift show that video models greatly suffer in out-of-context generalization, as OoD performance significantly drops for all of the methods as presented in Table 1. Notice that $v$-Supervised achieves the best OoD performance under linear evaluation (**Lin.**) and $v$-MAE achieves the best when finetuned (**FT**), for both benchmarks. Intuitively speaking, the models need to learn strong temporal dynamics to generalize well under context shift, as the background or contextual information may be misleading or absent. $v$-MAE and $v$-Supervised show strong temporal learning capabilities as they learn *time-variant* representations. This is in contrast to the other (contrastive and non-contrastive) methods which encourage learning *time-invariant* or time-persistent representations by minimizing the embedding distance of the positive pairs sampled from two different timestamps. Additionally, our statistical analysis in Appendix D.7 confirms the higher robustness of $v$-Supervised and $v$-MAE against context shift (10 class) in both linear and finetuned setups. Moreover, in context shift with 50 classes, $v$-Supervised exhibits more robustness in linear evaluation, while $v$-MAE is more robust when finetuned.

Table 1: Comparison under **context** shift.

| Method | Context (10 class) | | | | Context (50 class) | | | |
| | Lin. | | FT | | Lin. | | FT | |
| | OoD | InD | OoD | InD | OoD | InD | OoD | InD |
|---|---|---|---|---|---|---|---|---|
| $v$-Supervised | **37.0**±0.4 | 89.2±0.5 | **41.2** | 92.7 | **15.0**±0.5 | 66.3±0.5 | 19.0 | 78.1 |
| $v$-SimCLR | 31.4±1.5 | **92.5**±0.2 | 35.3 | 94.4 | 14.9±0.5 | 71.7±0.5 | 19.1 | **81.8** |
| $v$-MoCo | 29.7±0.4 | 92.2±0.6 | 39.0 | 94.2 | **15.0**±0.2 | **74.0**±0.2 | 20.5 | **81.8** |
| $v$-BYOL | 31.6±0.7 | 89.3±0.2 | **41.2** | 94.6 | 14.4±0.2 | 71.8±0.2 | 21.1 | 80.8 |
| $v$-SimSiam | 30.8±0.7 | 89.5±0.5 | 40.4 | 93.8 | 13.8±0.6 | 67.9±0.6 | 19.0 | 78.2 |
| $v$-DINO | 34.8±0.8 | 90.4±0.1 | 40.4 | 93.6 | 13.0±0.3 | 68.7±0.3 | 19.0 | 78.8 |
| $v$-MAE | 33.3±1.8 | 82.9±1.0 | **41.2** | **95.2** | 12.3±0.4 | 56.4±0.4 | **26.0** | 81.4 |

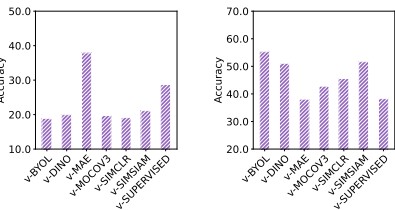

(a) Transformation recog.  (b) Object recog.

Figure 3: Comparing VSSL methods on **disentangled** (a) temporal and (b) spatial representations.

To further investigate the performance of the video models in learning temporal dynamics, we conduct a toy experiment in a controlled setup using ToyBox [70]. ToyBox consists of 9 distinct temporal transformations of 12 different toy objects and transformations include positive rotations, negative rotations, and translations across the $x$, $y$, and $z$ axes. We use the frozen VSSL encoders to extract embeddings, which are then used to perform a K-means clustering to distinguish the transformations. As the static spatial information is non-discriminative to the temporal transformations, the models have to understand the underlying temporal dynamics to correctly distinguish the applied transformations. The results, presented in Figure 3 (a), show that $v$-MAE and $v$-Supervised outperform the other methods by a large margin on this task, confirming that they are better temporal learners. Additionally, to ensure that the superiority of $v$-MAE and $v$-Supervised are not due to spatial representations, we also test the frozen encoders in object recognition using the videos of the same static objects (i.e., no temporal information), and we find that $v$-MAE and $v$-Supervised are worse among all of them (see Figure 3 (b)). This proves that $v$-MAE and $v$-Supervised are indeed strong temporal learners while being weak spatial learners.

These findings raise an interesting question: if the contrastive and non-contrastive methods are not effective temporal learners, then how do they perform well on action recognition [72, 5, 81, 2, 79, 82, 3, 7]? We believe that the strong performance of these methods on action recognition can be attributed to the significance of spatial information in classifying human activities. The temporal sequence of representations might not be as crucial for these tasks, which possibly allows contrastive and non-contrastive methods to perform well.

Table 2: Comparison of video models in **viewpoint** and **actor** shifts.

| Method | Viewpoint (ego.) | | | | Viewpoint (surv.+low res) | | | | View+Act (t-down+syn.) | | | | Actor (animal) | | | |
|---|---|---|---|---|---|---|---|---|---|---|---|---|---|---|---|---|
| | Lin. | | FT | | Lin. | | FT | | Lin. | | FT | | Lin. | | FT | |
| | OoD | InD | OoD | InD | OoD | InD | OoD | InD | OoD | InD | OoD | InD | OoD | InD | OoD | InD |
| $v$-Supervised | 11.4±0.1 | 14.3±0.1 | 13.6 | 17.8 | 23.1±0.2 | 33.4±0.2 | 24.5 | 43.4 | 28.5±0.6 | 62.3±0.6 | 44.3 | 76 | 64.8±0.6 | 90.0±0.2 | 69.6 | 92.3 |
| $v$-SimCLR | 12.7±0.2 | 14.7±0.1 | 15.6 | 19.6 | **26.1**±0.6 | 39.1±0.6 | 28.0 | 47.5 | **42.4**±1.8 | 67.8±1.8 | **63.1** | 78.8 | 67.9±0.6 | 91.7±0.2 | **73.2** | **92.9** |
| $v$-MoCo | **13.3**±0.1 | **15.0**±0.2 | **16.1** | 19.4 | 24.8±1.2 | **40.0**±1.2 | 27.6 | **48.6** | 41.1±0.4 | **67.9**±0.4 | 62.8 | **80.0** | 68.1±0.9 | **92.2**±0.2 | 71.4 | **92.9** |
| $v$-BYOL | 12.0±0.1 | 14.4±0.1 | 15.1 | 18.7 | 22.7±0.7 | 37.8±0.7 | 24.7 | 46.9 | 37.3±0.2 | 65.6±0.2 | 56.2 | 78.2 | **68.3**±0.3 | 91.5±0.0 | 71.4 | **92.9** |
| $v$-SimSiam | 11.6±0.2 | 14.1±0.1 | 13.7 | 17.6 | 23.3±0.4 | 34.3±0.4 | 25.8 | 45.4 | 40.0±1.0 | 65.5±1.0 | 53.6 | 76.0 | 68.1±1.5 | 91.1±0.2 | 71.4 | 92.1 |
| $v$-DINO | 12.0±0.2 | 14.4±0.1 | 13.7 | 17.4 | 22.3±0.9 | 35.3±0.9 | 24.2 | 45.1 | 35.3±0.4 | 62.9±0.4 | 50.3 | 77.5 | 66.7±0.5 | 90.7±0.1 | 71.4 | 92.2 |
| $v$-MAE | 10.9±0.0 | 13.7±0.1 | 14.2 | **21.4** | 23.5±0.9 | 32.0±0.9 | **29.1** | **48.6** | 37.8±3.0 | 58.0±3.0 | 61.1 | 76.2 | 59.8±0.7 | 85.9±0.1 | 72.6 | **92.9** |

The results on viewpoint shifts, presented in Table 2 reveal that contrastive methods ($v$-SimCLR and $v$-MoCo) generally achieve superior performance in all three setups in both linear and finetuning schemes. Moreover, based on the statistical analysis in Appendix D.7, $v$-SimCLR exhibits more robustness to all three viewpoint shifts, while $v$-MoCo exhibits robustness in egocentric and top-down viewpoint shifts. We believe the better performance of $v$-SimCLR and $v$-MoCo is due to the availability of negative samples during pretraining which improves viewpoint invariance compared to the other approaches. Note that while aggressive cropping also improves viewpoint invariance as discussed in [83], we do not notice such improvements in non-contrastive methods although a similar aggressive cropping is applied. It is worth noting that while $v$-Supervised exhibits a comparable performance to VSSL methods in egocentric and surveillance camera viewpoint shifts, its performance decreases significantly when multiple shifts are applied concurrently, as evident in the case of synthetic top-down viewpoint shift. This suggests that supervised learning may not generalize well in scenarios with more complex and realistic distribution shifts.

For further investigation, we study viewpoint invariance in a controlled setup using the pretrained VSSL encoders through unsupervised evaluation, avoiding any form of additional training which may alter the learned representations. Moreover, to limit the influence of temporal representations we use an image-based toy dataset COIL100 [34] which is commonly used to test viewpoint invariance [84, 85], as it contains images of similar objects from different angles. Our results presented in Figure 4 (left) show that $v$-MoCo achieves the best performance, fol-

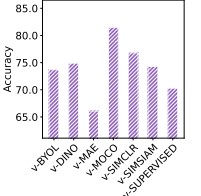

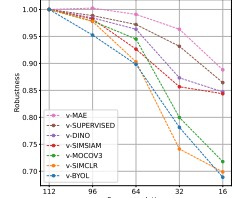

(a) Viewpoint invariance     (b) Low-resolution

Figure 4: Comparing models on **viewpoint invariance** (a) and **low-resolution robustness** (b).

lowed by $v$-SimCLR. Moreover, the results show that $v$-MAE is susceptible to viewpoint shifts as it performs worse. We believe that as $v$-MAE and $v$-Supervised are trained with a single view of frame sequences, they are more sensitive to viewpoint shifts. Additionally, amongst the non-contrastive methods, $v$-DINO shows a better performance. Notice that our results so far have revealed a trade-off between learning viewpoint invariance vs. strong temporal dynamics. While single-stream networks learn better temporal dynamics, Siamese frameworks learn better viewpoint invariance.

Notice that while contrastive methods dominate performance under viewpoint shifts (Table 2), $v$-MAE shows the best performance only in the case of *low-resolution* surveillance cameras. We hypothesize that this is due to the fact that in addition to $v$-MAE being a strong temporal learner (see Figure 3), it is also inherently robust to low-resolution inputs. This is likely since it learns strong pixel-level relations [86] through reconstruction of highly occluded inputs. To further investigate this, we perform a follow-up study in a controlled toy setup where we systematically decrease the resolution of input frames from $112^2$ to $16^2$ and measure the robustness as $1 - (\text{acc}_{112} - \text{acc}_N)/100$, following [53]. Here $\text{acc}_N$ refers to accuracy at resolution $N^2$. We use $112^2$ as the maximum resolution as this is used during pretraining, and $16^2$ is set as the lowest resolution as we use a patch size of $16^2$. To limit the interference of temporal representations with the outcome, we use an image dataset, STL10 [71], to perform unsupervised image classification using the frozen pretrained VSSL encoders. The results presented in Figure 4 (right) show that $v$-MAE achieves a more steady performance on low-resolution inputs compared to the other methods, which is in alignment with our hypothesis.

> **Highlights:** (*a*) Video models generally struggle in out-of-context generalization, while $v$-Supervised and $v$-MAE exhibit better performance as they are strong temporal learners. (*b*) Contrastive methods ($v$-SimCLR, $v$-MoCo) exhibit better performance to viewpoint shifts. (*c*) $v$-MAE is robust against extremely low-resolution inputs. (*d*) $v$-Supervised shows extreme vulnerability in complex scenarios when multiple distribution shifts are applied concurrently.

Table 3: Comparison under **source** shift.

| Method | UCF/HMDB | | | | HMDB/UCF | | | |
| | Lin. | | FT | | Lin. | | FT | |
| | OoD(H) | InD(U) | OoD(H) | InD(U) | OoD(U) | InD(H) | OoD(U) | InD(H) |
|---|---|---|---|---|---|---|---|---|
| $v$-Supervised | 45.8±0.8 | 92.7±0.2 | 56.3 | 96.8 | 50.3±0.9 | 75.9±0.9 | 51.1 | 84.6 |
| $v$-SimCLR | 47.7±0.3 | 96.0±0.4 | 53.7 | 97.7 | 55.1±0.4 | 82.0±0.4 | 56.7 | 85.6 |
| $v$-MoCo | **51.5**±0.6 | **97.1**±0.5 | 55.7 | **98.9** | 57.2±0.4 | **84.5**±0.4 | 58.3 | 86.3 |
| $v$-BYOL | 51.4±0.7 | 94.5±0.2 | **59.4** | 97.5 | **63.1**±0.2 | 79.9±0.2 | **60.4** | 87.3 |
| $v$-SimSiam | 46.1±1.0 | 92.6±0.7 | 50.6 | 95.5 | 52.2±0.3 | 76.2±0.3 | 53.2 | 83.0 |
| $v$-DINO | 49.3±0.4 | 94.3±0.5 | 51.4 | 96.3 | 53.8±0.4 | 77.5±0.4 | 55.6 | 82.8 |
| $v$-MAE | 39.5±0.2 | 89.2±0.9 | 55.3 | 96.5 | 39.1±1.4 | 72.8±1.4 | 43.6 | 81.3 |

Table 4: Comparison in **zero-shot** action recognition.

| Method | UCF | | HMDB | | RareAct | | K400 (InD) | |
| | Lin. | FT | Lin. | FT | Lin. | FT | Lin. | FT |
|---|---|---|---|---|---|---|---|---|
| $v$-Supervised | n/a | 37.4 | n/a | 19.0 | n/a | 9.9 | n/a | 59.0 |
| $v$-SimCLR | **37.2**±1.8 | 40.3 | 18.6±1.7 | 24.9 | 7.7±0.6 | 10.4 | 56.8±0.2 | **69.8** |
| $v$-MoCo | 35.2±3.0 | **46.3** | 19.5±2.6 | 22.3 | **8.7**±0.6 | 10.5 | **58.4**±0.4 | 69.3 |
| $v$-BYOL | 33.0±1.4 | 41.0 | **22.4**±2.1 | 24.7 | 7.5±0.4 | 10.3 | 57.4±0.1 | 69.3 |
| $v$-SimSiam | 34.0±1.6 | 38.7 | 18.8±1.0 | 21.0 | 7.7±0.3 | **11.1** | 50.4±0.0 | 64.3 |
| $v$-DINO | 34.3±1.4 | 41.3 | 17.2±0.3 | 22.8 | 8.1±0.3 | 10.5 | 53.4±0.2 | 65.7 |
| $v$-MAE | 25.5±1.4 | 42.1 | 14.2±0.7 | **25.8** | 5.8±0.9 | 10.7 | 35.9±0.0 | 68.4 |

**Q2: Effect of finetuning under distribution shifts**

Our thorough empirical experiments presented in Tables 1, 2, and 3, show several interesting aspects of finetuning under distribution shifts. First, finetuning generally results in better InD and OoD performance compared to linear evaluation (see Table 5). Moreover, the benefits of finetuning vary between different VSSL methods; for example, $v$-MAE benefits the most from finetuning (see Figures 5 and 6). Second, the benefits of finetuning depend on the nature of the distribution shift. Specifically, by comparing OoD and InD performance, we observe that finetuning is more helpful for actor shift in both animal and synthetic domains (see Figure 5), whereas it is less beneficial for zero-shot and viewpoint shift in both egocentric and surveillance camera view. Moreover, finetuning often leads to 'in-distribution overfitting' where the InD validation improves with additional training, while leading to poor OoD

| | InD | OoD |
|---|---|---|
| Lin. | 63.2% | 31.5% |
| FT. | 71.9% (↑ 8.7) | 37.4% (↑ 5.9) |

Table 5: **Linear vs. finetuned** comparison summary. Please see Appendix D.5 for additional details.

generalization (see Appendix D). Overall, the results of zero-shot recognition presented in Table 4 show that no single method dominates in all three benchmarks for zero-shot recognition. $v$-MAE, $v$-MoCo, and $v$-SimSiam achieve the best performances on HMDB, UCF, and RareAct, respectively, when finetuned. Third, the benefits of finetuning also depend on the InD training benchmark. For example, as shown in Figure 5, while finetuning is significantly beneficial under the source shift of UCF to HMDB, it interestingly hurts the performance when this shift is reversed (HMDB to UCF). This can be attributed to the limited utility of small-scale datasets, such as HMDB, in learning strong

representations. Such datasets could potentially even diminish the effectiveness of robust pretrained models. For instance, in the case of $v$-BYOL, the performance of the frozen encoder exceeds that of the finetuned model, as seen in Table 3 (HMDB/UCF).

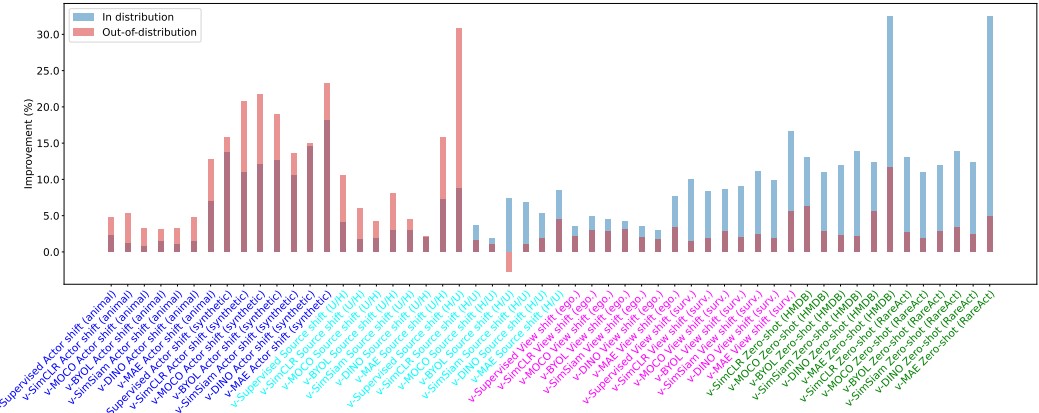

Figure 5: **Linear vs. finetuned** performance comparison under real-world distribution shifts. We measure the improvements as the difference between finetuned and linear evaluation accuracy.

To further investigate scenarios where finetuning may impair VSSL pretraining, we evaluate the video models on temporally perturbed inputs (e.g., freeze frames, random sampling, reverse sampling) in a controlled setup. As the VSSL methods are never trained with temporally perturbed inputs, this constitutes a distribution shift. The results presented in Figure 6 show that end-to-end finetuning of the contrastive and non-contrastive video encoders diminishes the time-invariance of the learned representations, which decreases robustness to temporal perturbations. As a result, in such cases, the frozen encoders perform better than the finetuned ones. Moreover, the frozen $v$-MAE shows the worst performance under temporal shifts as it learns time-variant representations through self-supervised pretraining.

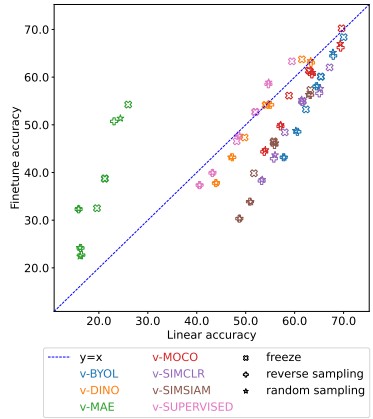

Figure 6: **Linear vs. finetuned** performance comparison under synthetic temporal perturbations.

> **Highlights:** (*a*) As opposed to LLMs, finetuning generally helps VSSL in both InD and OoD. (*b*) The benefits of finetuning largely vary between different VSSL methods and the type of distribution shifts. (*c*) Finetuning provides more benefits against actor shifts in both animal and synthetic domains in comparison to viewpoint shifts like egocentric and surveillance camera views. (*d*) Finetuning degrades robustness to temporal perturbations as it impairs the time-invariant representations of contrastive and non-contrastive methods. It can also degrade performance under source shift depending on the quality of the training benchmark.

## Q3: Closed-set vs. open-set performance

We perform open-set recognition in two setups: (1) we use the Kinetics400 for closed-set recognition, while the non-overlapping classes from the UCF101 and HMDB51 are used for open-set. (2) we use UCF101 and the non-overlapping classes of HMDB for closed-set and open-set recognition, respectively. To perform a comprehensive analysis on the performance of closed-set vs. open-set, we adopt two approaches, i.e, single-objective (closed-set only with cross-entropy error) and joint-objective (closed-set and open-set recognition simultaneously with DEAR loss [45]) variants.

Table 6: Comparison in **open-set** recognition. $v$-Supervised trained from scratch with DEAR loss failed to converge in K/U and K/H. Here, closed/open-set pairs are denoted as K/H: Kinetics/HMDB, K/U: Kinetics/UCF, U/H: UCF/HMDB, CE: cross-entropy error.

(a.) End-to-end finetuned.

| Method | Open-set (AUC) | | | Closed-set w/ DEAR (Acc.) | | Closed-set w/ CE (Acc.) | |
|---|---|---|---|---|---|---|---|
| | K/U | K/H | U/H | K400* | U101* | K400* | U101* |
| $v$-Supervised | - | - | 77.7 | - | 86.5 | 57.6 | 87.4 |
| $v$-SimCLR | **63.0** | 60.1 | 84.3 | 69.8 | **90.6** | 72.5 | 90.3 |
| $v$-MoCo | 61.3 | **61.0** | **85.2** | **70.1** | 90.4 | 72.8 | **90.7** |
| $v$-BYOL | 60.5 | 60.1 | 81.7 | 69.4 | 89.4 | 72.5 | 90.2 |
| $v$-SimSiam | 60.6 | 55.9 | 77.8 | 64.7 | 85.9 | 69.3 | 87.2 |
| $v$-DINO | 60.0 | 58.3 | 81.7 | 65.9 | 87.0 | 71.3 | 88.5 |
| $v$-MAE | 55.1 | 55.5 | 73.1 | 67.5 | 87.8 | **73.3** | 90.5 |

(b.) Linear evaluation.

| Method | Open-set (AUC) | Closed-set w/ DEAR (Acc.) | Closed-set w/ CE (Acc.) |
|---|---|---|---|
| | U/H | U101* | U101* |
| $v$-Supervised | $77.3_{\pm0.1}$ | $80.4_{\pm0.1}$ | 81.7 |
| $v$-SimCLR | $49.9_{\pm0.3}$ | $11.6_{\pm1.0}$ | 84.1 |
| $v$-MoCo | $50.7_{\pm0.6}$ | $32.7_{\pm0.5}$ | 84.9 |
| $v$-BYOL | $56.0_{\pm1.5}$ | $63.7_{\pm0.8}$ | **85.4** |
| $v$-SimSiam | $73.7_{\pm0.3}$ | $79.5_{\pm0.2}$ | 82.5 |
| $v$-DINO | $\mathbf{79.4_{\pm0.0}}$ | $\mathbf{80.4_{\pm0.2}}$ | 80.9 |
| $v$-MAE | $66.0_{\pm0.0}$ | $74.6_{\pm0.2}$ | 76.2 |

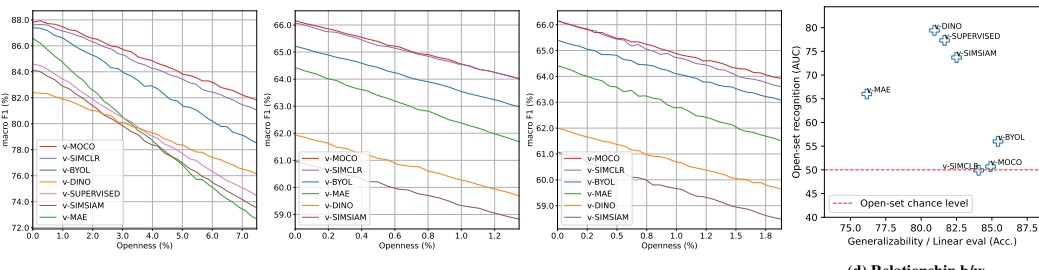

(a) UCF/HMDB    (b) Kinetics400/UCF    (c) Kinetics400/HMDB    (d) Relationship b/w closed-set and open-set performance.

Figure 7: (a-c) Comparing **open macro-F1 scores** of finetuned models vs. **openness** (openness is measured as the ratio of unknown to known classes). (d) The relationships between **closed-set** and **open-set** recognition performance of frozen pretrained encoders.

The results in Table 6 (a) demonstrate that the models struggle in open-set recognition when using Kinetics400 as closed-set in comparison to UCF101. This is due to the large number of known classes in Kinetics400 compared to the unknowns (400 known classes in Kinetics400 vs. 31 and 22 unknown classes from UCF and HMDB, respectively). Therefore, the models tend to become over-confident and make false predictions. Evident from Table 6 (a), contrastive methods are robust to open-set recognition in all setups. This is likely due to the auxiliary information contributed by the negative samples used in these methods. To verify this hypothesis, we analyze the open macro-F1 scores vs. openness of the video models following [45, 44] by incrementally adding more unknown classes. The results in Figure 7 (a-c) show that both $v$-SimCLR and $v$-MoCo consistently achieve better performances compared to others. An additional statistical analysis on the performance of contrastive methods in open-set recognition is presented in Appendix D.7, which confirms their better performance relative to other methods.

Next, instead of end-to-end finetuning, we use the frozen encoders to train a linear head for open-set recognition. The top 3 closed-set performers, $v$-BYOL, $v$-MoCo, and $v$-SimCLR (see Figure 7 (d) or U101* in Table 6 (b)), show the worst performance in open-set recognition. However, the pretrained encoders, $v$-SimSiam, $v$-DINO, and $v$-Supervised, which are considered weaker in closed-set, perform better in open-set recognition, which might be due to their lack of over-confidence. $v$-DINO outperforms other methods in open-set recognition achieving $79.4\%$ while the performance of $v$-SimCLR and $v$-MoCo drops to almost chance levels. In comparison, $v$-MoCo reports $84.9\%$ in standard closed-set recognition, whereas $v$-DINO achieves $80.9\%$. The results presented in Figure 7 (d) suggest that there is a trade-off between closed-set and open-set recognition performances, particularly when frozen encoders are used without finetuning.

**Highlights:** (*a*) Contrastive methods demonstrate superior performance in open-set recognition when finetuned. (*b*) There is a trade-off between closed-set and open-set recognition performance when frozen pretrained encoders are used. (*c*) Strong frozen encoders ($v$-MoCo, $v$-SimCLR) have no open-set generalization performance due to their over-confident predictions. (*d*) On the other hand, slightly weak VSSL frozen encoders ($v$-DINO, $v$-SimSiam) show better open-set performance, while $v$-MAE seems to perform poorly in both settings.

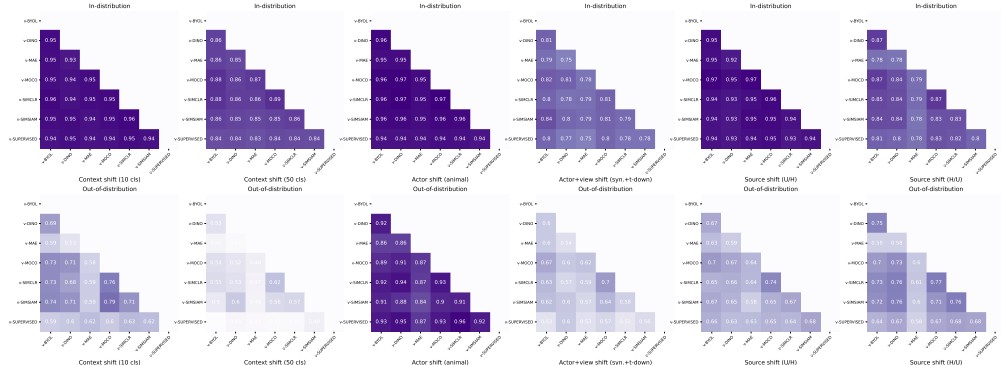

Figure 8: The **decision similarity** between the video models in InD (top) vs. OoD (bottom). The lighter color indicates less similarity.

**Q4: Decision similarity under distribution shifts**

To measure 'decision similarity' between the video models, we evaluate whether the models make similar predictions, regardless of their correctness. The results presented in Figure 8 demonstrate that decision similarity varies between the VSSL methods both InD and OoD, but is generally lower in OoD settings. Specifically, while we observe only a slight decrease in decision similarity in the case of animal domain actor shift, we observe significant drops in decision similarity in the case of context shifts and source shifts. Moreover, the results reveal that the decision similarity between supervised and self-supervised methods significantly drops under distribution shifts, indicating that under such shifts, VSSL methods make considerably different decisions than supervised learning. We also observe low decision similarity between $v$-MAE and other VSSL methods, which can be due to the generative nature of $v$-MAE vs. the others. Lastly, we note that contrastive methods tend to have greater decision similarities with each other than the similarities observed among non-contrastive methods. Additional results are presented in Appendix D.

> **Highlights:** (*a*) The decision similarity of video models decreases under distribution shifts, which further varies based on the type of shift. (*b*) Context and source shifts cause the most dissimilarity between decisions. (*c*) Overall, the predictions between the supervised and self-supervised methods, as well as between $v$-MAE and other VSSL methods exhibit the least similarity.

## 7 Discussion and Summary

**Limitations.** We consider contrastive, non-contrastive, and generative VSSL methods in our work, as they are well-established and have demonstrated strong performance in previous studies [72, 1, 3, 5] on various video benchmarks [1, 87, 72]. However, there exists another category of VSSL methods which uses pretext tasks for self-supervision, e.g., rotation prediction [88], frame and clip order prediction [14, 89], motion prediction [90], and others [91, 92, 93, 94]. While these methods are not included in our study, they are worth exploring in future research. Moreover, our work primarily focuses on various *self-supervised methods* as opposed to network architectures. It would be valuable to further investigate VSSL under distribution shifts with larger Transformer [95] or convolutional [96] architectures , which we could not perform due to resource constraints. Nonetheless, our findings serve as a foundation for future studies. We discuss the broader impact of our work in Appendix A.

**Summary.** In this work, we thoroughly examine the behavior of VSSL methods under real-world distribution shifts that commonly occur in videos due to changes in context, viewpoint, actor, and source. Moreover, our study delves into investigating the generalizability of VSSL methods in zero-shot and open-set recognition. To rigorously evaluate the robustness of video models, we introduce a comprehensive OoD test bed curated from existing literature. This test bed is carefully designed to stress test the robustness of video models and provide a comprehensive evaluation of their capabilities. Our study uncovers a wide range of interesting dynamics of various VSSL methods under different distribution shifts, which can be instrumental in guiding future research and algorithm development in video representation learning. To the best of our knowledge, this is the first work to systematically investigate VSSL under real-world distribution shifts.

## Acknowledgments and Disclosure of Funding

PS and AE are grateful to the Bank of Montreal and Mitacs for funding their research. PS and AE are also thankful to SciNet HPC Consortium for helping with the computation resources. We thank Vandad Davoodnia (Queen's University) and Ananth Balashankar (Google Research) for fruitful discussions and feedback.

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

# Appendix

The organization of the supplementary material is as follows:

# A  Broader impact

In this work, we conduct a thorough study on the behavior of Video Self-supervised Learning (VSSL) methods in the face of distribution shifts. We believe this work has far-reaching implications for advancing the field of video representation learning and its real-world applications. This work is of utmost importance primarily for the following two reasons.

**Real-world deployment:** Video models have a wide range of real-world applications across various domains. These applications include surveillance and security, autonomous vehicles, sports analysis, video content recommendation, human-computer interaction, video captioning and summarization, among many others. However, in real-world scenarios, distribution shifts pose a significant challenge, where the distribution of the data that the system encounters may differ from the training data distribution. Understanding how VSSL algorithms behave under real-world distributional shifts is crucial for ensuring their reliable performance and generalization capabilities for real-world applications. By studying VSSL methods under distribution shifts, we identify the limitations and challenges faced by the popular VSSL methods. We believe our work is a step towards enabling the development of robust and reliable solutions for real-world deployment.

**Algorithmic robustness:** Our study provides valuable insights into the algorithmic robustness of VSSL methods, advancing our understanding of VSSL algorithms. By systematically exploring the behavior of these methods under distribution shifts, we have uncovered intriguing findings and observed interesting behaviors which are previously unknown. Our work not only highlights the strengths of VSSL methods but also identifies their weaknesses in different scenarios. We find that different VSSL methods exhibit varying levels of robustness when confronted with different types of distribution shifts. Some methods demonstrate resilience against certain shifts, while they prove to be vulnerable to others. These findings pave the way for future research to develop robust frameworks that leverage the strengths of different approaches, thereby enhancing the overall performance of VSSL algorithms.

# B  Video self-supervised learning

## B.1  Methods

In the following section, we provide a comprehensive overview of our implementation of the VSSL methods explored in this study. We define an encoder $f(\cdot)$, which is composed of a Transformer backbone $\theta(\cdot)$ and an MLP projection head $h(\cdot)$. We further define the auxiliary network components, a target encoder $f_t(\cdot)$ (utilized in $v$-MoCo, $v$-BYOL, and $v$-DINO), a decoder $f_d(\cdot)$ (employed in $v$-MAE), and a predictor head $f_p(\cdot)$ (utilized in $v$-BYOL, $v$-SimSiam, and $v$-MoCo). Note, $f_t(\cdot)$ and $f_d(\cdot)$ are built on the Transformer architecture, while $f_p(\cdot)$ utilizes an MLP head. For given samples $v$, we generate two augmented views as $v_1$ and $v_2$, where $v_1$ and $v_2$ are randomly sampled from different timestamps and differently augmented. Moreover, an augmented view $v_i \in \mathbb{R}^{T \times H \times W \times C}$ can be expressed into a sequence of P spatio-temporal patches of size $t \times h \times w \times C$, where, $P = T/t \times H/h \times W/w$. We project the spatio-temporal patches to a linear layer followed by feeding to the encoders. Following, we briefly summarize the training algorithms of these VSSL methods along with their design differences. An overview of these frameworks is presented in Figure S1.

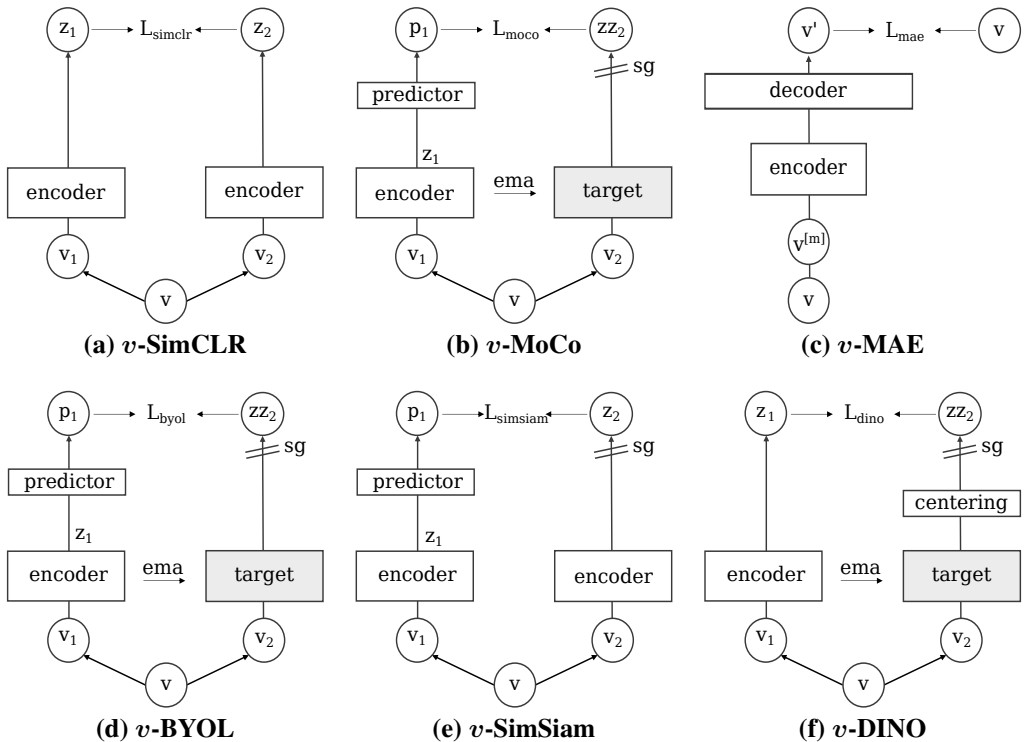

(a) $v$-SimCLR  (b) $v$-MoCo  (c) $v$-MAE

(d) $v$-BYOL  (e) $v$-SimSiam  (f) $v$-DINO

Figure S1: A simplified version of the video self-supervised methods are presented.

$v$-**SimCLR.** The objective of $v$-SimCLR is to maximize the similarity between positive pairs and minimize the similarity between negative pairs using the InfoNCE loss function. Given two augmented views $v_1$ and $v_2$, we obtain $z_1$ and $z_2$ as $f(v_1)$ and $f(v_2)$. The L2 normalized cosine similarity scores, referred to as $\mathrm{logits}$, are then calculated as $\mathrm{sim}(z_1, z_2)$, where $\mathrm{sim}(z_1, z_2) = \frac{z_1^\top z_2}{||z_1||||z_2||}$. The pretraining loss is calculated as $\mathcal{L}_{simclr} = \mathrm{InfoNCE(logits)}$ as per Equation S1, where $k+$ refers to the positive embeddings (i.e., embeddings of the augmented views) and $k-$ refers to the negative embeddings (embeddings of other samples in a minibatch).

$$\mathrm{InfoNCE(logits)} = -\log \frac{\sum_{k \in \{k+\}} \exp(\mathrm{logits}/\tau)}{\sum_{k \in \{k+, k-\}} \exp(\mathrm{logits}/\tau)}, \text{where } \tau \text{ is temperature.} \tag{S1}$$

**$v$-MoCo.** Similar to $v$-SimCLR, $v$-MoCo also calculates InfoNCE loss for given query-key pairs. We obtain the query embeddings as $p_1 = f_p(f(v_1))$ and its corresponding key embeddings are obtained as $zz_2 = \text{sg}(f_t(v_2))$, where sg refers to stop-gradient operation. In Equation S1, the positive and negative key embeddings $\{k+, k-\} \in zz_2$ and the query embedding $q \in p_1$. To calculate a symmetrized loss we also obtain $p_2$ and $zz_1$. The training objective $\mathcal{L}_{moco}$ is defined as $\text{InfoNCE}(\text{sim}(p_1, zz_2)) + \text{InfoNCE}(\text{sim}(p_2, zz_1))$. The encoder $f$ is updated using $\mathcal{L}_{moco}$, and the target encoder $f_t$ is updated using the exponential moving average (EMA) [56] as $f_t \leftarrow \alpha f + (1-\alpha)f$, where $\alpha$ is the EMA coefficient.

**$v$-BYOL.** Different from $v$-SimCLR and $v$-MoCo, $v$-BYOL does not require any negative pairs and only relies on positive views. Therefore, we obtain $p_1$ and $zz_2$ as $f_p(f(v_1))$ and $\text{sg}(f_t(v_2))$ to calculate the similarity scores as $\text{sim}(p_1, zz_2)$. Moreover, to obtain a symmetrized loss, we also calculate $\text{sim}(p_2, zz_1)$ and define total loss $\mathcal{L}_{byol} = -\text{sim}(p_1, zz_2) - \text{sim}(p_2, zz_1)$. Please note, we use $\mathcal{L}_{byol}$ to train the $f$, and the weights of $f_t$ are updated using an EMA similar to $v$-MoCo.

**$v$-SimSiam.** The approach for $v$-SimSiam is very similar to the $v$-BYOL, except it does not use $f_t$, and directly uses the representations of the $f$ as the target. We obtain $p_1$ and $z_2$ as $f_p(f(v_1))$ and $\text{sg}(f(v_2))$ to calculate the similarity scores as $\text{sim}(p_1, z_2)$. Similarly, we also obtain $\text{sim}(p_2, z_1)$ and define total loss $\mathcal{L}_{simsiam} = -\text{sim}(p_1, z_2) - \text{sim}(p_2, z_1)$.

**$v$-DINO.** Similar to $v$-BYOL and $v$-MoCo, $v$-DINO uses a target encoder $f_t(\cdot)$ but does not use a predictor head. We obtain $z_1$ and $zz_2$ as $f(v_1)$ and $\text{sg}(f_t(v_2) - \mathbf{C})$ respectively, where $\mathbf{C}$ refers to the Centering operation which can be interpreted as adding a bias term, preventing the model from collapsing [26]. Next, we calculate the cross-entropy (CE) error between $z_1$ and $zz_2$ as $\text{CE}(z_1, zz_2) = -\text{softmax}(zz_2/\tau_t)\log(\text{softmax}(z_1/\tau_s))$, where $\tau_t$ and $\tau_s$ denote the temperatures used to sharpen the representations of the target and base encoder respectively. Finally, we calculate loss $\mathcal{L}_{dino} = \text{CE}(z_1 + zz_2) + \text{CE}(z_2, zz_1)$ which is used to update $f$ and the weights of $f_t$ are updated using EMA similar to $v$-BYOL and $v$-MoCo.

**$v$-MAE.** Unlike the methods mentioned above, $v$-MAE takes a single view and performs masking and reconstruction using an autoencoder architecture [27, 1, 57]. Moreover, different from the other models, the encoder does not contain a projection head, i.e., $f = \theta$, instead uses a decoder $f_d$ for reconstruction. For a given sample $v$, we obtain masked tokens $v^{[m]}$ as $\{v^i | m^i = 1\}_{i=1}^P$ and the corrupted inputs as $\hat{v} = \{v | m^i = 0\}_{i=1}^P$. We calculate the reconstruction loss as $\mathcal{L}_{mae} = (f_d(\theta(\hat{v})) - v^{[m]})^2$, which is used to train the $\theta$ and $f_d$.

After pretraining using the aforementioned methods, we discard the auxiliary network components and utilize only $\theta$ for downstream evaluations.

## B.2 Implementation details

**Datasets.** We use 2 popular large-scale datasets Kinetics400 [61] and Kinetics700 [62] for pretraining. Kinetics400 consists of 240K samples including 400 action classes, whereas, Kinetics700 consists of 537K training samples spread over 700 action classes. Following [97, 42], we discard the overlapping classes between Kintics700 and UCF101 [66], HMDB51 [67] from pretraining, which results in approximately 480K samples spread over 663 action classes. We follow such a setup so that the pretrained encoder can be used for zero-shot recognition on full UCF101 and HMDB51.

**Pretraining.** To ensure a fair comparison between the VSSL methods, we pretrain them in identical setups with necessary adjustments in hyperparameters. Specifically, we keep the encoder, inputs, batch size, and maximum number of pretraining epochs identical for all the methods. However, we adjust the data augmentation strategy and auxiliary network components (e.g., predictor head, projector head, decoder) as required to achieve their best performance on 2 validation sets Kinetics400 and UCF101. For example, the projector head of $v$-DINO is based on the configuration proposed in [26], whereas, the projector head for the other Siamese frameworks is made of an MLP head similar to [24, 25]. The details of the frameworks are presented in Table S1. Considering the widespread use of Transformers in vision, VSSL methods are implemented using ViT-B [75] as the encoder with a patch size of $4 \times 16^2$. Videos are downsampled to 8 FPS and resized to $3 \times 112^2$ frames. Except for $v$-MAE, all VSSL methods are fed with 16 frames in each view and 32 frames are fed as input to $v$-MAE. We use AdamW [98, 99] optimizer with cosine learning rate scheduler and train all methods up to 800 epochs with a batch size of 768. We present the hyperparameters in Table S3. All the methods are pretrained using 8 V100 32 GB GPUs in parallel. We present the computational specifics

in Table S2, which shows that $v$-MAE is the most computationally efficient framework amongst all the VSSL methods.

Table S1: Architecture details of the VSSL frameworks.

| | $v$-SimCLR | $v$-MoCo | $v$-BYOL | $v$-SimSiam | $v$-DINO | $v$-MAE |
|---|---|---|---|---|---|---|
| Encoder | | | ViT-B | | | |
| Target Encoder | ✗ | ✓ | ✓ | ✗ | ✓ | ✗ |
| Predictor (# layers) | ✗ | 1 | 2 | 1 | ✗ | ✗ |
| Projector (# layers) | 4 | 3 | 3 | 4 | 3 | ✗ |
| Decoder (# depth) | ✗ | ✗ | ✗ | ✗ | ✗ | 4 |

Table S2: Computational details of VSSL pretraining.

| | $v$-SimCLR | $v$-MoCo | $v$-BYOL | $v$-SimSiam | $v$-DINO | $v$-MAE |
|---|---|---|---|---|---|---|
| Encoder Param | | | 88M | | | |
| Total Params | 106M | 110M | 114M | 114M | 96M | 96M |
| Pretraining Flops | 36G | 72G | 72G | 36G | 72G | 10G |
| Time/100 epochs | | | 23Hrs. | | | 11 Hrs. |

Table S3: Hyperparameters of VSSL pretraining.

| | $v$-SimCLR | $v$-MoCo | $v$-BYOL | $v$-SimSiam | $v$-DINO | $v$-MAE |
|---|---|---|---|---|---|---|
| Batch size | | | 768 | | | |
| Crop scale | [0.08, 1.0] | | | [0.2, 0.766] | | [0.5, 1] |
| Clip duration | | | 2.0 | | | 4.0 |
| Color | | | [0.6, 0.6, 0.6, 0.15] | | | − |
| Grayscale | | | 0.2 | | | − |
| Gaussian blur | | | 0.5 | | | − |
| Hroizontal flip | | | 0.5 | | | |
| Epochs | | | 800 | | | |
| Warmup epochs | | | 30 | | | |
| Optimizer | | | AdamW [0.9, 0.95] | | | |
| Weight decay | | | 0.05 | | | 0.05 to 0.5 |
| Learning rate (Base LR) | $2e^{-4}$ | $3e^{-4}$ | $3e^{-4}$ | $1e^{-4}$ | $3e^{-4}$ | $3e^{-4}$ |
| Pred LR ($\times$ Base LR) | − | 10× | 10× | 10× | − | − |
| EMA (cosine sch.) | − | 0.998−1 | 0.997−1 | − | 0.997−1 | − |
| Online temp. (fixed) | 0.1 | 0.1 | − | − | 0.1 | − |
| Target temp. (cosine sch.) | − | − | − | − | 0.04−0.07 | − |
| Centering momentum | − | − | − | − | 0.9 | − |
| Masking ratio | − | − | − | − | − | 85% |

## B.3 Additional insights

**Aggressive cropping.** As discussed in the main paper, aggressive cropping leads to viewpoint invariance, therefore we experiment with various multi-crop ratios commonly used in the literature [22, 72] across all six VSSL methods. We present the best setups corresponding to individual methods in Table S3. We observe that aggressive cropping negatively impacts the performance of $v$-MAE. This is primarily because reconstructing a local crop serves as a weak pseudo task due to limited pixel-level variation, resulting in suboptimal representation learning. Additionally, as depicted in Figure S2, the impact of cropping varies between contrastive and non-contrastive methods. Very high aggressive cropping ratios $(0.08 - 1.0)$ prove beneficial for contrastive methods, while slightly less aggressive cropping ratios $(0.2 - 0.766)$ yield better results for non-contrastive methods.

**Pretraining epoch vs. accuracy.** We track the performance of different videl learning methods on two downstream benchmarks, Kinetics400 and split-1 of UCF101. In particular, Kinetics400 is used for finetuning and UCF101 for linear evaluation. The results are presented in Figure S3 shows that while $v$-MAE achieves the lowest linear evaluation accuracy, it shows the best performance when finetuned. Moreover, $v$-BYOL, $v$-SimCLR, and $v$-MoCo exhibit superior performance compared to $v$-DINO and $v$-SimSiam in both the linear evaluation and finetuning setups. Interestingly, contrastive

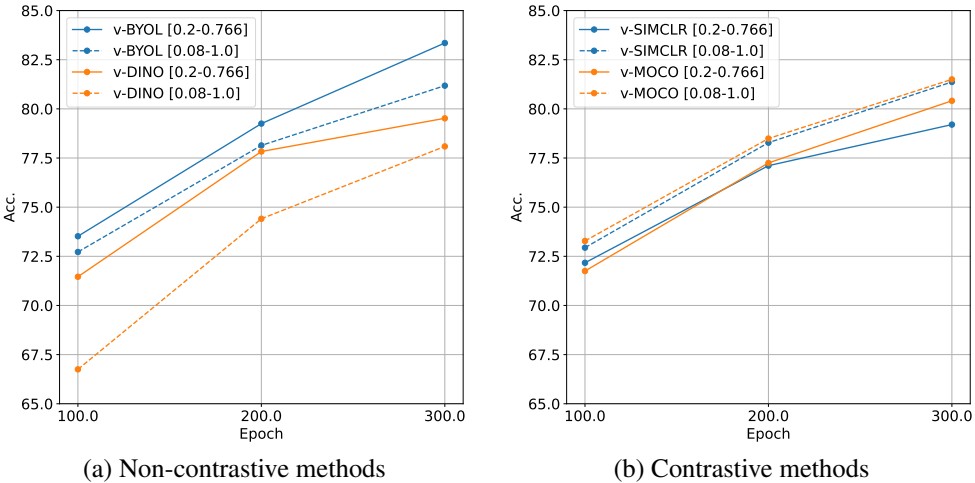

(a) Non-contrastive methods        (b) Contrastive methods

Figure S2: The **impact of cropping strategies** slightly differs between non-contrastive methods ($v$-BYOL, $v$-DINO) and contrastive methods ($v$-SimCLR, $v$-MoCo). Aggressive cropping with a crop ratio of $[0.08, 1.0]$, enhances the performance of contrastive methods, while a ratio of $[0.2, 0.766]$ yields the best performance on non-contrastive methods. We show the performance with a crop ratio of $[0.2, 0.766]$ with solid lines and $[0.08, 1.0]$ with dashed lines. Evaluation is carried out on UCF101 using a linear SVM.

methods $v$-SimCLR and $v$-MoCo show comparable performance among them, however, there is a notable performance gap between the non-contrastive methods. Specifically, $v$-BYOL outperforms both $v$-SimSiam and $v$-DINO by a significant margin in both the linear evaluation and finetuning setups. Moreover, Figure  indicates minimal or no improvements between 600 to 800 epochs. Therefore, we refrain from training the models beyond 800 epochs.

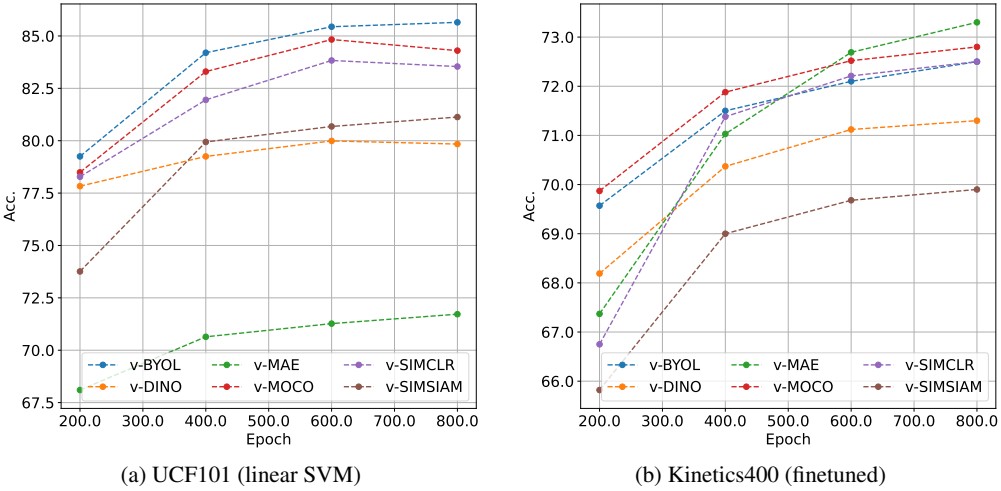

(a) UCF101 (linear SVM)        (b) Kinetics400 (finetuned)

Figure S3: The **pretraining epochs vs. top-1 accuracies** are presented. VSSL methods are validated in two setups. Kinetics400 is used for finetuning and UCF101 is used for linear evaluation. $v$-BYOL, $v$-SimCLR, and $v$-MoCo show superior performance compared to $v$-DINO and $v$-SimSiam in both setups. Interestingly, $v$-MAE achieves the lowest linear evaluation accuracy, while performing the best when finetuned.

**Spatial vs. temporal dependency.** In addition to the out-distribution tests discussed in the main paper, we also conduct several interesting analyses to understand the dependency of these VSSL

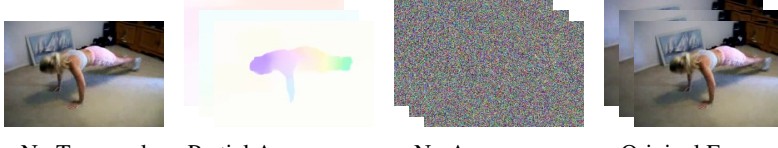

No Temporal    Partial Appearance    No Appearance    Original Frames

Figure S4: Sample frames from UCF101 depicting different spatio-temporal dependency scenarios.

Table S4: Comparing video models on controlled temporal and appearance information.

| Method | No Temporal | Partial Appe. | No Appe. | Original |
|---|---|---|---|---|
| | Lin. / FT. | Lin. / FT. | Lin. / FT. | Lin. / FT. |
| $v$-Supervised | 43.7 / 36.6 | 4.1 / 4.9 | 1.3 / 1.0 | 81.7 / 87.4 |
| $v$-SimCLR | 57.1 / 46.5 | 4.3 / 4.9 | 0.9 / 1.1 | 84.1 / 90.3 |
| $v$-MoCo | 56.0 / 48.8 | 4.7 / 6.7 | 1.7 / 1.1 | 84.9 / **90.7** |
| $v$-BYOL | **58.2 / 50.9** | 3.5 / 6.0 | 1.1 / **2.6** | **85.4** / 90.2 |
| $v$-SimSiam | 47.7 / 28.3 | **5.6** / 4.5 | **2.0** / 1.5 | 82.5 / 87.2 |
| $v$-DINO | 41.9 / 38.6 | 3.2 / **7.7** | 1.2 / 1.4 | 80.9 / 88.5 |
| $v$-MAE | 5.6 / 25.7 | 3.1 / 5.3 | 1.1 / 2.7 | 76.2 / 90.5 |

methods towards spatial vs. temporal representations. To explore the spatial vs. temporal dependency of the VSSL methods, we use UCF101 as the base validation dataset and create 3 synthetic variant of it, (*i*) **No Temporal**: we completely remove the temporal information from the input and treat single frames as an input. In particular, we obtain 80 frames per video that represent 10 seconds of video sampled at 8 FPS. (*ii*) **Partial Appearance**: Next, to suppress the spatial information, we obtain the optical flow [100] from the original videos. (*iii*) **No Appearance**: Following [101], we also obtain the videos of no appearance, i.e., the static frames have no meaning and actions are present in the temporal dimension. We present sample frames of these variants in Figure S4.

The results are presented in Table S4 show several intriguing observations. First, removing the temporal dimension from the videos leads to a significant drop in performance in all the methods. Amongst all the video models, $v$-MAE suffers the worst in the absence of temporal information in both linear and finetune setups, which shows its overly dependent nature on temporal information and it is a weak spatial learner (also discussed in the main paper). Second, in the absence of temporal information, the frozen encoder outperforms the finetuned variant even in the case of $v$-Supervised. We also notice the performance of the non-contrastive methods significantly varies under such setups, for example, while $v$-BYOL achieves the best performance in linear evaluation, $v$-DINO performs worse other than $v$-MAE. Additionally, our tests under partial or no appearance reveal that none of the video models is capable of understanding actions just from the temporal information. In fact, the performance of video models drops near chance level ( 1%) in no appearance setup and achieve slightly better (3 − 7%) in partial appearance setup.

## C  Distribution Shifts

### C.1  Overview

We provide a summary of our proposed out-of-distribution test bed in Table S5, and the details of our setup are described in the following subsections. Due to the large number of experiments conducted across various setups, we will release configuration files for individual evaluation on the project website rather than sharing them here.

Table S5: An overview of our **out-of-distribution test-bed**. #Samples are in following order training samples/InD test samples/OoD test samples; In zero-shot and open-set recognition, #Classes indicates the number of InD/OoD classes and for others the number of InD and OoD classes remains the same.

| # Distribution Shift | InD | OoD | #Classes | #Samples |
|---|---|---|---|---|
| 1. Context shift (10 classes) | Kinetics400 | Mimetics10 | 10 | 5930/494/136 |
| 2. Context shift (50 classes) | Kinetics400 | Mimetics50 | 50 | 34K/2481/713 |
| 3. Viewpoint shift (egocentric) | CharadesEgo | CharadesEgo | 157 | 34K/9386/9145 |
| 4. Viewpoint shift (surveillance) | MiT-v2 | TinyVirat-v2 | 14 | 41K/1400/2644 |
| 5. Actor shift (animal) | Kinetics400 | ActorShift | 7 | 15K/1018/165 |
| 6. Viewpoint + Actor shift (top-down+synthetic) | MiTv2 | Sims4Action | 6 | 19K/600/950 |
| 7. Source shift (UCF/HMDB) | UCF | HMDB | 17 | 1877/746/510 |
| 8. Source shift (HMDB/UCF) | HMDB | UCF | 17 | 1190/510/746 |
| 9. Zero-shot (K400/UCF) | Kinetics400 | UCF | 400/31 | 240K/20K/3965 |
| 10. Zero-shot (K400/HMDB) | Kinetics400 | HMDB | 400/22 | 240K/20K/3288 |
| 11. Zero-shot (K400/RareAct) | Kinetics400 | RareAct | 400/149 | 240K/20K/1961 |
| 12. Zero-shot (K700/UCF) | Kinetics700 | UCF | 663/101 | 480K/ − /13K |
| 13. Zero-shot (K700/HMDB) | Kinetics700 | HMDB | 663/51 | 480K/ − /6.7K |
| 14. Zero-shot (K700/RareAct) | Kinetics700 | RareAct | 663/149 | 480K/ − /1961 |
| 15. Open-set (K400/UCF) | Kinetics400 | UCF | 400/31 | 240K/20K/3965 |
| 16. Open-set (K400/HMDB) | Kinetics400 | HMDB | 400/22 | 240K/20K/3288 |
| 17. Open-set (U101/HMDB) | UCF101 | HMDB | 101/34 | 9537/3783/4366 |

### C.2  Context shift

We use the two splits of Mimetics (i.e., Mimetics10 [102] and Mimetics50 [28]) as the OoD validation benchmark to study video models under context shift. Mimetics consists of a subset of action classes from Kinetics400, where the context in the videos is either partial or misleading or completely absent. We train the pretrained encoders using videos of corresponding action classes from Kinetics400, and subsequently perform InD and OoD validations using Kinetics400 and Mimetics, respectively. For linear evaluation, we extract the fixed features by taking the mean of all the patches from the last layer of the encoder. We then employ an SVM with a linear kernel for classification. In the finetuning stage, we add a fully-connected layer on top of the encoder and train it end-to-end using the AdamW optimizer and cross-entropy error. The top-1 accuracy is reported for both InD and OoD validation. Please note, the different VSSL methods are individually tuned to achieve their best performance.

### C.3  Viewpoint shift

We study three types of viewpoint shifts, (*i*) egocentric view, (*ii*) surveillance camera view, and (*iii*) top-down view. We use CharadesEgo [36], TinyVirat-v2 [63], and Sims4Action [64] as the OoD benchmark for egocentric view, low-resolution surveillance camera view, and top-down view respectively. CharadesEgo is a collection of paired third-person and first-person videos of human activities. Therefore, we use the corresponding third-person videos from CharadesEgo [36] for InD training and validation split while the egocentric view is used as OoD validation. We use MiT-v2 [69] to create the corresponding InD split TinyVirat-v2 and Sims4Action [64]. MiT-v2 is a large-scale video benchmark of 1M videos spread over 339 different human actions. We use the videos of overlapping classes between MiT-v2 and TinyVirat-v2 to study surveillance camera viewpoint shift.

Similarly, the videos of overlapping classes between MiT-v2 and Sims4Action are used to study top-down viewpoint shifts. The class assignments between InD and OoD benchmarks are mentioned at the end of this subsection.

For training the models on CharadesEgo, we adopt the approach proposed in [103]. In this setup, instead of using one-hot encoded labels, we utilize the text descriptions associated with the videos. We generate tokens from these descriptions and train a video-text encoder using an InfoNCE objective function [55]. In particular, we use the DistilBERT base [104] as the text encoder along with the pretrained ViT-B is used as the video encoder. We redirect the readers to [103], to know more about the training scheme of CharadesEgo. In the case of linear evaluation, the above-mentioned setup remains the same, except we keep the video encoder frozen. Moreover, our training strategy with MiT-v2 to study surveillance camera viewpoint and top-down viewpoint shifts follow a standard finetuning setup using cross-entropy error [72, 2, 4, 6, 78, 79, 7]. However, due to the difference in the nature of the datasets, as TinyVirat contains multiple actions per video, whereas, MiT-v2 consists of single actions per video, we make an adjustment in the prediction layer. Instead of using a Softmax layer, we utilize a Sigmoid layer to predict multiple possible outputs for a given input video. We set the threshold for classification at 0.5. Similarly, the setup for linear evaluation on TinyVirat remains the same, except the video encoder is kept frozen. Next, in the case of Sims4Action, we use SVM for linear evaluation. We report mean average precision and F1-score for CharadesEgo and TinyVirat, respectively, and top-1 accuracy for the rest of them.

**TinyVirat-v2 and MiT-v2.** opening: opening, pull: pulling, activity_carrying: carrying, entering: entering, exiting: exiting, loading: loading, talking: talking, activity_running: running, riding: riding, closing: closing, activity_walking: walking, push: pushing, activity_standing: standing, specialized_talking_phone: telephoning.

**Sims4Action and MiT-v2.** cook: cooking, drink: drinking, eat: eating, read_book: reading, use_phone: telephoning, walk: walking.

## C.4  Actor shift

We study actor shift in two setups, i.e., animal domain and synthetic domain. We use the ActorShift [65] as the OoD validation set to test generalizability on animal actions, while using the video of overlapping classes from Kinetics700 [62] as InD training and validation. We use SVM for linear evaluation and follow standard techniques [72, 2, 4, 6, 78, 79, 7] as discussed earlier for finetuning. We report top-1 accuracy for both InD and OoD validation. We present the overlapping classes between Kinetics700 and ActorShift at the end of this subsection. The setup for synthetic domain shift is already discussed in the previous subsection (Viewpoint shift).

**ActorShift and Kinetics700.** sleeping, watching tv, eating, drinking, swimming, running, opening door.

## C.5  Source shift

To study source shift, we use the videos from the common classes of UCF101 and HMDB51. While using the UCF as the training split, we use the corresponding videos from HMDB as the OoD validation set and vice versa. Following standard protocol [72, 2, 4, 6, 78, 79, 7], we use SVM for linear evaluation and the pretrained encoder along with a fully-connected layer for finetuning. We report the top-1 accuracy for both setups. The class assignments are as follows.

**UCF101 and HMDB51.** FrisbeeCatch: catch, RockClimbingIndoor: climb, Diving: dive, (BasketballDunk, Basketball): dribble, Fencing: fencing, GolfSwing: golf, (HandstandPushups, HandstandWalking): handstand, (LongJump, JumpingJack): jump, SoccerPenalty: kick_ball, PullUps: pullup, Punch: punch, PushUps: pushup, Biking: ride_bike, HorseRiding: ride_horse, ThrowDiscus: throw, WalkingWithDog: walk, Archery: shoot_bow,

## C.6  Zero-shot

We perform zero-shot in two setups: (1) we train the video-text encoder on Kinetics400, while the non-overlapping classes from UCF101 and HMDB51 are used for OoD validation [105]. (2) We train the models using the videos from Kinetics700 that do not share classes with UCF101 and HMDB51,

while the full UCF101 and HMDB51 are used for OoD validation. Moreover, in both cases, we also use RareAct as the OoD benchmark, which consists of rare or unusual video actions.

Since, the pretraining of VSSL methods does not employ language, first, a video-text encoder is jointly trained to learn the mapping between video and text representations. Following [42], we use Word2Vec [80] to extract the word embeddings from the labels, which are then fed to a text encoder consists of an MLP head. We measure the similarity between the video and text pairs as the dot product of L2 normalized video-text embeddings, referred to as logits. Next, the video text encoders are trained to minimize the Softmax cross-entropy error using the obtained logits and the class labels. We redirect readers to [42] for additional details of the training method. Please note, we also try with a stronger language model (e.g., BERT[106], DistilBERT[104]) than Word2Vec, however, it does not improve the performance, in fact, slightly worsens it. Since the labels in Kinetics are typically composed of only a few words (mostly less than 2), the average pooled word embeddings work well in such cases. This finding is consistent with the observations of [42]. Moreover, we perform zero-shot in both setups when the pretrained video encoder is kept frozen and finetuned in an end-to-end manner.

## C.7 Open-set

We perform open-set recognition in two setups: (1) We use the Kinetics400 for closed-set recognition, while the non-overlapping classes from the UCF101 and HMDB51 are used for open-set, similar splits that have been in zero-shot. (2) We use UCF101 and the non-overlapping classes of HMDB for closed-set and open-set recognition, respectively. We obtain the non-overlapping classes by manual inspection, presented at the end of this subsection.

To enable the models for open-set recognition, we follow [45] and train the pretrained encoders with the DEAR loss which aims to calibrate the uncertainty to ensure the models show high uncertainty towards the unknown and low uncertainty towards the known. We redirect readers to [45] for more information about open-set training. We conduct open-set recognition in both finetuned and linear evaluation setups. We observe that both supervised training and linear evaluation setups using VSSL encoders did not converge when Kinetics400 was used for closed set recognition. This is likely because the models tend to become over-confident due to the large number of known classes in Kinetics400 compared to the unknowns. To measure the models' performance we report the area under the ROC curve (AUC) for open-set recognition and accuracy for closed-set recognition.

**HMDB51.** The non-overlapping classes from HMDB51 used in open-set recognition while using UCF101 as the closed-set: brush_hair, cartwheel, chew, clap, climb_stairs, drink, eat, fall_floor, flic_flac, hit, hug, kick, kiss, laugh, pick, pour, push, run, shake_hands, shoot_ball, shoot_bow, shoot_gun, sit, situp, smile, smoke, somersault, stand, swing_baseball, sword, sword_exercise, talk, turn, wave.

## C.8 Toy experiments

In all of our toy experiments, we use the pretrained frozen encoders to extract the fixed representations, which are then used for K-means clustering[2]. Additionally, we apply the Hungarian algorithm [107] for assigning the formed cluster to the targets and measure the accuracy. The details of the datasets used to conduct the toy experiments are as follows.

### C.8.1 ToyBox

The ToyBox [108] consists of egocentric views of 9 distinct temporal transformations of different toy objects. The transformations include positive rotations, negative rotations, and translations across the $x$, $y$, and $z$ axis. A few representative frames illustrating such transformations are presented in Figure S5. Using the pretrained frozen encoders, we aim to correctly cluster different transformations across various objects. Moreover, as the static spatial information is non-discriminative to the temporal transformations, we aim to form clusters solely based on temporal representations. The main motivation behind this experiment is to disentangle spatial representations from the embedding space to independently evaluate the models on their ability to learn temporal dynamics.

---

[2]`sklearn.cluster.KMeans`

### C.8.2 COIL100

We use COIL100 [34] to study viewpoint invariance which consists of images of different objects captured from different camera angles. We particularly choose an image-based dataset for this experiment to restrict any discriminatory temporal features in the embedding space that may attribute to the outcome of the models' prediction. We present a few representative frames in Figure S6

### C.8.3 STL10

We use an image dataset STL10 [71] to test the robustness of the pretrained encoder against low-resolution inputs. We systematically reduce the resolution from $112^2$ to $16^2$, please see a few examples in Figure S7. Similar to our viewpoint shift setup, we choose an image dataset to restrict any influence of temporal representations in the outcome.

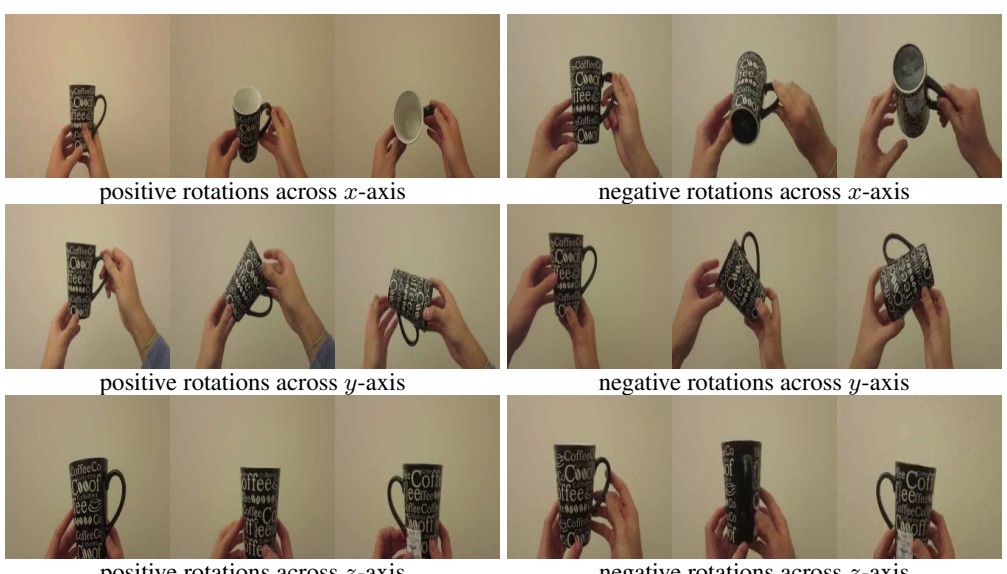

|  positive rotations across $x$-axis | negative rotations across $x$-axis |
| positive rotations across $y$-axis | negative rotations across $y$-axis |
| positive rotations across $z$-axis | negative rotations across $z$-axis |

Figure S5: Samples from ToyBox showing rotations across different axes.

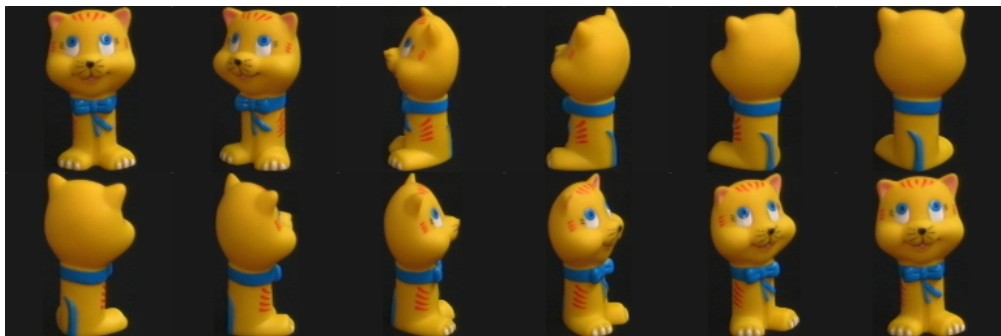

Figure S6: Samples from COIL100 showing different viewpoints.

### C.9 License of Dataset

In Table S6, we summarize the licensing terms for the datasets used in this study.

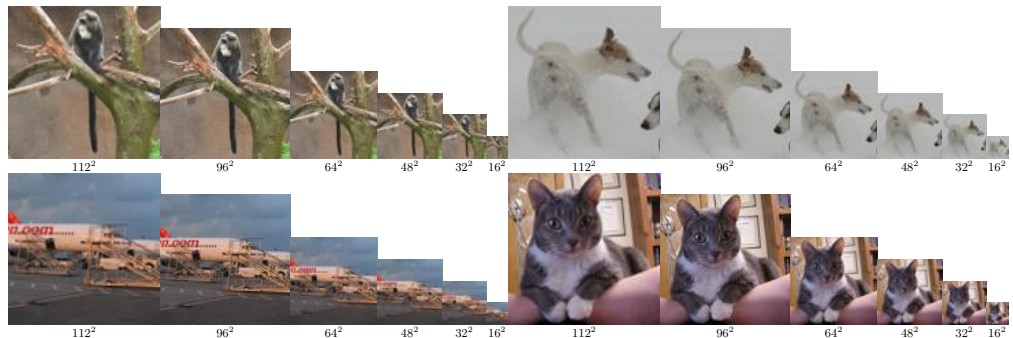

Figure S7: Samples from STL10 presenting low-resolution inputs.

Table S6: Licensing terms for the datasets used in this study.

| Dataset | License | Link |
|---------|---------|------|
| CharadesEgo | Non-commercial use | https://prior.allenai.org/projects/charades-ego |
| Moments-in-Time-v2 | Non-commercial use | http://moments.csail.mit.edu/ |
| Kinetics | CC BY 4.0 | https://www.deepmind.com/open-source/kinetics |
| HMDB51 | CC BY 4.0 | https://serre-lab.clps.brown.edu/resource/hmdb-a-large-human-motion-database/ |
| ToyBox | CC BY 4.0 | https://aivaslab.github.io/toybox/ |
| Mimetics | Open access | https://europe.naverlabs.com/research/computer-vision/mimetics/ |
| UCF101 | Open access | https://www.crcv.ucf.edu/data/UCF101.php |
| TinyVirat-v2 | Open access | https://www.crcv.ucf.edu/tiny-actions-challenge-cvpr2021/#tabtwo |
| COIL100 | Open access | https://www.cs.columbia.edu/CAVE/software/softlib/coil-100.php |
| STL-10 | Open access | https://cs.stanford.edu/~acoates/stl10/ |
| ActorShift | MIT | https://uvaauas.figshare.com/articles/dataset/ActorShift_zip/19387046 |
| Sims4Action | MIT | https://github.com/aroitberg/sims4action |
| RareAct | Apache | https://github.com/antoine77340/RareAct |

# D   Additional Experiments and Results

## D.1   Pretrained on Kinetics700

In addition to the Kinetics400 used in the main paper, we further utilize the Kinetics700 to evaluate the benefits of incorporating more diverse data for out-of-distribution generalization. We use a subset of Kinetics700 comprising 480K samples, which doubles the size of Kinetics400. We use this subset to pretrain the VSSL methods for the same number of total iterations as Kinetics400. Unless stated otherwise, we compare the performance between Kinetics400 and Kinetics700 pretraining in linear evaluation, averaging over 3 trials. In Table S7 and Figure S8, we present a high-level overview showing the impact of using more diverse data in pretraining. Table S7 highlights that non-contrastive methods ($v$-BYOL, $v$-SimSiam, $v$-DINO) benefit more from the inclusion of diverse data compared to the contrastive approaches ($v$-SimCLR, $v$-MoCo) and $v$-MAE. Interestingly, we observe a slight decrease in the performance of $v$-MAE in this setting. This could be attributed to several factors, such as the pretrained encoder ViT-B reaching saturation in the pretraining setup of $v$-MAE, resulting in no improvements in performance from the additional data. To address this, a larger backbone like ViT-L or ViT-H, as used in prior works [57, 1], could be explored. Unfortunately, resource limitations prevented us from adopting a larger network to test this. Detailed results are presented in Tables S8, S9, S10, S11 and S12.

Table S7: Summary of the models' behavior with more **diverse pretraining data** is presented. In terms of overall performance, we report the average accuracy across all VSSL methods and distribution shifts, encompassing a total of 66 experiments. Additionally, we present results from 11 experiments for each individual VSSL method. The results demonstrate that non-contrastive methods ($v$-BYOL, $v$-SimSiam, $v$-DINO) benefit more from the inclusion of diverse data compared to the contrastive approaches ($v$-SimCLR, $v$-MoCo) and $v$-MAE. Moreover, we observe a slight decrease in the performance of $v$-MAE  when pretrained with more diverse videos from Kinetics700.

| Methods | Pretraining | InD | OoD |
|---|---|---|---|
| Overall | Kinetics400 | 63.0 | 31.6 |
|  | Kinetics700 | 64.5 (↑ 1.5) | 32.5 (↑ 0.9) |
| $v$-SimCLR | Kinetics400 | 66.0 | 32.9 |
|  | Kinetics700 | 67.0 (↑ 1.0) | 33.0 (↑ 0.1) |
| $v$-MoCo | Kinetics400 | 67.1 | 33.1 |
|  | Kinetics700 | 67.7 (↑ 0.6) | 33.9 (↑ 0.8) |
| $v$-BYOL | Kinetics400 | 65.2 | 33.1 |
|  | Kinetics700 | 68.3 (↑ 3.1) | 35.0 (↑ 1.9) |
| $v$-SimSiam | Kinetics400 | 62.0 | 31.5 |
|  | Kinetics700 | 63.9 (↑ 1.9) | 32.4 (↑ 0.9) |
| $v$-DINO | Kinetics400 | 63.1 | 31.5 |
|  | Kinetics700 | 63.9 (↑ 0.8) | 34.1 (↑ 2.6) |
| $v$-MAE | Kinetics400 | 54.4 | 27.4 |
|  | Kinetics700 | 54.1 (↓ 0.3) | 26.6 (↓ 0.8) |

Table S8: Comparison of video models under **context** shift when pretrained with Kinetics400 (#240K) vs. Kinetics700 (#480K). We highlight the best result in each scenario in **bold**. Interestingly, we observe a consistent benefit from the inclusion of more diverse data in the InD validation sets of both splits. However, the improvements in OoD are not consistently observed.

| Method | Context (10 class) | | | | Context (50 class) | | | |
|---|---|---|---|---|---|---|---|---|
| | Kinetics400 | | Kinetics700 | | Kinetics400 | | Kinetics700 | |
| | OoD | InD | OoD | InD | OoD | InD | OoD | InD |
| $v$-SimCLR | 31.4 | **92.5** | 30.2 | 91.2 | 14.9 | 71.7 | 15.3 | 73.3 |
| $v$-MoCo | 29.7 | 92.2 | 27.7 | **93.1** | **15.0** | **74.0** | 14.1 | 75.0 |
| $v$-BYOL | 31.6 | 89.3 | 32.4 | 91.4 | 14.4 | 71.8 | **15.8** | **75.2** |
| $v$-SimSiam | 30.8 | 89.5 | 33.6 | 91.4 | 13.8 | 67.9 | 14.7 | 70.3 |
| $v$-DINO | **34.8** | 90.4 | **34.3** | 91.0 | 13.0 | 68.7 | 14.5 | 72.3 |
| $v$-MAE | 33.3 | 82.9 | 31.9 | 83.4 | 12.3 | 56.4 | 12.7 | 54.5 |

Table S9: Comparison of video models under **viewpoint** and **actor** shifts, when pretrained with Kinetics400 (#240K) vs. Kinetics700 (#480K). We highlight the best results in each scenario in **bold**. The results demonstrate that adding diverse pretraining data significantly improves the performance under egocentric viewpoint shifts and top-down synthetic domain shifts. However, less significant improvements are observed for surveillance viewpoint shifts and animal domain actor shifts. Moreover, the performance of $v$-MAE deteriorates when pretrained with Kinetics700.

| Method | View (ego.) | | | | View (surv.+low res) | | | | View+Act (t-d+syn.) | | | | Actor (animal) | | | |
|---|---|---|---|---|---|---|---|---|---|---|---|---|---|---|---|---|
| | K400 | | K700 | | K400 | | K700 | | K400 | | K700 | | K400 | | K700 | |
| | OoD | InD | OoD | InD | OoD | InD | OoD | InD | OoD | InD | OoD | InD | OoD | InD | OoD | InD |
| $v$-SimCLR | 12.7 | 14.7 | 13.7 | 15.7 | **26.1** | 39.1 | 24.9 | 40.2 | **42.4** | 67.8 | 43.1 | 71.2 | 67.9 | 91.7 | 68.7 | 92.4 |
| $v$-MoCo | **13.3** | 15.0 | **14.1** | 16.1 | 24.8 | **40.0** | **25.9** | 40.5 | 41.1 | **67.9** | **49.4** | 69.7 | 68.1 | **92.2** | 67.1 | 92.5 |
| $v$-BYOL | 12.0 | 14.4 | 13.9 | **16.1** | 22.7 | 37.8 | 22.2 | **41.7** | 37.3 | 65.6 | 45.5 | **71.4** | **68.3** | 91.5 | **69.1** | **93.3** |
| $v$-SimSiam | 11.6 | 14.1 | 12.8 | 14.9 | 23.3 | 34.3 | 21.4 | 37.3 | 40.0 | 65.5 | 37.9 | 67.0 | 68.1 | 91.1 | 67.7 | 91.5 |
| $v$-DINO | 12.0 | 14.4 | 13.4 | 15.7 | 22.3 | 35.3 | 22.9 | 38.0 | 35.3 | 62.9 | 42.0 | 68.5 | 66.7 | 90.7 | 68.3 | 91.2 |
| $v$-MAE | 10.9 | 13.7 | 10.9 | 13.8 | 23.5 | 32.0 | 23.1 | 31.2 | 37.8 | 58.0 | 33.5 | 57.6 | 59.8 | 85.9 | 56.2 | 86.3 |

Table S10: Comparison of video models under **source** shift, when pretrained with Kinetics400 (#240K) vs. Kinetics700 (#480K). We highlight the best results in each scenario in **bold**. $v$-DINO and $v$-SimSiam benefit the most from the inclusion of diverse pretraining data, while the other methods show the same or worse performance.

| Method | UCF/HMDB | | | | HMDB/UCF | | | |
|---|---|---|---|---|---|---|---|---|
| | Kinetics400 | | Kinetics700 | | Kinetics400 | | Kinetics700 | |
| | OoD(H) | InD(U) | OoD(H) | InD(U) | OoD(U) | InD(H) | OoD(U) | InD(H) |
| $v$-SimCLR | 47.7 | 96.0 | 45.1 | 96.5 | 55.1 | 82.0 | 55.1 | **83.6** |
| $v$-MoCo | **51.5** | **97.1** | 48.3 | 96.3 | 57.2 | **84.5** | 54.6 | 82.6 |
| $v$-BYOL | 51.4 | 94.5 | **50.5** | **96.8** | **63.1** | 79.9 | **61.1** | 82.8 |
| $v$-SimSiam | 46.1 | 92.6 | 48.0 | 92.5 | 52.2 | 76.2 | 53.1 | 80.5 |
| $v$-DINO | 49.3 | 94.3 | 50.3 | 95.8 | 53.8 | 77.5 | 58.2 | 81.8 |
| $v$-MAE | 39.5 | 89.2 | 39.5 | 89.3 | 39.1 | 72.8 | 37.5 | 72.4 |

Table S11: Comparison of video models in **zero-shot** recognition, when pretrained with Kinetics400 (#240K) vs. Kinetics700 (#480K). We highlight the best results in each scenario in **bold**. The results demonstrate that the inclusion of diverse data generally improves the zero-shot performance. We observe significant improvements in most cases, with the exception of $v$-MAE on UCF and $v$-BYOL on HMDB. Moreover, we also notice significant improvements in InD performance, particularly in the case of non-contrastive methods and $v$-MAE.

| Method | UCF | | HMDB | | RareAct | | Kinetics400 (InD) | |
|---|---|---|---|---|---|---|---|---|
| | K400 | K700 | K400 | K700 | K400 | K700 | K400 | K700 |
| $v$-SimCLR | **37.2** | 39.0 | 18.6 | 19.0 | 7.7 | 9.4 | 56.8 | 57.8 |
| $v$-MoCo | 35.2 | 40.6 | 19.5 | **21.6** | **8.7** | 9.4 | **58.4** | 59.5 |
| $v$-BYOL | 33.0 | **44.0** | **22.4** | 21.1 | 7.5 | **9.9** | 57.4 | **61.0** |
| $v$-SimSiam | 34.0 | 39.2 | 18.8 | 20.3 | 7.7 | 8.3 | 50.4 | 52.3 |
| $v$-DINO | 34.3 | 41.1 | 17.2 | 20.0 | 8.1 | 9.5 | 53.4 | 58.3 |
| $v$-MAE | 25.5 | 24.1 | 14.2 | 17.2 | 5.8 | 6.2 | 35.9 | 35.4 |

Table S12: Comparison of video models in **open-set** recognition, when pretrained with Kinetics400 (#240K) vs. Kinetics700 (#480K). We highlight the best results in each scenario in **bold**. The results exhibit a very similar trend in both pretraining setups. For example, while $v$-SimCLR and $v$-MoCo perform fairly well in closed-set, they show near chance-level performance in open-set. $v$-MAE performs poorly in both open-set and closed-set, in both pretraining setups. $v$-DINO achieves the best open-set performance in both setups while retaining decent closed-set performance.

| Method | Pretrained on Kinetics400 | | | Pretrained on Kinetics700 | | |
|---|---|---|---|---|---|---|
| | Open-set (AUC) U/H | Closed-set w/ DEAR (Acc.) U101* | Closed-set w/ CE (Acc.) U101* | Open-set (AUC) U/H | Closed-set w/ DEAR (Acc.) U101* | Closed-set w/ CE (Acc.) U101* |
| $v$-SimCLR | 49.9 | 11.6 | 84.1 | 49.7 | 18.5 | 83.6 |
| $v$-MoCo | 50.7 | 32.7 | 84.9 | 53.1 | 53.6 | 84.7 |
| $v$-BYOL | 56.0 | 63.7 | **85.4** | 74.5 | **85.5** | **86.7** |
| $v$-SimSiam | 73.7 | 79.5 | 82.5 | 69.6 | 80.5 | 82.3 |
| $v$-DINO | **79.4** | **80.4** | 80.9 | **82.0** | 84.5 | 85.0 |
| $v$-MAE | 66.0 | 74.6 | 76.2 | 65.6 | 74.2 | 75.1 |

Table S13: Comparison of VSSL methods in **zero-shot** recognition, when pretrained with Kinetics700 (#480K). We highlight the best results in each scenario in **bold**. In addition to Table S11, we also finetune video-text encoders with Kinetics700 and perform zero-shot recognition using the full UCF101 and HMDB51, along with RareAct. Please note that during both pretraining and finetuning, we discard the videos from Kinetics700 that belong to the overlapping classes from UCF101 and HMDB51. $v$-MAE clearly shows superiority when validated on UCF101 and HMDB51, while $v$-MoCo outperforms others when validated on RareAct.

| Method | UCF | HMDB | RareAct |
|---|---|---|---|
| $v$-SimCLR | 45.7 | 34.0 | 15.3 |
| $v$-MoCo | 48.4 | 33.9 | **18.4** |
| $v$-BYOL | 47.3 | 33.4 | 15.5 |
| $v$-SimSiam | 37.9 | 29.1 | 13.3 |
| $v$-DINO | 46.6 | 32.9 | 16.2 |
| $v$-MAE | **50.1** | **37.3** | 16.0 |

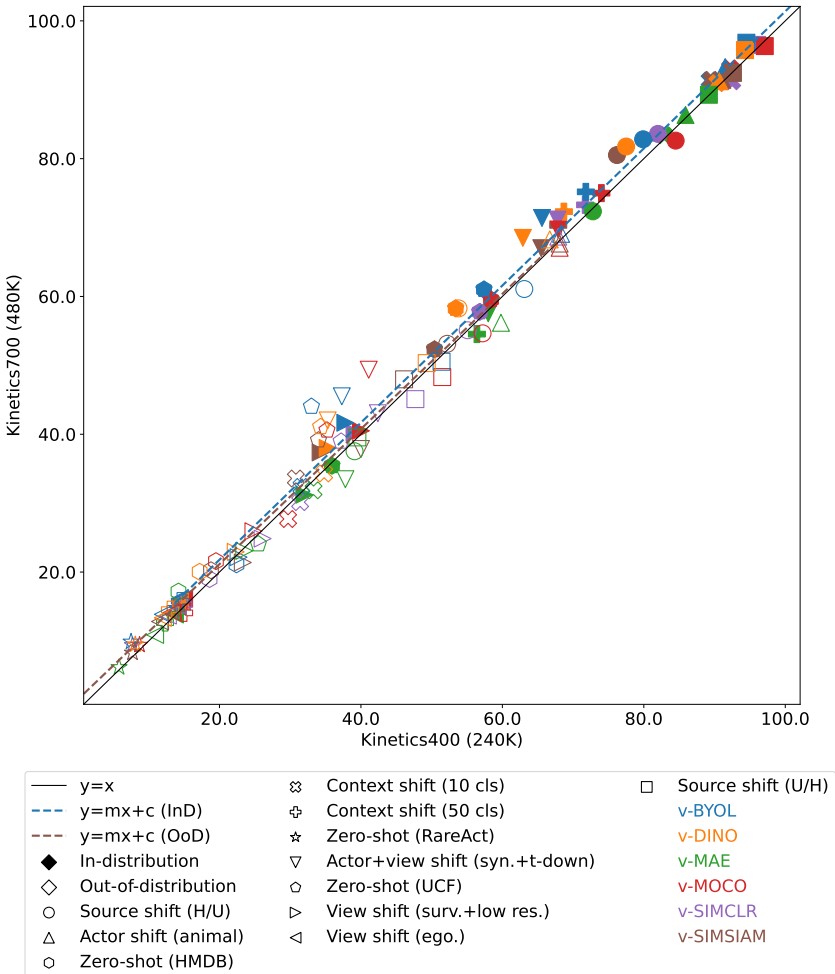

Figure S8: We present a high-level depiction of the models' performance when **pretrained with Kinetics400 vs. Kinetics700**. When pretrained with Kinetics700, the trend line of InD performance (blue dashed line) is slightly higher than that of the OoD performance (brown dashed line).

## D.2 Temporal dynamics

In this subsection, we provide the additional results of our experiment on the temporal dynamics of VSSL methods. We present the performance on temporal transformation recognition in Figure S9. Moreover, we present the performance of VSSL methods in classifying the same objects from static videos, in Figure S10.

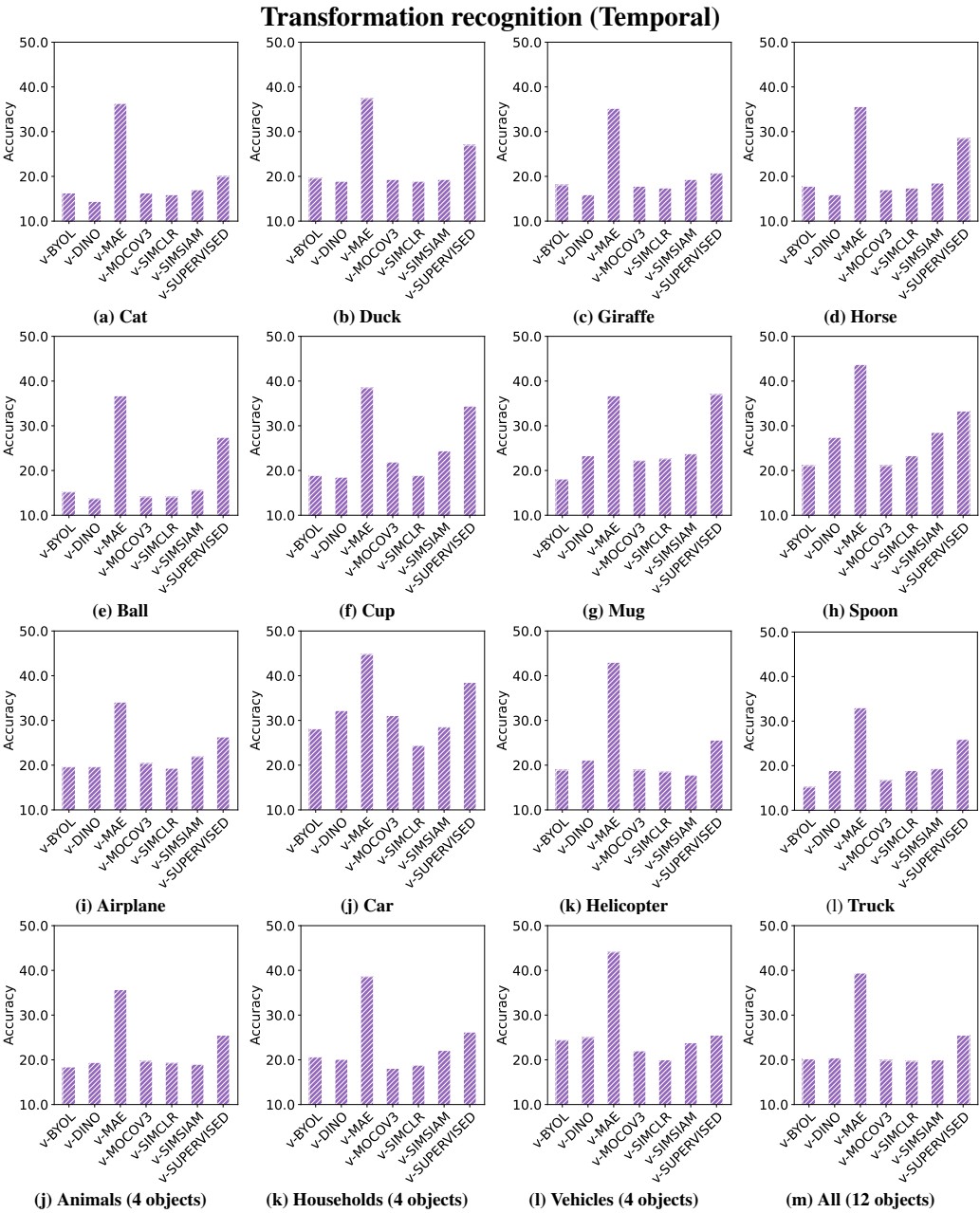

Figure S9: Evaluating performance of video models on egocentric **transformation recognition** using ToyBox [70]. The models are used to classify a total of 9 types of temporal transformations including positive rotations, negative rotations, and transformations across the $x$, $y$, and $z$ axes. In all of the cases, $v$-MAE consistently shows superior performance.

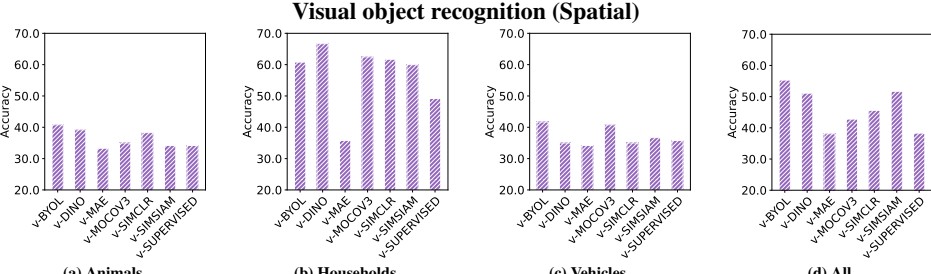

**Visual object recognition (Spatial)**

(a) Animals  (b) Households  (c) Vehicles  (d) All

Figure S10: Evaluating performance of video models in egocentric **object recognition**. The models are used to classify the videos of static objects including 4 types of animals, household items, cars, and all of them together (a total of 12 classes). In all cases $v$-MAE consistently exhibits the lowest performance. Among all the categories, the models perform significantly better in classifying household objects than animals or vehicles. This is likely because the pretraining data Kinetics400 consist of several human actions involving household objects e.g., kitchen utensils, etc.

### D.3 Comparing InD vs. OoD performance under natural distribution shifts

To gain a high-level understanding of the models' performance in InD vs. OoD, we visualize their accuracy in a 2D space in Figure S11. It is noteworthy that superior InD performance does not necessarily translate to better OoD performance. We observe cases where models exhibit similar InD performance but significantly differ in their performance under distribution shifts. Ablation variants of Figure S11 are presented in Figure S12. Statistical analyses on the robustness of video models are presented in Appendix D.7.

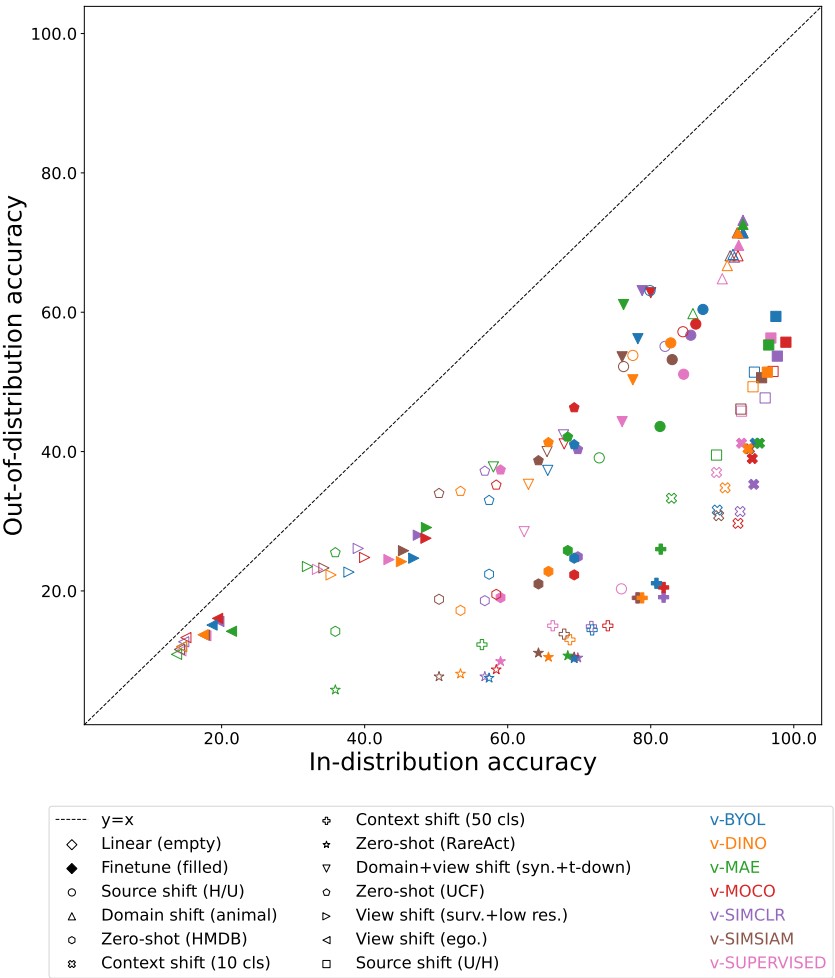

Figure S11: We compare the **InD vs. OoD** performance of video models under different distribution shifts. The filled markers indicate the finetuned results, while the empty markers represent the results from linear evaluations. In all cases, we observe a decrease in performance in the OoD validation set, as indicated by the data points below the $y = x$ line.

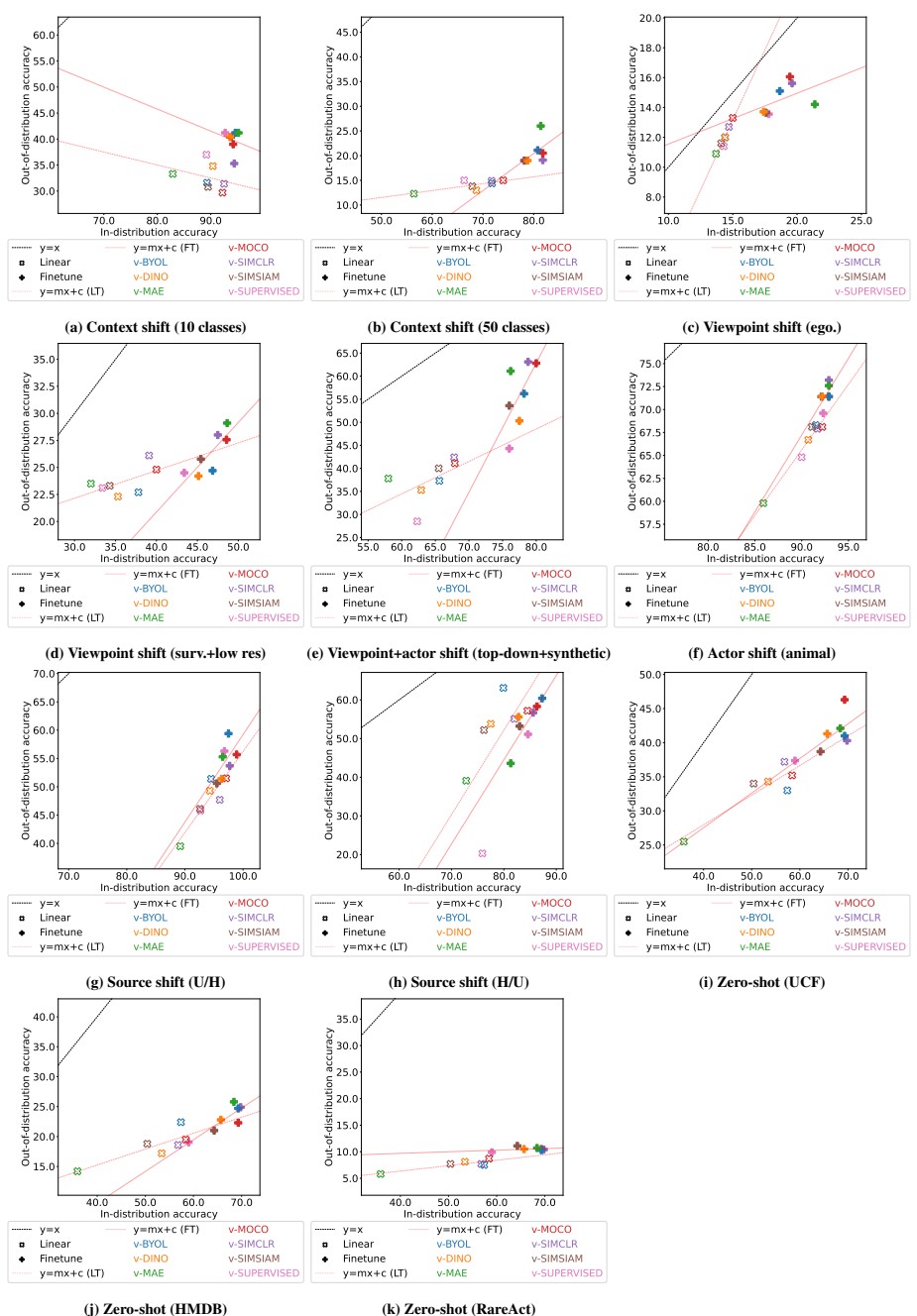

Figure S12: An ablated view of **InD vs. OoD** performance under natural distribution shifts, utilizing both frozen (empty markers) and finetuned (filled markers) encoders. Here, $y = mx + c$ shows a linear fit for the given data points (i.e., projected InD and OoD performance metrics). The data points above linear fit indicate more robustness. (**a and b**) $v$-Supervised demonstrates superior performance in linear evaluation, while $v$-MAE achieves the best results when finetuned. On the other hand, although $v$-MoCo and $v$-SimCLR show strong performance in InD validation, they exhibit weaker generalization in out-of-context scenarios. (**c, d, and e**) Overall, $v$-SimCLR and $v$-MoCo perform better in viewpoint shifts in both linear and finetuning. $v$-MAE (finetuned) shows a single instance of superior results for the specific case of low-resolution surveillance camera shift due to its robustness in low-resolution inputs. (**f**) In animal domain actor shift, $v$-BYOL achieves the best results in linear evaluation, whereas, $v$-SimCLR outperforms others when finetuned. (**g and h**) $v$-BYOL achieves the best performance under source shifts in all setups. (**i and j**) Frozen $v$-BYOL achieves the best zero-shot recognition on UCF but performs poorly on HMDB. On the other hand, the frozen $v$-SimCLR performs the best on HMDB, while generalizing poorly on UCF. (**k**) Overall, video models generalize poorly in zero-shot recognition of unusual actions (RareAct).

## D.4 Effect of synthetic perturbations

In addition to the natural distribution shifts, we further extend our work investigating the performance of VSSL methods under synthetic perturbations. We follow the setup proposed in [53] to create the synthetic perturbation of varying severity on a scale of 1 to 5. We apply a total of 16 different synthetic perturbations belonging to the common spatio-temporal augmentation techniques such as noise addition, blurring, temporal perturbation, and camera motion. We test the robustness of the frozen encoders against synthetic perturbations on 2 benchmarks UCF101 and Kinetics400, presented in Figures S13 and S14 respectively. The results demonstrate that with increasing perturbation severity, the performance of video models deteriorates. However, it is important to acknowledge that certain visual augmentations, such as noise and blur, are already applied during the self-supervised pretraining phase. As a result, they may not be considered as distribution shifts within the scope of our setup. Furthermore, it is neither possible to train self-supervised methods without such augmentation techniques. Despite these considerations, we present these results for the sake of completeness.

Since spatial perturbations are extensively covered in literature, we refrain from explaining them here and redirect readers to [53, 54, 49]. Instead, we briefly discuss the temporal perturbations [53]. 'Sampling': using different frame rates than used in training; 'Reverse sampling': reversing the temporal order with different sampling rates; 'Jumble': shuffling frames in segments; 'Box jumble': shuffling the segments instead of frames; 'Freezing': randomly freezing video frames; 'Random shuffle': randomly shuffle the frame orders, please note random shuffle has just one severity level.

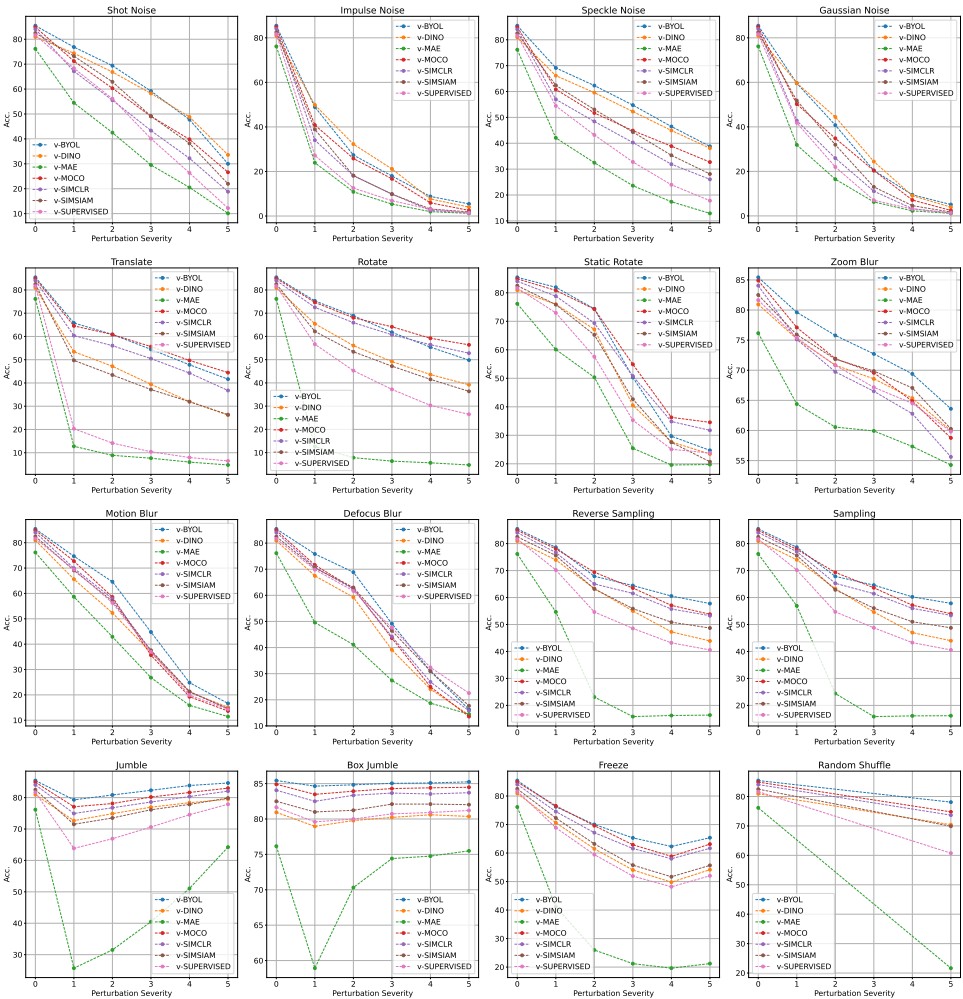

Figure S13: Spatio-temporal perturbations on UCF101 (linear evaluation). When subjected to extreme perturbations, the performance of VSSL methods experiences a significant decline, reaching near-chance levels. Notably, $v$-MAE exhibits the worst performance across all setups. Among all methods, $v$-BYOL demonstrates superiority in scenarios involving blur and temporal perturbations, while $v$-MoCo outperforms others in rotations and translations. Furthermore, $v$-DINO excels when tested on noisy videos.

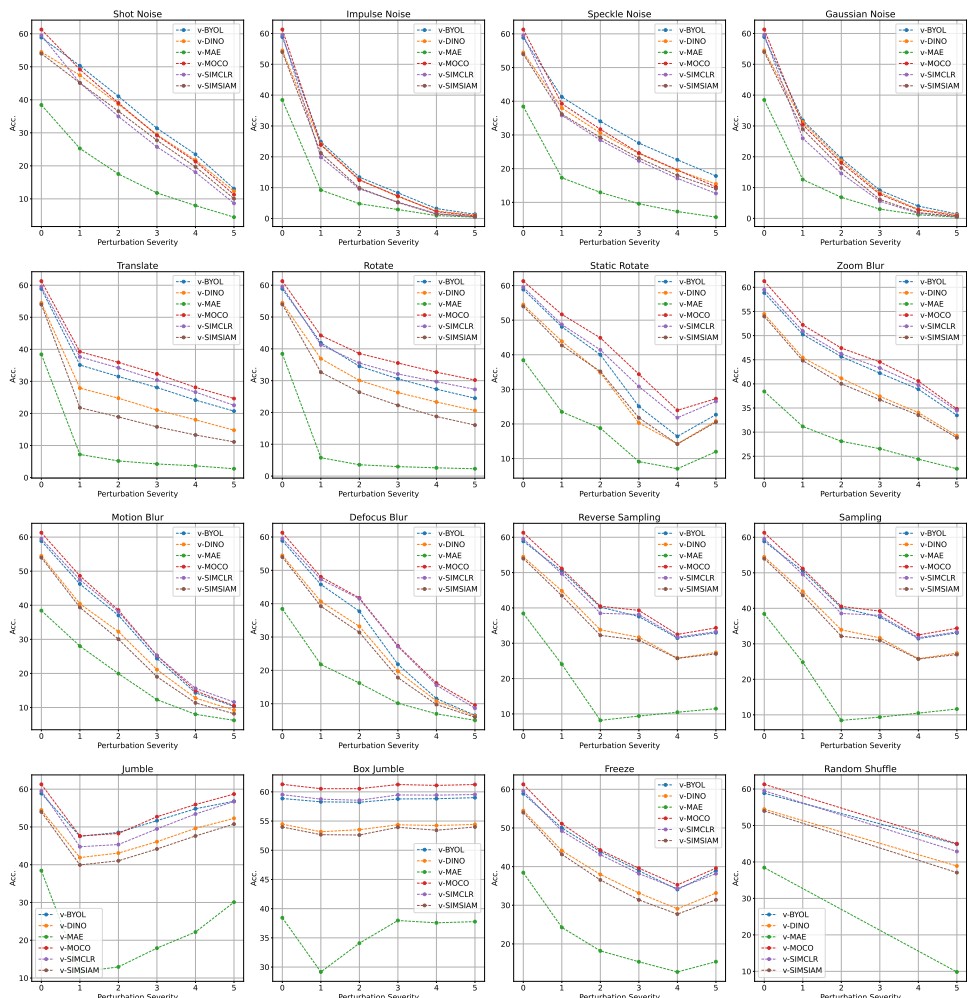

Figure S14: Spatio-temporal perturbations on Kinetics400 (linear evaluation). When subjected to extreme perturbations, the performance of VSSL methods experiences a significant decline, reaching near-chance levels. Notably, $v$-MAE exhibits the worst performance across all setups. Among all methods, $v$-MoCo demonstrates superiority in scenarios involving blur, rotations, and temporal perturbations, while $v$-BYOL outperforming the others when tested on noisy videos.

## D.5 Effect of finetuning in out-of-distribution generalization

### D.5.1 Linear vs. finetuning performance comparison

We conduct a thorough study evaluating the impact of finetuning across different VSSL methods. We present a high-level overview in Table S14 and more detailed results in Figures S15, S16, and S17.

Table S14: A high-level summary of the impact of the models' performance when finetuned, both InD and OoD. Overall, finetuning improves performance in both InD and OoD. However, the performance varies based on VSSL methods. Generally, the improvements in InD performance tend to be more significant than those in OoD. We note that $v$-MAE benefits the most from finetuning due to its extremely poor generalizability when using frozen encoders.

| Methods | Eval. setup | InD | OoD |
|---|---|---|---|
| Overall | Linear | 63.2 | 31.5 |
| | Finetune | 71.9 (↑ 8.7) | 37.4 (↑ 5.9) |
| $v$-Supervised | Linear | 65.5 | 30.7 |
| | Finetune | 69.0 (↑ 3.5) | 35.1 (↑ 4.4) |
| $v$-SimCLR | Linear | 66.0 | 32.9 |
| | Finetune | 73.4 (↑ 7.4) | 38.2 (↑ 5.9) |
| $v$-MoCo | Linear | 67.1 | 33.1 |
| | Finetune | 73.6 (↑ 6.5) | 39.1 (↑ 6.0) |
| $v$-BYOL | Linear | 65.2 | 33.1 |
| | Finetune | 73.2 (↑ 8.0) | 38.7 (↑ 4.6) |
| $v$-SimSiam | Linear | 62.0 | 31.5 |
| | Finetune | 70.4 (↑ 7.6) | 36.2 (↑ 4.7) |
| $v$-DINO | Linear | 63.1 | 31.5 |
| | Finetune | 71.0 (↑ 7.9) | 36.4 (↑ 4.9) |
| $v$-MAE | Linear | 54.4 | 27.4 |
| | Finetune | 72.6 (↑ 7.2) | 38.3 (↑ 10.9) |

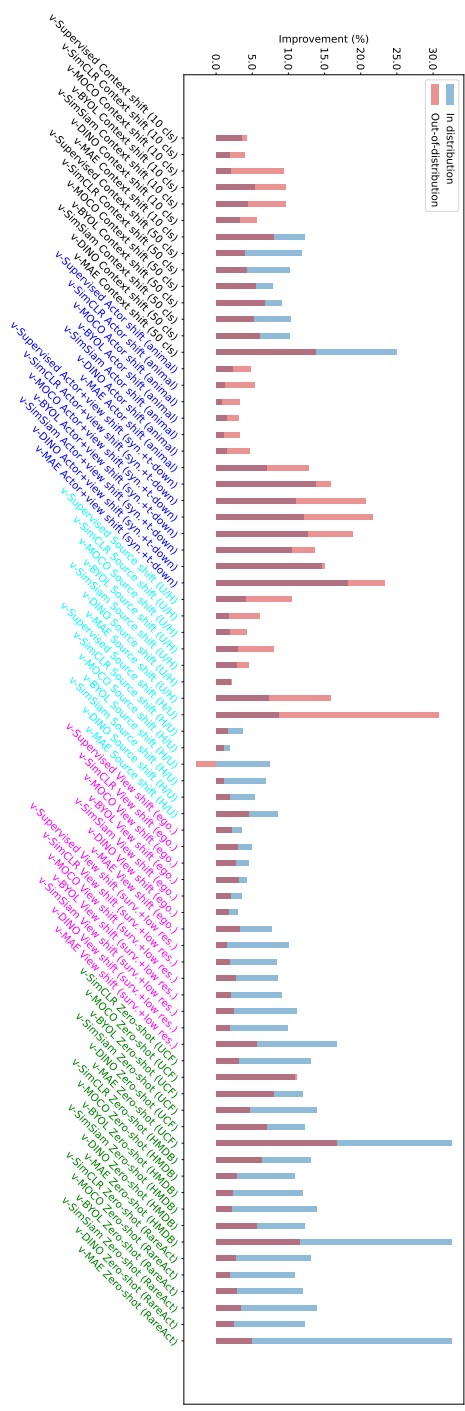

Figure S15: A detailed comparison of the impact of **finetuning on InD vs. OoD**. For optimal viewing, please rotate 90 degrees to the left ↺. The results demonstrate that the benefits of finetuning are highly dependent on the VSSL method and type of distribution shift. Our analysis reveals that finetuning tends to be more advantageous in scenarios involving actor shifts (animal domain and synthetic domain). Conversely, its benefits are relatively less pronounced under viewpoint shifts and zero-shot recognition. Moreover, the impact of finetuning in source shift and context shift is mixed. For example, while we notice significant benefits in UCF to HMDB shift, finetuning in fact worsens the performance in the case of HMDB to UCF shift. Additionally, finetuning is more beneficial when evaluated on a smaller benchmark with just 10 out-of-context classes. However, the improvements are relatively modest in a more challenging setup involving 50 out-of-context classes.

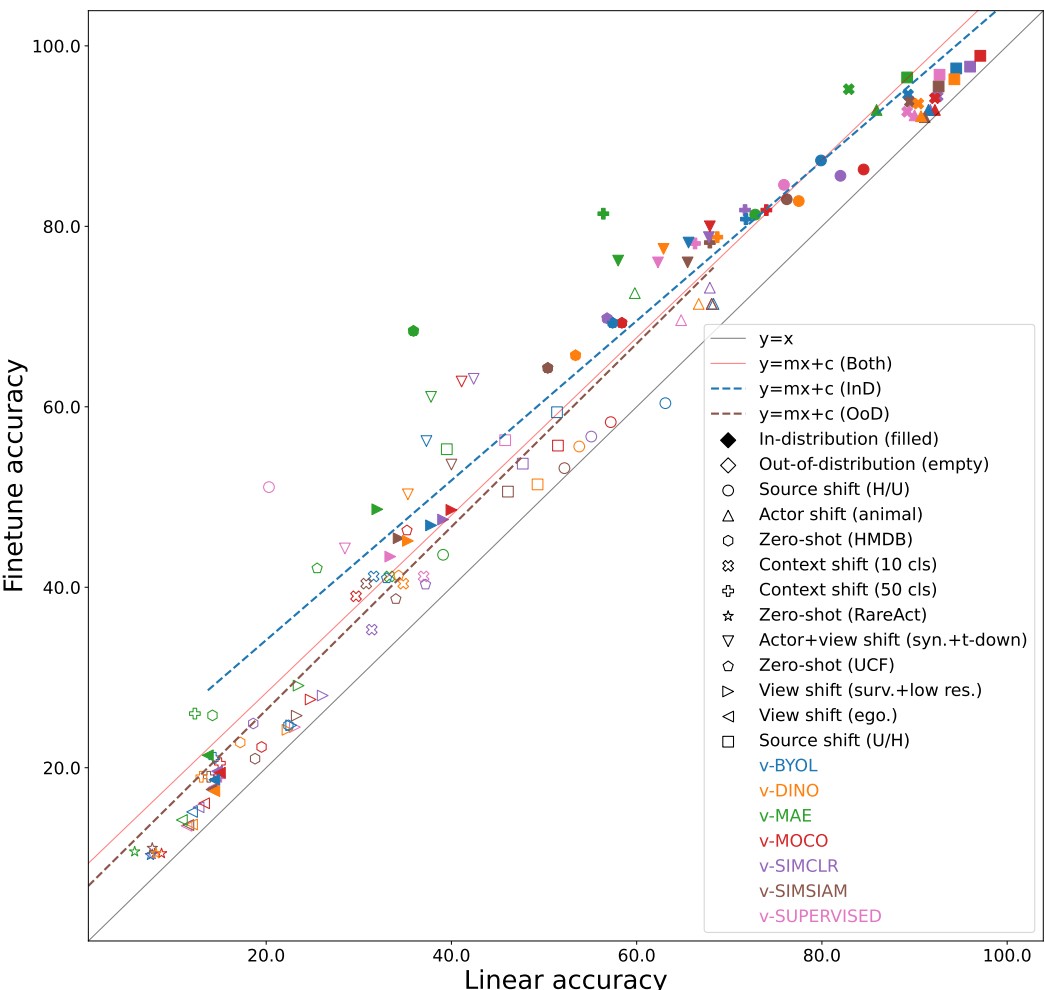

Figure S16: Comparing the performance of video models in **linear evaluation vs. finetuning**. In the plot, the filled markers represent InD results, while the empty markers represent the OoD results. Overall, we observe that finetuning consistently leads to improved performance, as indicated by the data points lying above the diagonal line ($y = x$), with only one exception. It is worth noting that the average improvement in InD performance (shown by the blue dashed line) is higher compared to the improvement in OoD performance (shown by the brown dashed line).

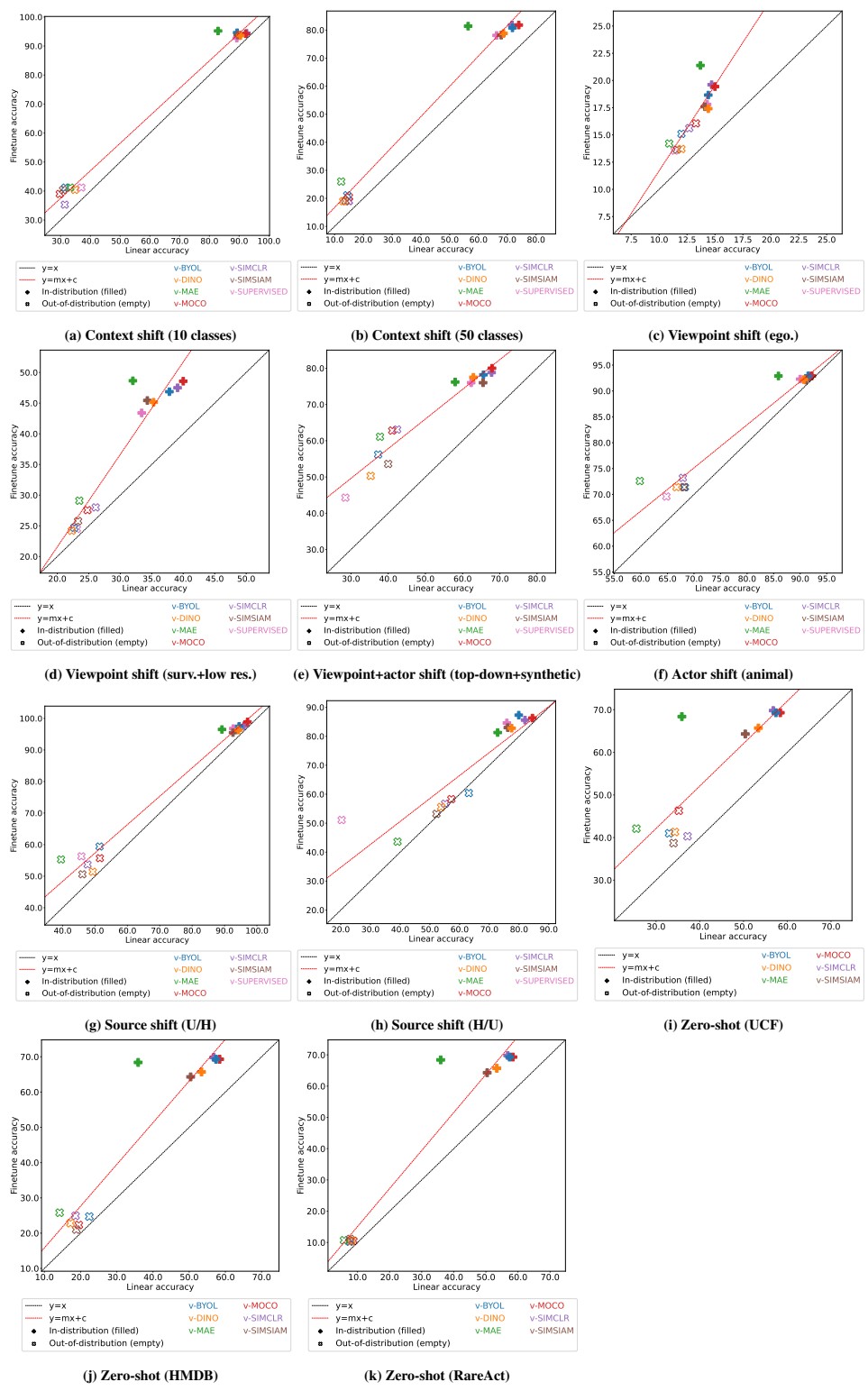

Figure S17: An ablated view of **linear vs. finetuning** performance comparison. Our results demonstrate that improved InD performance does not necessarily translate to better OoD performance. We observe cases where models exhibit similar InD performance but significantly differ in their performance under distribution shifts, e.g., please see the linear results in d, e, f, and i.

### D.5.2 In-distribution overfitting due to finetuning

As mentioned in the main paper, while finetuning enhances overall performance, it is susceptible to in-distribution overfitting. In other words, while additional training leads to improved performance on InD validation, it adversely affects OoD performance. We present a few examples of such instances in Figure S18. In practice, we apply early stopping to tackle such overfitting.

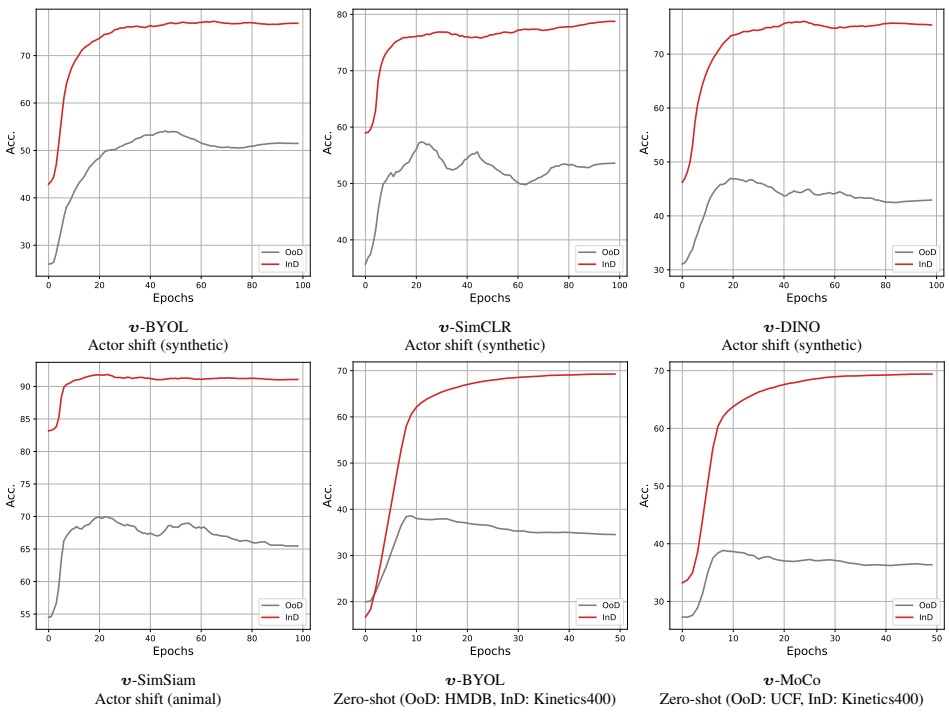

Figure S18: A few examples showing **in-distribution overfitting** caused during finetuning.

## D.6 High-level overview

The high-level overview presented in Figure 1 summarizes our findings from Tables 1, 2, 3, 4, and 6(a) of the main paper. To create the overview plot, we first normalize the OoD performance (e.g., accuracy, mAP) of each experiment to that of the best-performing method. Thus, the best performing method gets 1.0 and all other methods get a score in $[0, 1]$. Next, we group the experiments based on their super-categories and then take their average score. For example, in the case of source shift, we compute this average by grouping the normalized scores of 4 experiments, namely linear and finetuning evaluation of both UCF/HMDB and HMDB/UCF shifts. We follow similar steps for all the shifts (details are given below), resulting in a $7 \times 6$ matrix, where 7 represents the number of video methods, and 6 represents the total number of shifts. This final matrix is then used to color-code and present a high-level summary of all methods across all shifts.

- Context shift: Table 1
    - Context shift (10 classes) linear; best performing method: $v$-Supervised
    - Context shift (10 classes) finetune; best performing methods: $v$-Supervised, $v$-BYOL, $v$-MAE
    - Context shift (50 classes) linear; best performing methods: $v$-Supervised, $v$-MoCo
    - Context shift (50 classes) finetune; best performing method: $v$-MAE
- Viewpoint shift: Table 2
    - Viewpoint shift (egocentric) linear; best performing method: $v$-MoCo
    - Viewpoint shift (egocentric) finetune; best performing method: $v$-MoCo
    - Viewpoint shift (surveillance+low resolution) linear; best performing method: $v$-SimCLR
    - Viewpoint shift (surveillance+low resolution) finetune; best performing method: $v$-MAE
    - Viewpoint+actor shift (top-down+synthetic) linear; best performing method: $v$-SimCLR
    - Viewpoint+actor shift (top-down+synthetic) finetune; best performing method: $v$-SimCLR
- Actor shift: Table 2
    - Viewpoint+actor shift (top-down+synthetic) linear; best performing method: $v$-SimCLR
    - Viewpoint+actor shift (top-down+synthetic) finetune; best performing method: $v$-SimCLR
    - Actor shift (animal) linear; best performing method: $v$-BYOL
    - Actor shift (animal) finetune; best performing method: $v$-SimCLR
- Source shift: Table 3
    - Source shift (UCF/HMDB) linear; best performing method: $v$-MoCo
    - Source shift (UCF/HMDB) finetune; best performing method: $v$-BYOL
    - Source shift (HMDB/UCF) linear; best performing method: $v$-BYOL
    - Source shift (HMDB/UCF) finetune; best performing method: $v$-BYOL
- Zero-shot recognition: Table 4
    - Zero-shot (UCF) linear; best performing method: $v$-SimCLR
    - Zero-shot (UCF) finetune; best performing method: $v$-MoCo
    - Zero-shot (HMDB) linear; best performing method: $v$-BYOL
    - Zero-shot (HMDB) finetune; best performing method: $v$-MAE
    - Zero-shot (RareAct) linear; best performing method: $v$-MoCo
    - Zero-shot (RareAct) finetune; best performing method: $v$-SimSiam
- Open-set recognition: Table 6(a)
    - Open-set (K400/UCF) finetune; best performing method: $v$-SimCLR
    - Open-set (K400/HMDB) finetune; best performing method: $v$-MoCo
    - Open-set (U101/HHDB) finetune; best performing method: $v$-MoCo

## D.7 Statistical analysis

We present detailed statistical analyses to verify the statistical significance of all the major findings/-claims discussed in the main paper. In particular, we compare the robustness of the video models under different distribution shifts, after compensating their InD performance. The adjusted OoD performance difference between the two methods is calculated as:

$$\text{Adjusted } \Delta_{\text{OoD}} = \Delta_{\text{OoD}} - m \times \Delta_{\text{InD}} - 1.96 \times \text{SE}_{\text{OoD}}$$

Here, $m \in [0, 1]$ is the slope of a linear fit of projected InD and OoD performance measures of all the models for a particular shift, calculated as:

$$m = \frac{\sum_{i=1}^{n}(x_i - \bar{x})(y_i - \bar{y})}{\sum_{i=1}^{n}(x_i - \bar{x})^2},$$

where $x_i$ and $y_i$ refer to InD and OoD performance measures for $i^{\text{th}}$ method and $i \in [1, n]$, and $n$ is the total number of methods studied. Moreover, $\bar{x}$ and $\bar{y}$ refer to the mean InD and OoD performances, respectively. Between Method 1 and Method 2, we calculate $\Delta_{\text{OoD}} = y_1 - y_2$ and $\Delta_{\text{InD}} = x_1 - x_2$, while $\text{SE}_{\text{OoD}}$ is the standard error of Method 1's OoD performance across several trials. We consider the standard error with statistical significance at 95% confidence. Finally, **we accept Method 1 is more robust than Method 2, if Adjusted $\Delta_{\text{OoD}} > 0$**. The results of our statistical analyses are presented in Tables S15 through S26.

Table S15: Comparative statistical analysis of the robustness of $v$-MAE and $v$-Supervised under **context shift** (original numbers from Table 1).

| Distribution shift | Method 1 | Method 2 | Adjusted $\Delta_{\text{OOD}}$ | |
| --- | --- | --- | --- | --- |
| | | | Lin. | FT. |
| | $v$-Supervised | $v$-SimCLR | 5.24 | 5.90 |
| | $v$-Supervised | $v$-MoCo | 6.96 | 2.20 |
| | $v$-Supervised | $v$-BYOL | 5.00 | 0.00 |
| | $v$-Supervised | $v$-SimSiam | 5.73 | 0.80 |
| Context (10 class) | $v$-Supervised | $v$-DINO | 1.81 | 0.80 |
| | $v$-MAE | $v$-SimCLR | 0.25 | 5.90 |
| | $v$-MAE | $v$-MoCo | 1.97 | 2.20 |
| | $v$-MAE | $v$-BYOL | 0.01 | 0.00 |
| | $v$-MAE | $v$-SimSiam | 0.74 | 0.80 |
| | $v$-MAE | $v$-DINO | -3.17 | 0.80 |
| | $v$-Supervised | $v$-SimCLR | 0.44 | 2.68 |
| | $v$-Supervised | $v$-MoCo | 0.66 | 1.28 |
| | $v$-Supervised | $v$-BYOL | 0.92 | -0.07 |
| | $v$-Supervised | $v$-SimSiam | 0.99 | 0.08 |
| Context (50 class) | $v$-Supervised | $v$-DINO | 1.88 | 0.53 |
| | $v$-MAE | $v$-SimCLR | -0.82 | 7.20 |
| | $v$-MAE | $v$-MoCo | -0.59 | 5.80 |
| | $v$-MAE | $v$-BYOL | -0.34 | 4.45 |
| | $v$-MAE | $v$-SimSiam | -0.27 | 4.59 |
| | $v$-MAE | $v$-DINO | 0.63 | 5.04 |

Table S16: Comparative statistical analysis of the robustness of $v$-MAE and $v$-Supervised in learning **temporal dynamics** (original numbers from Figure 3(a)).

| Distribution shift | Method 1 | Method 2 | Adjusted $\Delta_{\text{OOD}}$ |
|---|---|---|---|
| Temporal learner | $v$-Supervised | $v$-BYOL | 0.06 |
| | $v$-Supervised | $v$-SimSiam | 0.05 |
| | $v$-Supervised | $v$-DINO | 0.05 |
| | $v$-Supervised | $v$-SimCLR | 0.06 |
| | $v$-Supervised | $v$-MoCo | 0.06 |
| | $v$-MAE | $v$-BYOL | 0.19 |
| | $v$-MAE | $v$-SimSiam | 0.19 |
| | $v$-MAE | $v$-DINO | 0.19 |
| | $v$-MAE | $v$-SimCLR | 0.20 |
| | $v$-MAE | $v$-MoCo | 0.20 |

Table S17: Comparative statistical analysis of the robustness of contrastive methods ($v$-SimCLR and $v$-MoCo) under **viewpoint shift** (original numbers from Table 2).

| Distribution shift | Method 1 | Method 2 | Adjusted $\Delta_{\text{OOD}}$ | |
|---|---|---|---|---|
| | | | Lin. | FT. |
| Viewpoint (ego.) | $v$-SimCLR | $v$-BYOL | 0.21 | 0.20 |
| | $v$-SimCLR | $v$-SimSiam | 0.35 | 1.28 |
| | $v$-SimCLR | $v$-DINO | 0.24 | 1.16 |
| | $v$-SimCLR | $v$-Supervised | 0.71 | 1.44 |
| | $v$-SimCLR | $v$-MAE | 0.65 | 2.01 |
| | $v$-MoCo | $v$-BYOL | 0.56 | 0.69 |
| | $v$-MoCo | $v$-SimSiam | 0.71 | 1.78 |
| | $v$-MoCo | $v$-DINO | 0.60 | 1.66 |
| | $v$-MoCo | $v$-Supervised | 1.06 | 1.94 |
| | $v$-MoCo | $v$-MAE | 1.00 | 2.51 |
| Viewpoint (surv.+low res) | $v$-SimCLR | $v$-BYOL | 2.50 | 2.77 |
| | $v$-SimCLR | $v$-SimSiam | 0.98 | 0.51 |
| | $v$-SimCLR | $v$-DINO | 2.22 | 1.84 |
| | $v$-SimCLR | $v$-Supervised | 0.89 | 0.10 |
| | $v$-SimCLR | $v$-MAE | 0.21 | -0.16 |
| | $v$-MoCo | $v$-BYOL | 0.49 | 1.44 |
| | $v$-MoCo | $v$-SimSiam | -1.03 | -0.81 |
| | $v$-MoCo | $v$-DINO | 0.22 | 0.52 |
| | $v$-MoCo | $v$-Supervised | -1.12 | -1.22 |
| | $v$-MoCo | $v$-MAE | -1.79 | -1.48 |
| View+Act (t-down+syn.) | $v$-SimCLR | $v$-BYOL | 1.87 | 6.30 |
| | $v$-SimCLR | $v$-SimSiam | -0.88 | 6.70 |
| | $v$-SimCLR | $v$-DINO | 2.06 | 11.5 |
| | $v$-SimCLR | $v$-Supervised | 8.45 | 16.0 |
| | $v$-SimCLR | $v$-MAE | -3.93 | -0.60 |
| | $v$-MoCo | $v$-BYOL | 1.81 | 4.80 |
| | $v$-MoCo | $v$-SimSiam | -0.94 | 5.20 |
| | $v$-MoCo | $v$-DINO | 2.00 | 10.0 |
| | $v$-MoCo | $v$-Supervised | 8.39 | 14.5 |
| | $v$-MoCo | $v$-MAE | -3.99 | -2.10 |

Table S18: Comparative statistical analysis of the robustness of contrastive methods ($v$-SimCLR and $v$-MoCo) in learning **viewpoint invariance** (original numbers from Figure 4(a)).

| Distribution shift | Method 1 | Method 2 | Adjusted $\Delta_{\text{OOD}}$ |
|---|---|---|---|
| | $v$-SimCLR | $v$-BYOL | 0.05 |
| | $v$-SimCLR | $v$-SimSiam | 0.04 |
| | $v$-SimCLR | $v$-DINO | 0.02 |
| | $v$-SimCLR | $v$-Supervised | 0.09 |
| | $v$-SimCLR | $v$-MAE | 0.09 |
| Viewpoint invariance | $v$-MoCo | $v$-BYOL | 0.04 |
| | $v$-MoCo | $v$-SimSiam | 0.03 |
| | $v$-MoCo | $v$-DINO | 0.02 |
| | $v$-MoCo | $v$-Supervised | 0.09 |
| | $v$-MoCo | $v$-MAE | 0.09 |

Table S19: Comparative statistical analysis of the robustness of single stream networks ($v$-Supervised and $v$-MAE) in learning **temporal dynamics** and Siamese networks ($v$-BYOL, $v$-SimSiam, $v$-DINO, $v$-SimCLR, and $v$-MoCo) in learning **viewpoint invariance** (original numbers from Figures 3(a) and 4(a)).

| Distribution shift | Method 1 | Method 2 | Adjusted $\Delta_{\text{OOD}}$ |
|---|---|---|---|
| | $v$-Supervised | $v$-BYOL | 0.06 |
| | $v$-Supervised | $v$-SimSiam | 0.05 |
| | $v$-Supervised | $v$-DINO | 0.05 |
| | $v$-Supervised | $v$-SimCLR | 0.06 |
| Temporal learner | $v$-Supervised | $v$-MoCo | 0.06 |
| | $v$-MAE | $v$-BYOL | 0.19 |
| | $v$-MAE | $v$-SimSiam | 0.19 |
| | $v$-MAE | $v$-DINO | 0.19 |
| | $v$-MAE | $v$-SimCLR | 0.20 |
| | $v$-MAE | $v$-MoCo | 0.20 |
| | $v$-SimCLR | $v$-Supervised | 0.09 |
| | $v$-SimCLR | $v$-MAE | 0.09 |
| | $v$-MoCo | $v$-Supervised | 0.09 |
| | $v$-MoCo | $v$-MAE | 0.09 |
| Viewpoint invariance | $v$-BYOL | $v$-Supervised | 0.05 |
| | $v$-BYOL | $v$-MAE | 0.05 |
| | $v$-SimSiam | $v$-Supervised | 0.06 |
| | $v$-SimSiam | $v$-MAE | 0.06 |
| | $v$-DINO | $v$-Supervised | 0.07 |
| | $v$-DINO | $v$-MAE | 0.07 |

Table S20: Comparative statistical analysis of the vulnerability (opposite of robustness) of $v$-Supervised under **multiple shifts** (original numbers from Table 2).

| Distribution shift | Method 1 | Method 2 | Adjusted $\Delta_{\text{OOD}}$ | |
|---|---|---|---|---|
| | | | Lin. | FT. |
| | $v$-Supervised | $v$-BYOL | -7.11 | -9.70 |
| | $v$-Supervised | $v$-SimSiam | -9.87 | -9.30 |
| View+Act (t-down+syn.) | $v$-Supervised | $v$-DINO | -6.92 | -4.50 |
| | $v$-Supervised | $v$-SimCLR | -10.6 | -16.0 |
| | $v$-Supervised | $v$-MoCo | -9.26 | -14.5 |
| | $v$-Supervised | $v$-MAE | -12.9 | -16.6 |

Table S21: Comparative statistical analysis of the robustness of $v$-BYOL under **source shift** (original numbers from Table 3).

| Distribution shift | Method 1 | Method 2 | Adjusted $\Delta_{OOD}$ | |
| --- | --- | --- | --- | --- |
| | | | Lin. | FT. |
| HMDB to UCF | $v$-BYOL | $v$-Supervised | 8.56 | 6.60 |
| | $v$-BYOL | $v$-SimSiam | 6.96 | 2.90 |
| | $v$-BYOL | $v$-DINO | 6.70 | 0.30 |
| | $v$-BYOL | $v$-SimCLR | 9.96 | 2.00 |
| | $v$-BYOL | $v$-MoCo | 10.3 | 1.10 |
| | $v$-BYOL | $v$-MAE | 16.7 | 10.8 |
| UCF to HMDB | $v$-BYOL | $v$-Supervised | 3.14 | 2.40 |
| | $v$-BYOL | $v$-SimSiam | 2.72 | 6.80 |
| | $v$-BYOL | $v$-DINO | 1.23 | 6.80 |
| | $v$-BYOL | $v$-SimCLR | 4.52 | 5.90 |
| | $v$-BYOL | $v$-MoCo | 1.89 | 5.10 |
| | $v$-BYOL | $v$-MAE | 6.00 | 3.10 |

Table S22: Comparative statistical analysis of the **impacts of finetuning over frozen encoder** under source shift (original numbers from Table 3).

| Distribution shift | Method 1 | Method 2 | Adjusted $\Delta_{OOD}$ |
| --- | --- | --- | --- |
| HMDB to UCF | Finetune | Linear | -8.67 |
| UCF to HMDB | Finetune | Linear | 1.97 |

Table S23: Comparative statistical analysis of the robustness of contrastive methods ($v$-SimCLR and $v$-MoCo) in **open-set recognition** when finetuned (original numbers from Table 6(a)).

| Distribution shift | Method 1 | Method 2 | Adjusted $\Delta_{OOD}$ |
| --- | --- | --- | --- |
| Kinetics400/UCF | $v$-SimCLR | $v$-BYOL | 2.44 |
| | $v$-SimCLR | $v$-SimSiam | 2.42 |
| | $v$-SimCLR | $v$-DINO | 2.94 |
| | $v$-SimCLR | $v$-MAE | 7.85 |
| | $v$-MoCo | $v$-BYOL | 0.72 |
| | $v$-MoCo | $v$-SimSiam | 0.70 |
| | $v$-MoCo | $v$-DINO | 1.22 |
| | $v$-MoCo | $v$-MAE | 6.13 |
| Kinetics400/HMDB | $v$-SimCLR | $v$-BYOL | 0.04 |
| | $v$-SimCLR | $v$-SimSiam | 2.12 |
| | $v$-SimCLR | $v$-DINO | 1.03 |
| | $v$-SimCLR | $v$-MAE | 5.09 |
| | $v$-MoCo | $v$-BYOL | 0.76 |
| | $v$-MoCo | $v$-SimSiam | 2.84 |
| | $v$-MoCo | $v$-DINO | 1.75 |
| | $v$-MoCo | $v$-MAE | 5.82 |
| UCF101/HMDB | $v$-SimCLR | $v$-BYOL | 2.58 |
| | $v$-SimCLR | $v$-SimSiam | 4.29 |
| | $v$-SimCLR | $v$-DINO | 1.31 |
| | $v$-SimCLR | $v$-MAE | 11.38 |
| | $v$-MoCo | $v$-BYOL | 3.20 |
| | $v$-MoCo | $v$-SimSiam | 4.91 |
| | $v$-MoCo | $v$-DINO | 1.94 |
| | $v$-MoCo | $v$-MAE | 12.0 |

Table S24: Comparative statistical analysis of the robustness of slightly weak frozen encoders ($v$-DINO, $v$-SimSiam, and $v$-Supervised) in **open-set recognition** in linear evaluation (the original numbers from Table 6(b)).

| Distribution shift | Method 1 | Method 2 | Adjusted $\Delta_{\text{OOD}}$ |
|---|---|---|---|
| UCF101/HMDB | $v$-DINO | $v$-BYOL | 23.4 |
| | $v$-DINO | $v$-SimCLR | 29.5 |
| | $v$-DINO | $v$-MoCo | 28.7 |
| | $v$-DINO | $v$-MAE | 13.4 |
| | $v$-SimSiam | $v$-BYOL | 17.5 |
| | $v$-SimSiam | $v$-SimCLR | 23.5 |
| | $v$-SimSiam | $v$-MoCo | 22.7 |
| | $v$-SimSiam | $v$-MAE | 7.40 |
| | $v$-Supervised | $v$-BYOL | 21.2 |
| | $v$-Supervised | $v$-SimCLR | 27.3 |
| | $v$-Supervised | $v$-MoCo | 26.5 |
| | $v$-Supervised | $v$-MAE | 11.2 |

Table S25: Part 1. Comparative statistical analysis of the robustness of all methods in **zero-shot recognition** (original numbers from Table 4).

| Distribution shift | Method 1 | Method 2 | Adjusted $\Delta_{\text{OOD}}$ | |
|---|---|---|---|---|
| | | | Lin. | FT. |
| Zero-shot on UCF | $v$-BYOL | $v$-SimSiam | -5.02 | -0.24 |
| | $v$-BYOL | $v$-DINO | -4.01 | -2.13 |
| | $v$-BYOL | $v$-MAE | -2.89 | -1.56 |
| | $v$-BYOL | $v$-SimCLR | -5.37 | 0.95 |
| | $v$-BYOL | $v$-MoCo | -2.70 | -5.30 |
| | $v$-SimSiam | $v$-BYOL | 2.31 | 0.24 |
| | $v$-SimSiam | $v$-DINO | -0.42 | -1.89 |
| | $v$-SimSiam | $v$-MAE | 0.70 | -1.32 |
| | $v$-SimSiam | $v$-SimCLR | -1.79 | 1.19 |
| | $v$-SimSiam | $v$-MoCo | 0.89 | -5.06 |
| | $v$-DINO | $v$-BYOL | 1.78 | 2.13 |
| | $v$-DINO | $v$-SimSiam | -1.96 | 1.89 |
| | $v$-DINO | $v$-MAE | 0.17 | 0.57 |
| | $v$-DINO | $v$-SimCLR | -2.31 | 3.08 |
| | $v$-DINO | $v$-MoCo | 0.36 | -3.17 |
| | $v$-MAE | $v$-BYOL | 0.28 | 1.56 |
| | $v$-MAE | $v$-SimSiam | -3.46 | 1.32 |
| | $v$-MAE | $v$-DINO | -2.45 | -0.57 |
| | $v$-MAE | $v$-SimCLR | -3.81 | 2.51 |
| | $v$-MAE | $v$-MoCo | -1.14 | -3.74 |
| | $v$-SimCLR | $v$-BYOL | 2.40 | -0.95 |
| | $v$-SimCLR | $v$-SimSiam | -1.34 | -1.19 |
| | $v$-SimCLR | $v$-DINO | -0.32 | -3.08 |
| | $v$-SimCLR | $v$-MAE | 0.79 | -2.51 |
| | $v$-SimCLR | $v$-MoCo | 0.99 | -6.25 |
| | $v$-MoCo | $v$-BYOL | -1.40 | 5.30 |
| | $v$-MoCo | $v$-SimSiam | -5.14 | 5.06 |
| | $v$-MoCo | $v$-DINO | -4.12 | 3.17 |
| | $v$-MoCo | $v$-MAE | -3.01 | 3.74 |
| | $v$-MoCo | $v$-SimCLR | -5.49 | 6.25 |

Table S25 Part 2. Comparative statistical analysis of the robustness of all methods in **zero-shot recognition** (original numbers from Table 4).

| Distribution shift | Method 1 | Method 2 | Adjusted $\Delta_{OOD}$ | |
|---|---|---|---|---|
| | | | Lin. | FT. |
| | $v$-BYOL | $v$-SimSiam | -0.18 | 1.05 |
| | $v$-BYOL | $v$-DINO | 2.17 | -0.01 |
| | $v$-BYOL | $v$-MAE | 0.55 | -1.58 |
| | $v$-BYOL | $v$-SimCLR | 1.64 | 0.06 |
| | $v$-BYOL | $v$-MoCo | 1.27 | 2.40 |
| | $v$-SimSiam | $v$-BYOL | -2.61 | -1.05 |
| | $v$-SimSiam | $v$-DINO | 1.48 | -1.06 |
| | $v$-SimSiam | $v$-MAE | -0.14 | -2.63 |
| | $v$-SimSiam | $v$-SimCLR | 0.94 | -0.99 |
| | $v$-SimSiam | $v$-MoCo | 0.58 | 1.35 |
| | $v$-DINO | $v$-BYOL | -4.35 | 0.01 |
| | $v$-DINO | $v$-SimSiam | -2.61 | 1.06 |
| | $v$-DINO | $v$-MAE | -1.88 | -1.57 |
| | $v$-DINO | $v$-SimCLR | -0.80 | 0.07 |
| | $v$-DINO | $v$-MoCo | -1.16 | 2.41 |
| Zero-shot on HMDB | $v$-MAE | $v$-BYOL | -3.15 | 1.58 |
| | $v$-MAE | $v$-SimSiam | -1.42 | 2.63 |
| | $v$-MAE | $v$-DINO | 0.94 | 1.57 |
| | $v$-MAE | $v$-SimCLR | 0.40 | 1.64 |
| | $v$-MAE | $v$-MoCo | 0.04 | 3.98 |
| | $v$-SimCLR | $v$-BYOL | -5.14 | -0.06 |
| | $v$-SimCLR | $v$-SimSiam | -3.40 | 0.99 |
| | $v$-SimCLR | $v$-DINO | -1.05 | -0.07 |
| | $v$-SimCLR | $v$-MAE | -2.67 | -1.64 |
| | $v$-SimCLR | $v$-MoCo | -1.95 | 2.34 |
| | $v$-MoCo | $v$-BYOL | -5.55 | -2.40 |
| | $v$-MoCo | $v$-SimSiam | -3.81 | -1.35 |
| | $v$-MoCo | $v$-DINO | -1.46 | -2.41 |
| | $v$-MoCo | $v$-MAE | -3.08 | -3.98 |
| | $v$-MoCo | $v$-SimCLR | -1.99 | -2.34 |

Table S25 Part 3. Comparative statistical analysis of the robustness of all methods in **zero-shot recognition** (original numbers from Table 4).

| Distribution shift | Method 1 | Method 2 | Adjusted $\Delta_{\text{OOD}}$ | |
| --- | --- | --- | --- | --- |
| | | | Lin. | FT. |
| | $v$-BYOL | $v$-SimSiam | -1.19 | -0.95 |
| | $v$-BYOL | $v$-DINO | -1.36 | -0.31 |
| | $v$-BYOL | $v$-MAE | -0.82 | -0.43 |
| | $v$-BYOL | $v$-SimCLR | -0.59 | -0.09 |
| | $v$-BYOL | $v$-MoCo | -1.38 | -0.20 |
| | $v$-SimSiam | $v$-BYOL | 0.59 | 0.95 |
| | $v$-SimSiam | $v$-DINO | -0.43 | 0.64 |
| | $v$-SimSiam | $v$-MAE | 0.11 | 0.52 |
| | $v$-SimSiam | $v$-SimCLR | 0.33 | 0.86 |
| | $v$-SimSiam | $v$-MoCo | -0.45 | 0.75 |
| | $v$-DINO | $v$-BYOL | 0.79 | 0.31 |
| | $v$-DINO | $v$-SimSiam | -0.08 | -0.64 |
| | $v$-DINO | $v$-MAE | 0.30 | -0.12 |
| | $v$-DINO | $v$-SimCLR | 0.53 | 0.22 |
| | $v$-DINO | $v$-MoCo | -0.26 | 0.11 |
| Zero-shot on RareAct | $v$-MAE | $v$-BYOL | -0.39 | 0.43 |
| | $v$-MAE | $v$-SimSiam | -1.25 | -0.52 |
| | $v$-MAE | $v$-DINO | -1.41 | 0.12 |
| | $v$-MAE | $v$-SimCLR | -0.65 | 0.34 |
| | $v$-MAE | $v$-MoCo | -1.43 | 0.23 |
| | $v$-SimCLR | $v$-BYOL | -0.26 | 0.09 |
| | $v$-SimCLR | $v$-SimSiam | -1.12 | -0.86 |
| | $v$-SimCLR | $v$-DINO | -1.28 | -0.22 |
| | $v$-SimCLR | $v$-MAE | -0.75 | -0.34 |
| | $v$-SimCLR | $v$-MoCo | -1.3 | -0.11 |
| | $v$-MoCo | $v$-BYOL | 0.50 | 0.20 |
| | $v$-MoCo | $v$-SimSiam | -0.36 | -0.75 |
| | $v$-MoCo | $v$-DINO | -0.52 | -0.11 |
| | $v$-MoCo | $v$-MAE | 0.01 | -0.23 |
| | $v$-MoCo | $v$-SimCLR | 0.24 | 0.11 |

Table S26: Comparative statistical analysis of the robustness of all methods under **animal domain actor shift** (original numbers from Table 2).

| Distribution shift | Method 1 | Method 2 | Adjusted $\Delta_{\text{OOD}}$ | |
| --- | --- | --- | --- | --- |
| | | | Lin. | FT. |
| | $v$-Supervised | $v$-SimCLR | -1.86 | -3.00 |
| | $v$-Supervised | $v$-MoCo | -1.50 | -1.20 |
| | $v$-Supervised | $v$-BYOL | -2.46 | -1.20 |
| | $v$-Supervised | $v$-SimSiam | -3.24 | -2.00 |
| | $v$-Supervised | $v$-DINO | -1.70 | -1.90 |
| | $v$-Supervised | $v$-MAE | 0.39 | -2.40 |
| | $v$-SimCLR | $v$-Supervised | 0.74 | 3.00 |
| | $v$-SimCLR | $v$-MoCo | -0.2 | 1.80 |
| | $v$-SimCLR | $v$-BYOL | -1.16 | 1.80 |
| | $v$-SimCLR | $v$-SimSiam | -1.94 | 1.00 |
| | $v$-SimCLR | $v$-DINO | -0.40 | 1.10 |
| | $v$-SimCLR | $v$-MAE | 1.69 | 0.60 |
| | $v$-MoCo | $v$-Supervised | 0.09 | 1.20 |
| | $v$-MoCo | $v$-SimCLR | -1.21 | -1.80 |
| | $v$-MoCo | $v$-BYOL | -1.82 | 0.00 |
| | $v$-MoCo | $v$-SimSiam | -2.59 | -0.80 |
| | $v$-MoCo | $v$-DINO | -1.05 | -0.70 |
| | $v$-MoCo | $v$-MAE | 1.04 | -1.20 |
| | $v$-BYOL | $v$-Supervised | 1.58 | 1.20 |
| | $v$-BYOL | $v$-SimCLR | 0.28 | -1.80 |
| | $v$-BYOL | $v$-MoCo | 0.64 | 0.00 |
| Animal domain actor shift | $v$-BYOL | $v$-SimSiam | -1.10 | -0.80 |
| | $v$-BYOL | $v$-DINO | 0.45 | -0.70 |
| | $v$-BYOL | $v$-MAE | 2.54 | -1.20 |
| | $v$-SimSiam | $v$-Supervised | 1.27 | 2.00 |
| | $v$-SimSiam | $v$-SimCLR | -0.03 | -1.00 |
| | $v$-SimSiam | $v$-MoCo | 0.33 | 0.80 |
| | $v$-SimSiam | $v$-BYOL | -0.63 | 0.80 |
| | $v$-SimSiam | $v$-DINO | 0.13 | 0.10 |
| | $v$-SimSiam | $v$-MAE | 2.22 | -0.40 |
| | $v$-DINO | $v$-Supervised | 0.67 | 1.90 |
| | $v$-DINO | $v$-SimCLR | -0.63 | -1.10 |
| | $v$-DINO | $v$-MoCo | -0.27 | 0.70 |
| | $v$-DINO | $v$-BYOL | -1.23 | 0.70 |
| | $v$-DINO | $v$-SimSiam | -2.01 | -0.10 |
| | $v$-DINO | $v$-MAE | 1.63 | -0.50 |
| | $v$-MAE | $v$-Supervised | -1.60 | 2.40 |
| | $v$-MAE | $v$-SimCLR | -2.90 | -0.60 |
| | $v$-MAE | $v$-MoCo | -2.55 | 1.20 |
| | $v$-MAE | $v$-BYOL | -3.51 | 1.20 |
| | $v$-MAE | $v$-SimSiam | -4.28 | 0.40 |
| | $v$-MAE | $v$-DINO | -2.74 | 0.50 |

### D.8 Exploring performance variation of VSSL methods across action classes

To provide a deeper understanding, we conduct a comprehensive analysis examining the impact of distribution shifts on the models' prediction across various action classes. Our investigation focuses on evaluating the performance of the finetuned models across different categories in both the InD and OoD. The findings are presented as follows:

- Figure S19 Context shift (10 classes)
- Figure S20 Context shift (50 classes)
- Figure S21 Viewpoint shift (egocentric)
- Figure S22 Viewpoint shift (surveillance camera+low resolution)
- Figure S23 Viewpoint + actor shift (top-down + synthetic)
- Figure S24 Actor shift (animal)
- Figure S25 Source shift (UCF to HMDB)
- Figure S26 Source shift (HMDB to UCF)
- Figure S28 Zero-shot (UCF)
- Figure S27 Zero-shot (HMDB)
- Figure S29 Zero-shot (RareAct)
- Figure S30 Open-set (UCF/HMDB)
- Figure S32 Open-set (Kinetics400/UCF)
- Figure S31 Open-set (Kinetics400/HMDB)

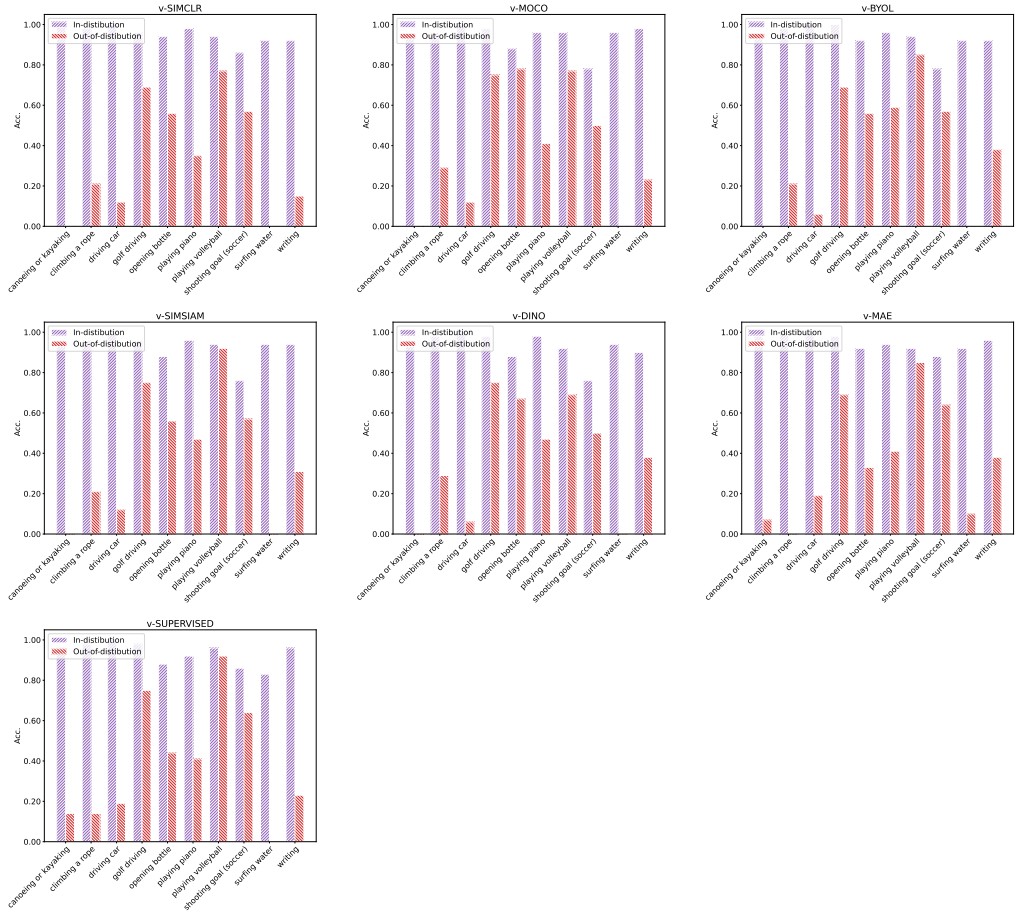

Figure S19: **Context shift (10 classes)**: In-distribution vs. out-of-distribution performance comparison per action class. For optimal viewing, please zoom in. All of the contrastive and non-contrastive methods fail to identify 'canoeing' in OoD, whereas, $v$-Supervised and $v$-MAE make a few correct predictions. In another extreme example, $v$-MAE makes a few correct predictions for 'surfing water', while other methods completely fail. In particular, VSSL methods demonstrate relatively good performance in identifying certain action classes in out-of-context scenarios, such as 'golf driving' and 'playing volleyball'.

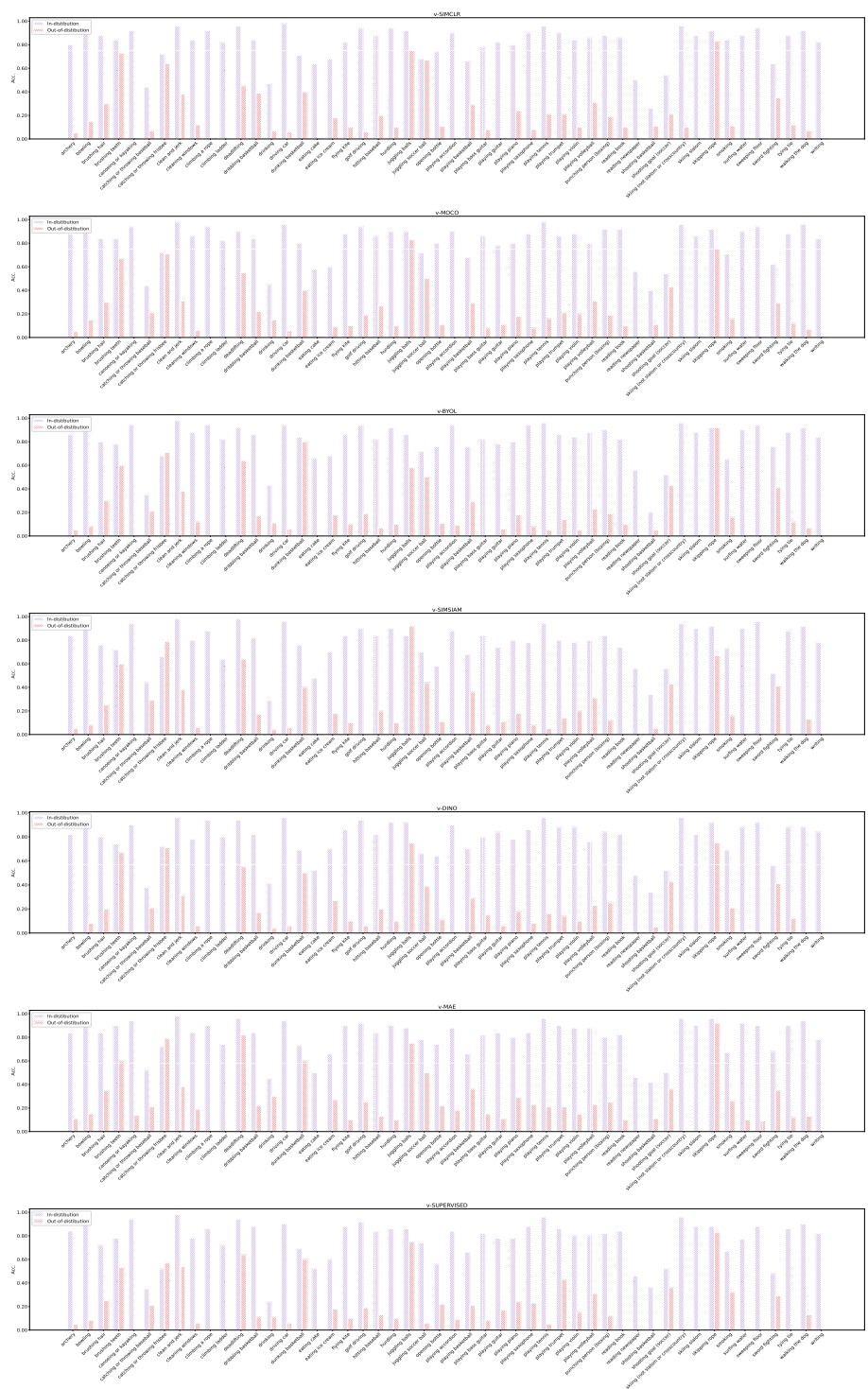

Figure S20: **Context shift (50 classes)**: In-distribution vs. out-of-distribution performance comparison per action class. For optimal viewing, please zoom in. Some of the top-performing action classes under context shift include 'brushing teeth', 'catching or throwing frisbee', 'juggling balls', 'juggling soccer ball', and 'skipping rope'. It is noteworthy that all of these action classes involve high temporal motion and cyclic patterns. This observation leads us to speculate that the presence of these attributes may assist video models in recognizing actions in out-of-context settings.

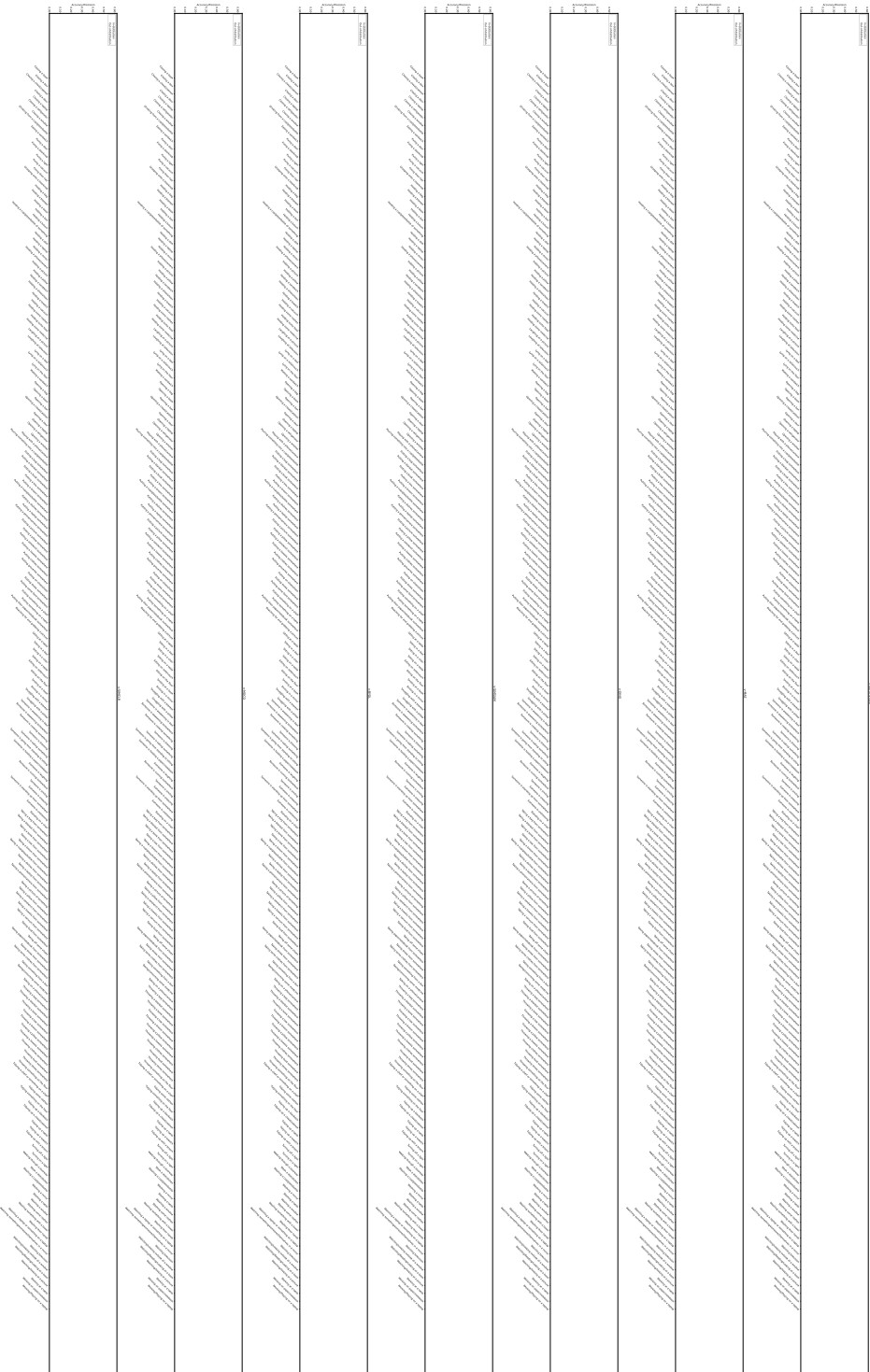

Figure S21: **Viewpoint shift (egocentric)**: In-distribution vs. out-of-distribution performance comparison per action class. For optimal viewing, please zoom in and rotate 90 degrees to the left ↻. A few examples of poor OoD generalization include action classes such as: 'lying on a bed', 'fixing a vacuum', and 'someone is closing something', among a few others. We also note a few instances, where models show very similar InD and OoD performance, such as 'holding some food', 'holding a boom', and 'taking food from somewhere', among a few others.

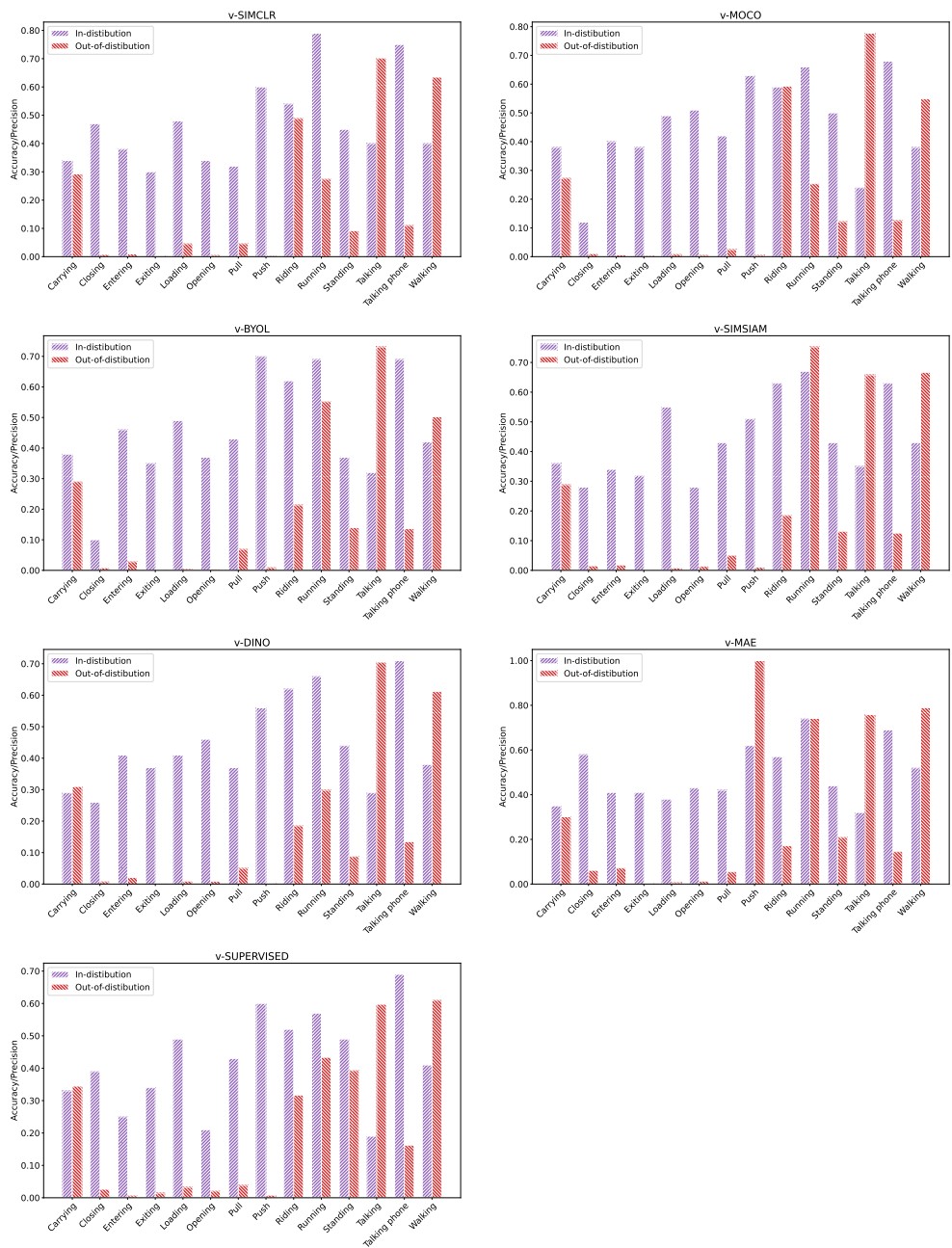

Figure S22: **Viewpoint shift (surveillance view + low resolution)**: In-distribution vs. out-of-distribution performance comparison per action class. For optimal viewing, please zoom in. Interestingly, both $v$-BYOL and $v$-SimSiam perform well in identifying 'running' but failed to identify 'riding'. On the other hand, $v$-SimCLR and $v$-MoCo correctly identify 'riding' but fail to predict 'running'. Overall, models show poor performance in distinguishing actions with subtle differences like 'closing' vs. 'opening', 'entering' vs. 'exiting', and 'pull' vs. 'push'.

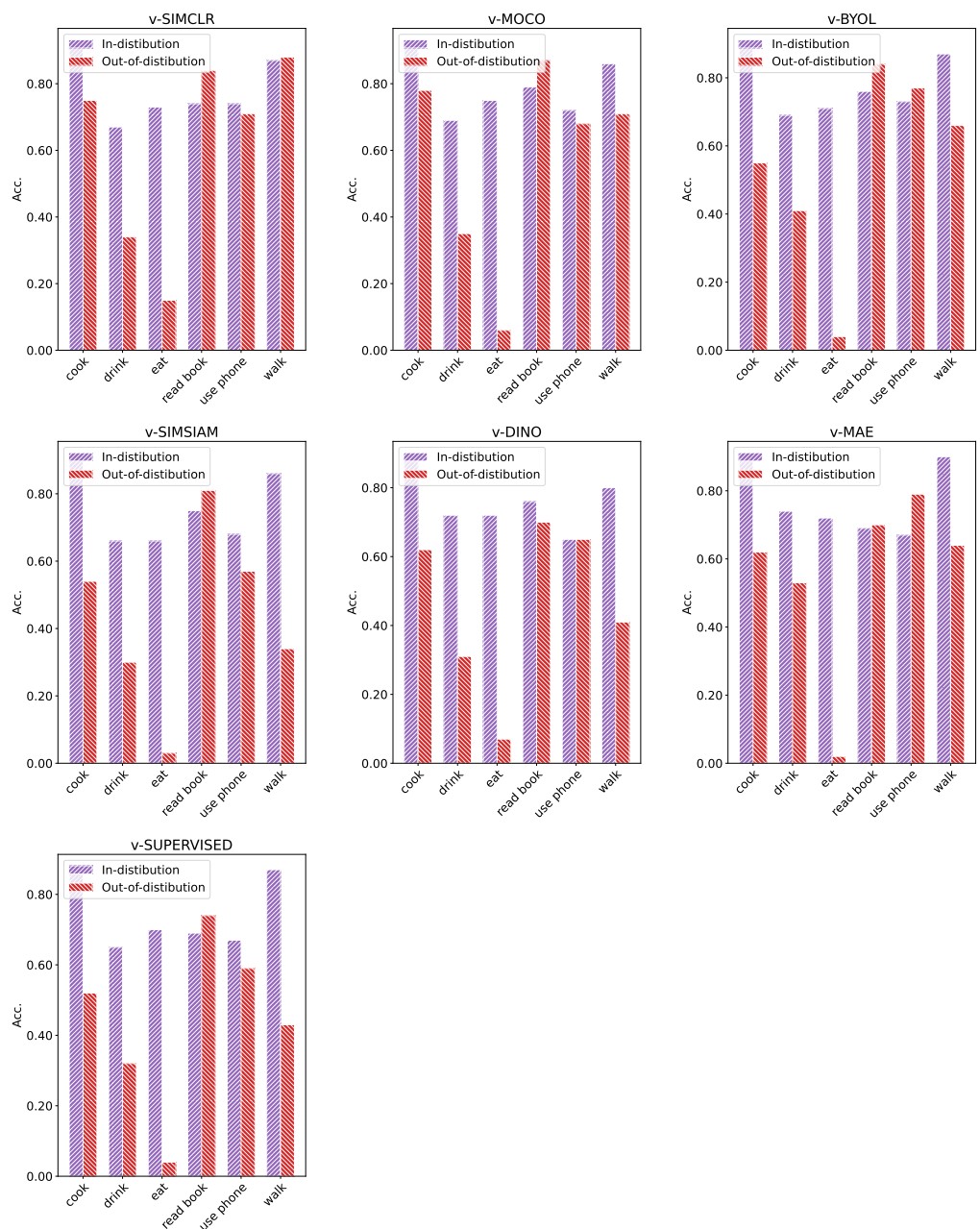

Figure S23: **Viewpoint + actor shift (top-down + synthetic)**: In-distribution vs. out-of-distribution performance comparison per action class. For optimal viewing, please zoom in. The models exhibit particularly poor performance in identifying the action classes 'eat', followed by 'drink'. Conversely, they demonstrate relatively better performance in identifying actions such as 'read book', 'use phone', and 'walk'.

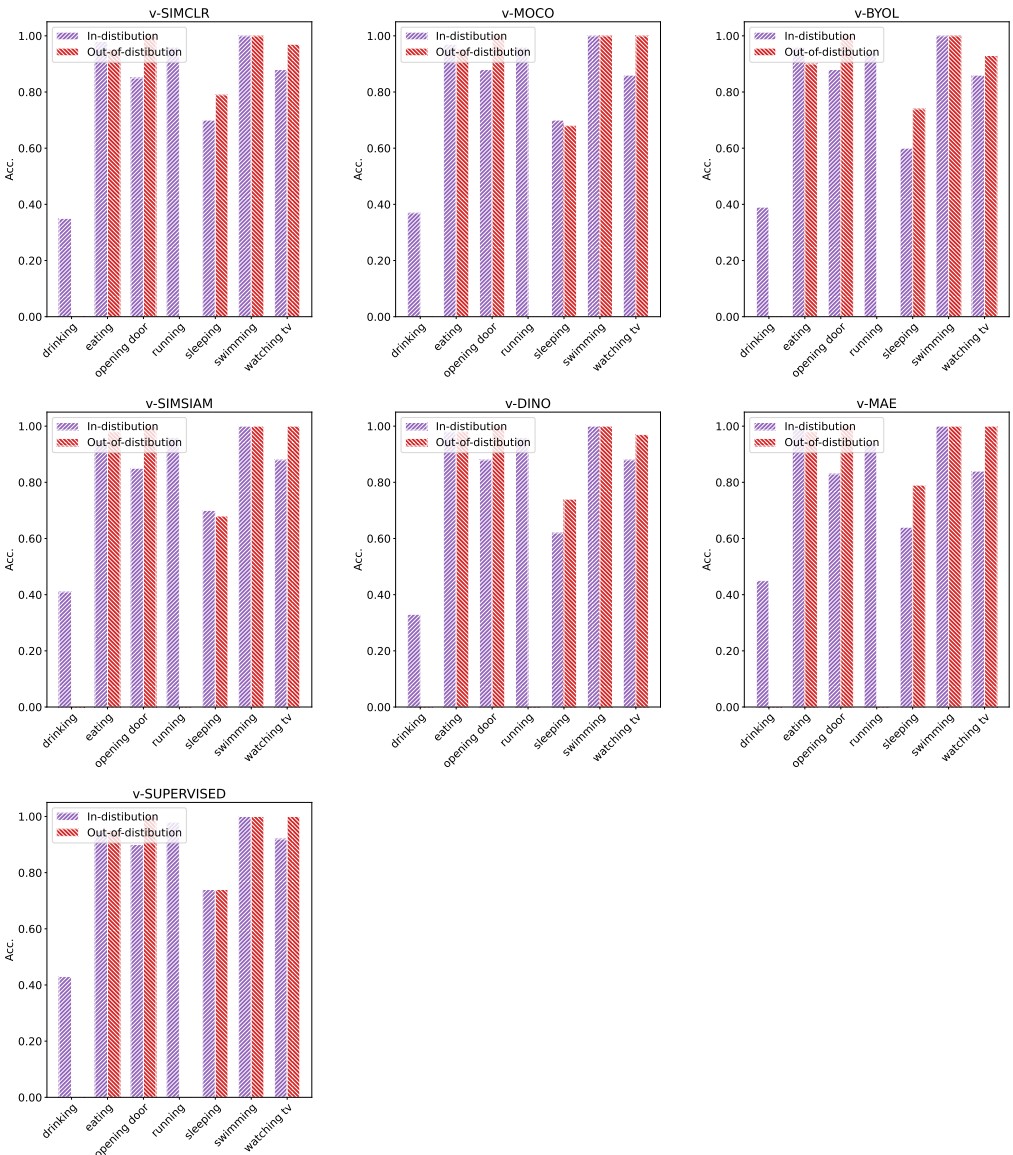

Figure S24: **Actor shift (animal)**: In-distribution vs. out-of-distribution performance comparison per action class. For optimal viewing, please zoom in. The models exhibit particularly poor performance in identifying action classes 'drinking' and 'running'. Conversely, they perform well in 'predicting eating', 'opening door', 'swimming', and 'watching tv'.

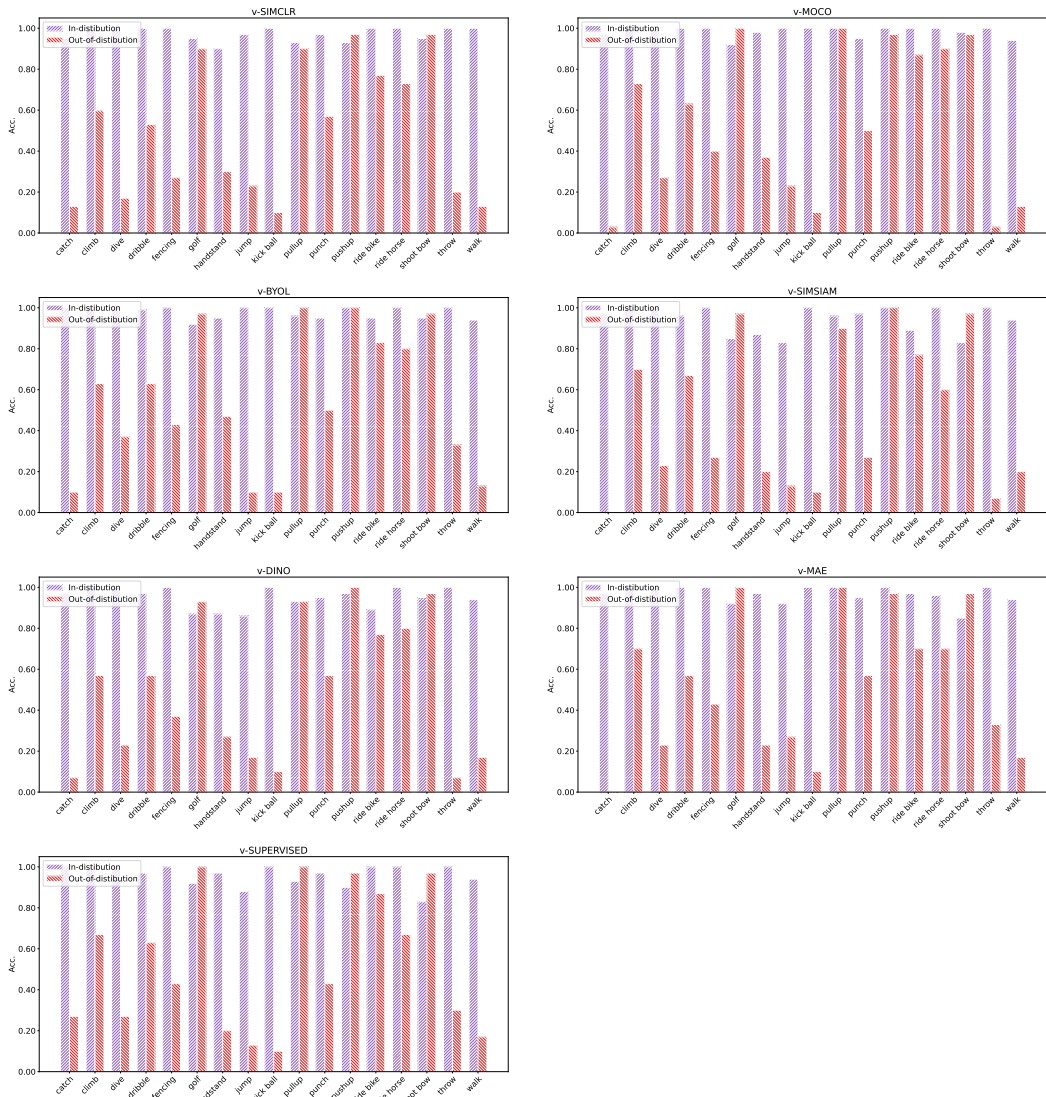

Figure S25: **Source shift (UCF to HMDB)**: In-distribution vs. out-of-distribution performance comparison per action class. For optimal viewing, please zoom in. The results demonstrate the challenges faced by the video models in generalizing across similar action classes from different datasets. For instance, the models trained on UCF videos which include 'soccer penalty', struggle to generalize to the 'kick ball' class in HMDB. Similarly, the 'walking with dog' class in UCF does not aid in generalizing to 'walk' in HMDB.

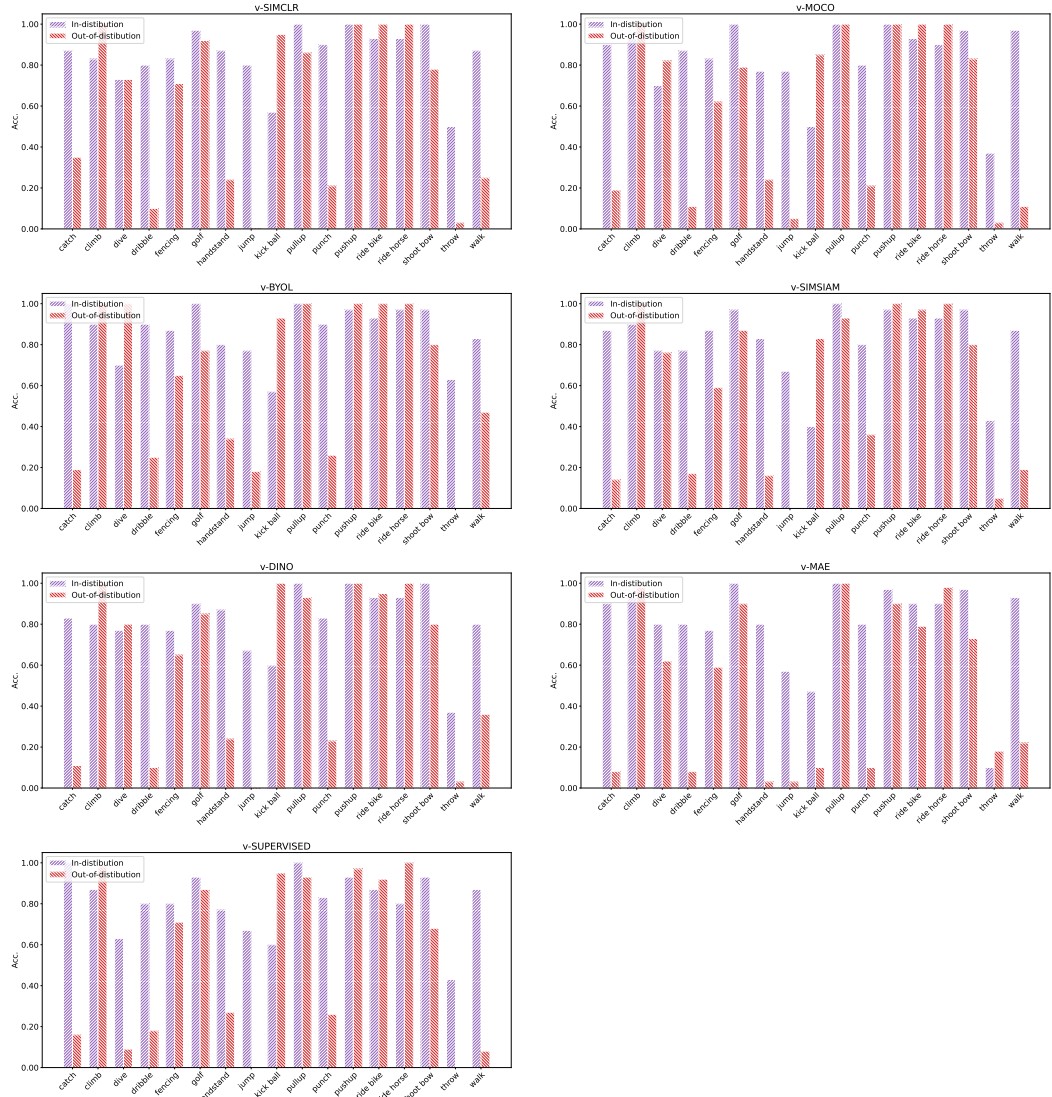

Figure S26: **Source shift (HMDB to UCF)**: In-distribution vs. out-of-distribution performance comparison per action class. For optimal viewing, please zoom in. We notice large variability in performances across different methods. For example, $v$-Supervised performs very poorly in identifying the action 'dive', whereas, all the self-supervised methods perform reasonably well. Furthermore, while $v$-BYOL and $v$-DINO demonstrate strong performance in identifying the action 'walk', the other methods struggle to achieve a similar level of accuracy. Overall, the VSSL methods tend to exhibit poor generalization capabilities for actions such as 'jump', 'throw', and 'dribble', among others.

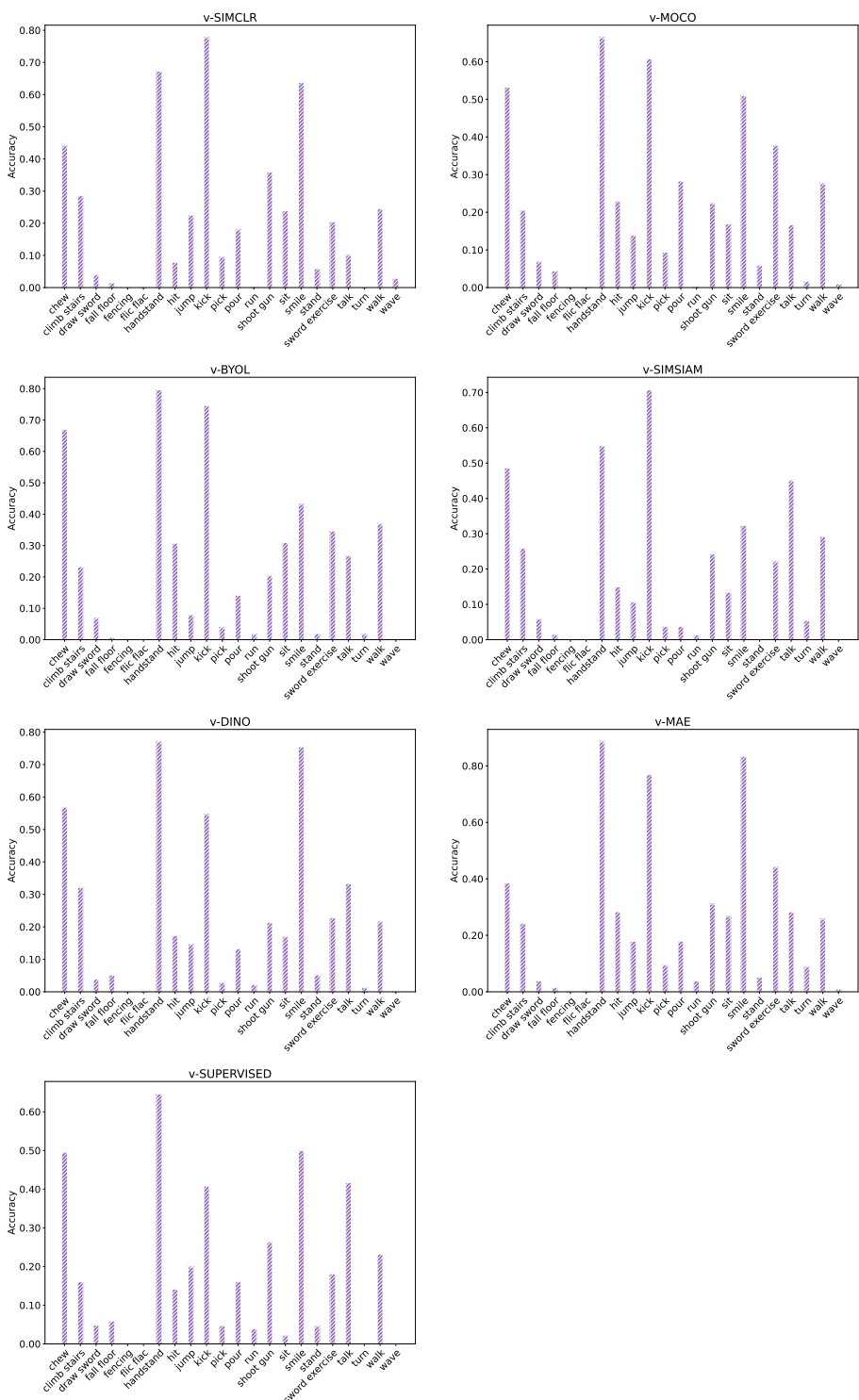

Figure S27: **Zero-shot recognition (HMDB)**: In-distribution vs. out-of-distribution performance comparison per action class. For optimal viewing, please zoom in. The video models overall generalize well on action classes such as 'chew', 'handstand', 'kick', and 'smile', whereas, they show poor generalizability on other action classes such as 'fencing', 'flic flac', 'pick', 'run', and 'turn' among others. Moreover, their performance largely varies across different actions.

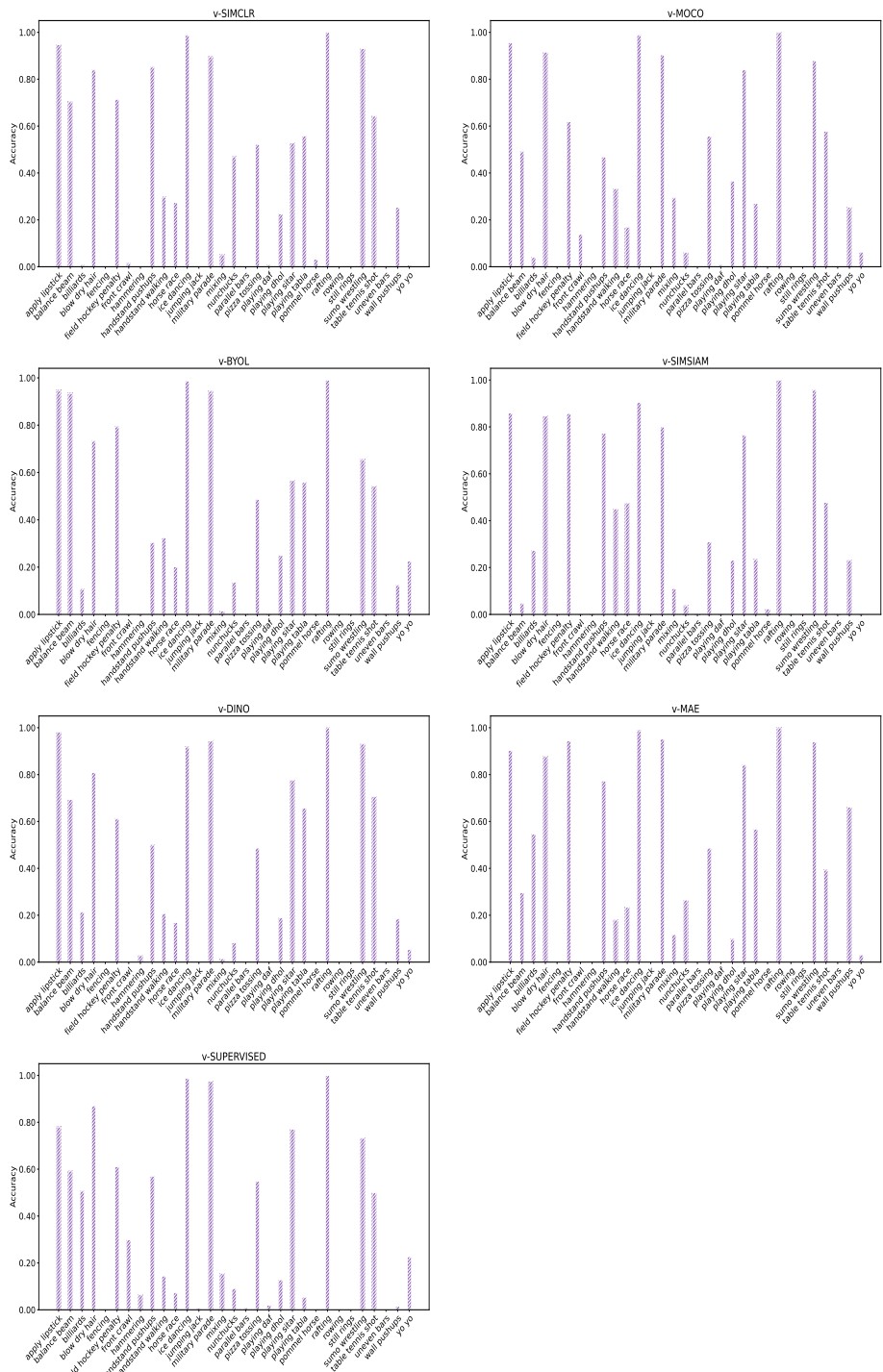

Figure S28: **Zero-shot recognition (UCF)**: In-distribution vs. out-of-distribution performance comparison per action class. For optimal viewing, please zoom in. The video models overall generalize well on action classes such as 'apply lipstick', 'rafting', 'ice dancing', and 'sumo wrestling', whereas, they show poor generalizability on action classes such as 'fencing', 'front crawl', 'hammering', 'mixing', and 'rowing' among others. Moreover, their performance largely varies across different actions.

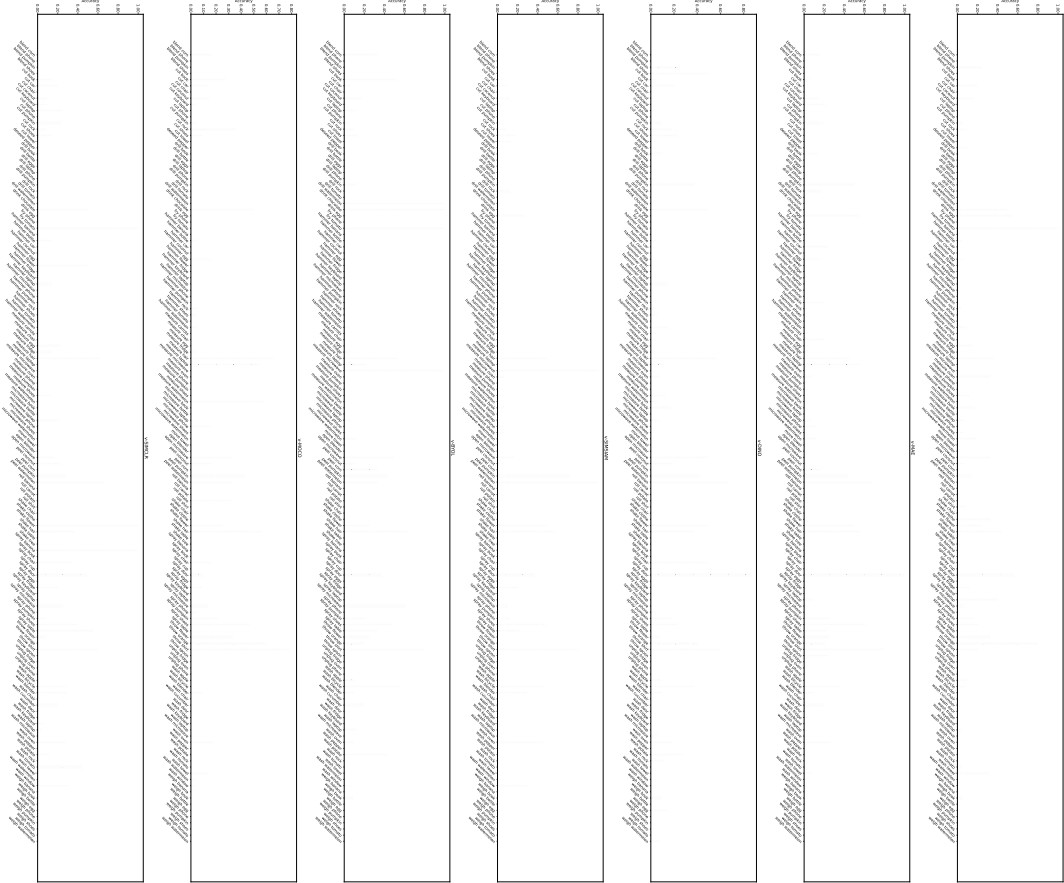

Figure S29: **Zero-shot recognition (RareAct)**: In-distribution vs. out-of-distribution performance comparison per action class. For optimal viewing, please zoom in and rotate 90 degrees to the left ↻. The models tend to generalize poorly on unusual actions. Moreover, compared to other zero-shot setups, we notice the maximum variability in model performance across different action classes.

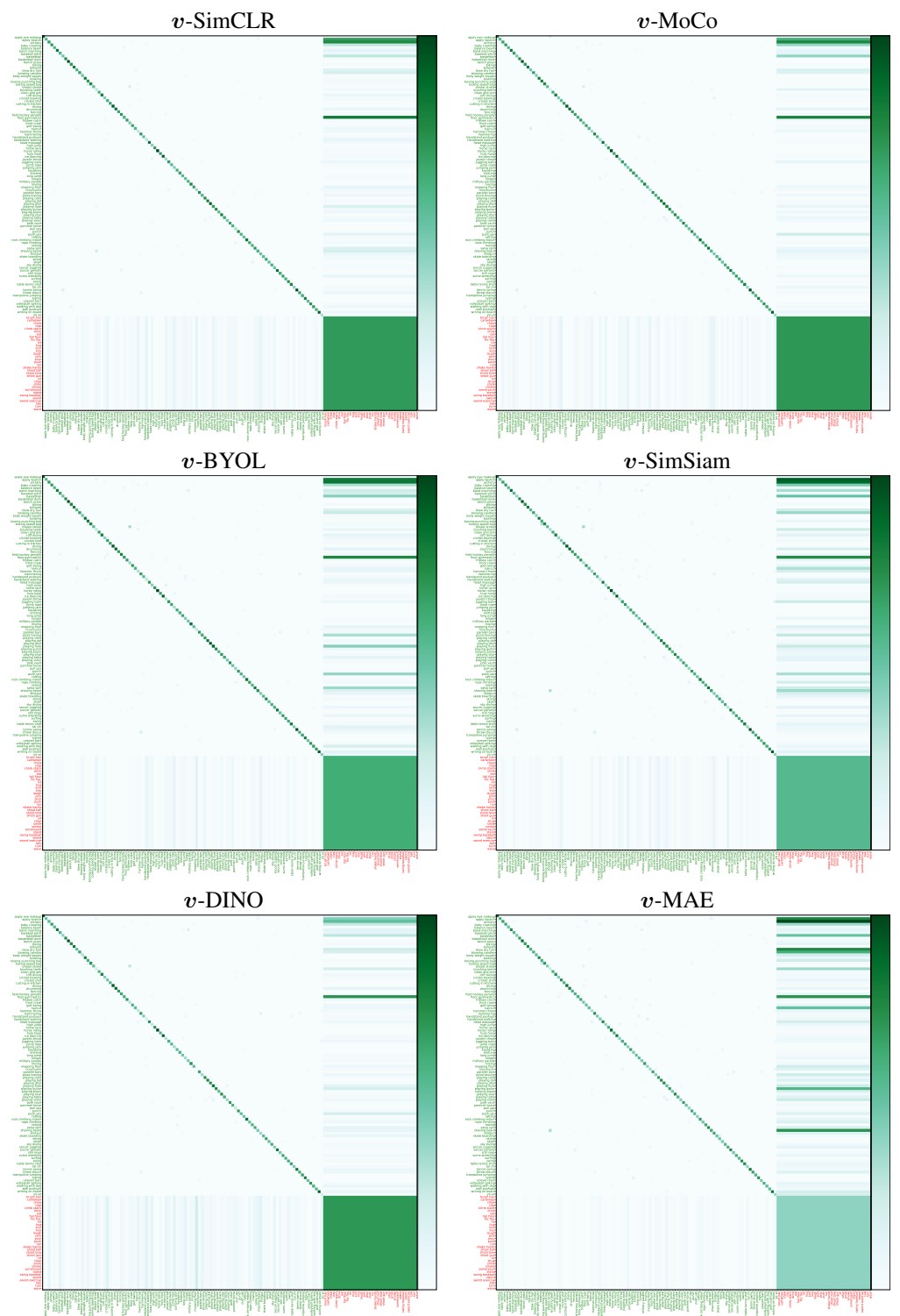

Figure S30: Confusion matrices for open-set recognition, using **UCF101 as closed-set and HMDB as open-set**. The x-axis represents the ground truth and the y-axis represents the predicted labels. We highlight known classes with green and unknown classes with red. Among all the methods $v$-SimSiam and $v$-MAE show worse performance in identifying 'unknown' classes from HMDB as 'unknown'.

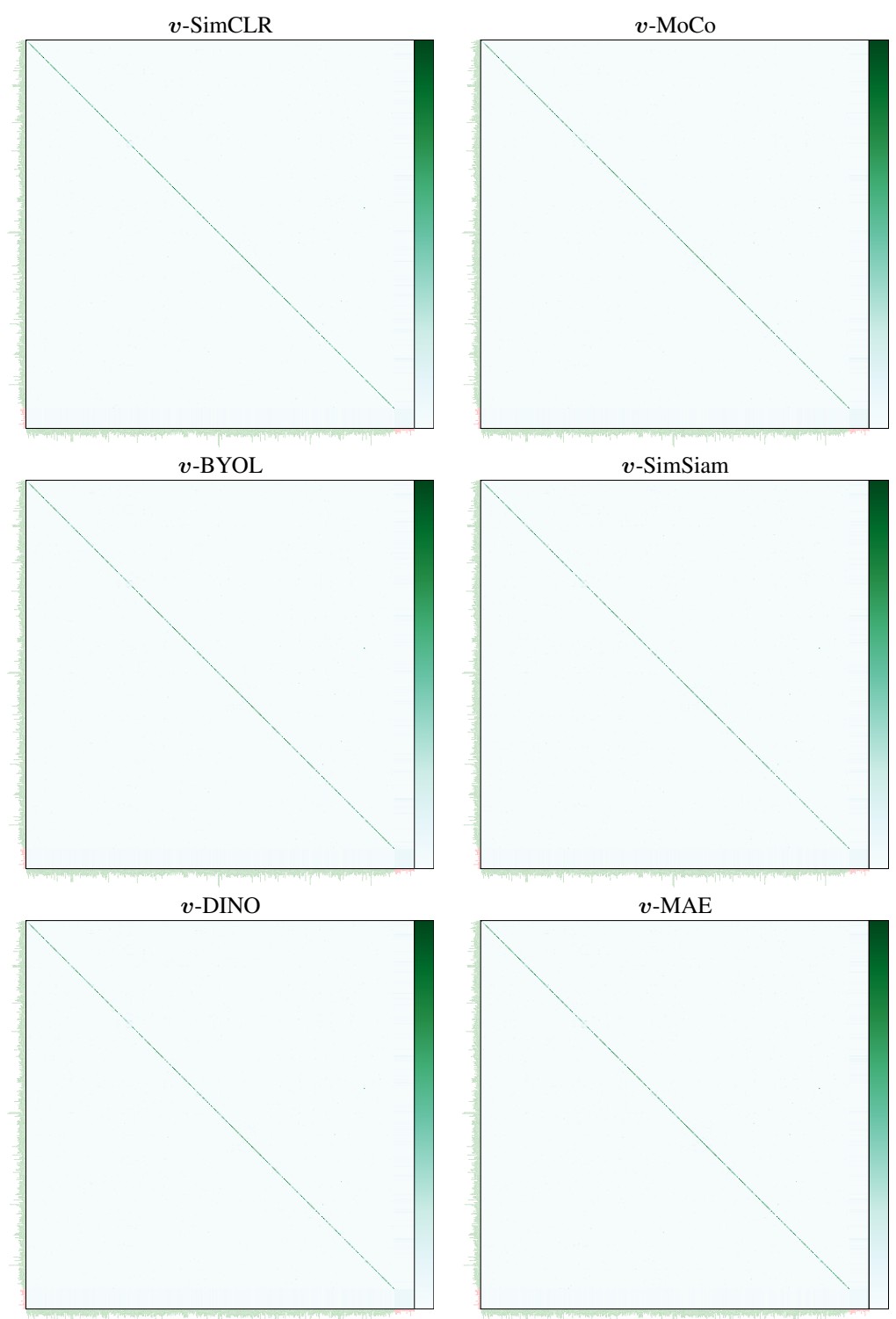

Figure S31: Confusion matrices of open-set recognition, using **Kinetics400 as closed-set and HMDB as open-set.** The x-axis represents the ground truth and the y-axis represents the predicted labels. We highlight known classes with green and unknown classes with red. The results exhibit that all the video models struggle in identifying 'unknown' classes from HMDB as 'unknown'. This is likely due to their over-confident predictions.

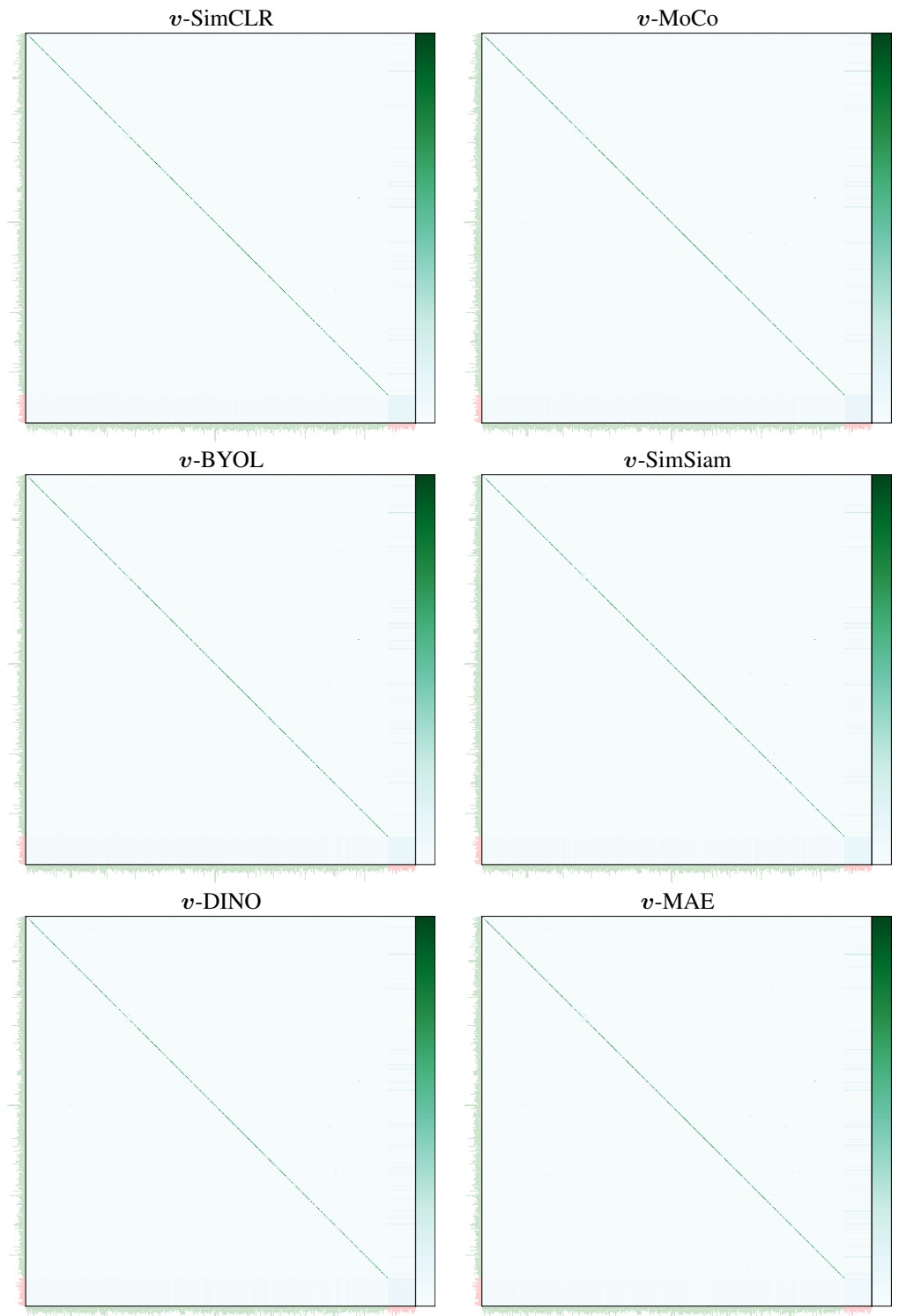

Figure S32: Confusion matrices of open-set recognition, using **Kinetics400 as closed-set and UCF as open-set**. The x-axis represents the ground truth and the y-axis represents the predicted labels. We highlight known classes with green and unknown classes with red. The results exhibit that all the video models struggle in identifying 'unknown' classes from UCF as 'unknown'. This is likely due to their over-confident predictions.

