# OpenReview forum: "Uncovering the Hidden Dynamics of Video Self-supervised Learning under Distribution Shifts"
_NeurIPS.cc/2023/Conference — NeurIPS 2023 spotlight_

### Official Review · Reviewer_E4vF · 2023-06-27

**Soundness:** 3 good
**Presentation:** 3 good
**Contribution:** 3 good
**Rating:** 6
**Confidence:** 4

**Summary:**

The authors studied the behavior of six popular self-supervised methods in response to various forms of natural distribution shift. And the study uncovers a series of interesting findings and behaviors of video self-supervised learning (VSSL) methods. The experiments and results are beneficial for the video representation learning community.

**Strengths:**

1. Extensive experiments are conducted. And sufficient experimental data and analysis are provided. It seems to the reviewer that all the conclusions are grounded.





**Weaknesses:**

1. It seems that the effects of data size and model size are not discussed.

2. Many experiments are conducted and some conclusions are obtained. But no improved VSSL are provided, which might be more interesting.

**Questions:**

1. Any suggestions for the future research to design better DSSL method?

2. Providing a table of summarisation of the 6 methods might be helpful for the readers. There are many experiments and data.

3. The methods are pre-trained on K400 and K700 datasets, which might not be large enough. If the models are pre-trained on a much larger dataset, will the conclusions in this paper be changed?

4. Does the model sizes effect the conclusions in this paper ?


**Limitations:**

Limitations have been discussed in the text. And no obvious negative societal impacts.

---

> ### Author Rebuttal · Authors · 2023-08-08
>
> We thank the reviewer for taking the time and providing such valuable feedback. We are happy to note that the reviewer finds our arguments well-grounded with the experimental results.
>
>
>
>
> > Effect of data and model size
>
>
> - In **Table S6 (Appendix D)**, we notice that using a larger dataset of more diverse videos generally helps in better performance in both InD and OoD, e.g., overall linear evaluation performance is improved by 1.5% and 0.9% in InD and OoD. Amongst the VSSL methods, v-BYOL benefits the most from the availability of diverse pretraining data, e.g., the overall performance of v-BYOL is improved by 3.1% and 1.9% in InD and OoD respectively. Please kindly refer to Appendix D1 where we discuss the effect of using larger data.
>
>
> - We acknowledge that the impact of model sizes would be another area for exploration, which we have mentioned in Section 7 under “Limitations”. While we could not perform experiments on a larger network due to resource constraints, we have now explored a different architecture, video ResNet50 (please see **Table R1 in the attached pdf**), which shows consistent trends with respect to the ViT backbones used in our paper. Given this consistency when changing the backbone, we anticipate similar observations should the model sizes also be reasonably changed. But we agree that further experiments are required to substantiate this.
>
>
>
>
>
>
> > No improved VSSL are provided; Suggestions for future research
>
>
> We thank the reviewer for this question. As our overall goal in this work is to study the robustness of *existing* VSSL models under distribution shifts, introducing a new method is beyond the scope of this paper.
>
> We do believe, however, that our work can be used to drive the design of future VSSL methods with improved performance. Following, we share some of our thoughts that can be used as a suggestion for future work, we will also include this in the final version.
>
> A general guideline could be to train video SSL frameworks to learn *local time-invariant representations* and *global time-variant representations*.
>
> Our intuition is that such models would be
> - robust against viewpoint shifts as it learns *view-invariance* through local time-invariant representations, similar to v-MoCo and v-SimCLR
> - robust against mere temporal perturbations as they learn locally time-invariant representations, similar to v-BYOL and v-DINO
> - robust to context shifts as it understands the global temporal dynamics well, similar to v-MAE.
>
>
> The objective function can be designed as a combination of masked reconstruction and contrastive/non-contrastive methods. We will further investigate such approaches in future.
>
>
> > Summary table
>
>
> We thank the reviewer for sharing this great idea. Please see **Figure R1 (attached pdf)**, we will also add it in the final version.
>
>
> > K400 and K700 datasets might not be large enough; If the models are pre-trained on a much larger dataset, will the conclusions in this paper be changed?
>
>
> - Kinetics700 (K700) consists of 0.5 million videos and is one of the widely used large-scale **open-source** pretraining benchmarks for video self-supervised learning [1,3,72].
> - Amongst the other open-source video datasets, a potential alternative is AudioSet 2M, which is popularly used in audiovisual self-supervised learning. We internally experiment with AudioSet as well and the results are added to **Table R3 (attached pdf)**. We find that AudioSet also shows a similar trend to our findings from K700, confirming that our findings are likely to be aligned even when pretrained on other large datasets.
> - Despite AudioSet being larger than Kinetics700, we do not notice a clear benefit as shown in **Table R3 (attached pdf)**. This is likely since the action classes present in Kinetics are more aligned with the downstream actions compared to AudioSet.
> - We note a few prior works that used even larger datasets like IG65M [98] which could have been beneficial in our study; however, they are not open-sourced, hence we could not use them.
>
>
>
>
> References:
>
>
> - [1] VideoMAE: Masked Autoencoders are Data-Efficient Learners for Self-Supervised Video Pre-Training, Z Tong et al., NeurIPS 2022
> - [3] Spatiotemporal Contrastive Video Representation Learning, R Qian et al., CVPR 2021
> - [72] A Large-Scale Study on Unsupervised Spatiotemporal Representation Learning, C Feichtenhofer et al., CVPR 2021
> - [98] Large-scale weakly-supervised pre-training for video action recognition, D Ghadiyaram et al., CVPR 2019

---

> > ### Comment · Area_Chair_hcRo · 2023-08-18
> > **Reviewer E4vF**
> >
> > The author has posted their rebuttal, but you have not yet posted your response. Please post your thoughts after reading the rebuttal and other reviews as soon as possible. All reviewers are requested to post this after-rebuttal-response.

---

### Official Review · Reviewer_jxy6 · 2023-07-06

**Soundness:** 3 good
**Presentation:** 2 fair
**Contribution:** 3 good
**Rating:** 6
**Confidence:** 3

**Summary:**

The paper studies the generalizability of many self-supervised training approaches in the video domain. The paper uses 17 tasks to completely test different aspects of the models, including view-point change, temporal modeling, and open set generalizability.

**Strengths:**

1. Most of the cutting-edge video pre-training approaches are covered.
2. Generalizability is an important problem for current large foundation models.

**Weaknesses:**

1. It's good to have a hyper-parameter table in the appendix. However, I am not sure if the authors have swept hyper-parameters for each method, and make sure each pre-training is complete. I understand the training for each approach is expensive, but the value will be reduced if the pre-training recipe hasn't been fully explored.
2. Given the paper studies the pros and cons of each approach, I would expect there will be a proposed approach that performs better in most cases or a guideline that can help readers to develop such a method.
3. Now the conclusion of each question scatters in different sections. This is fine but I think having a summary table to compare all approaches and show the findings at the beginning or at the end could help readers to understand the pros and cons of each approach better.

**Questions:**

1. How is the v-supervised trained? Does it use kinetics' labeled data for pre-training?
2. In lines 194 - 198 and Table 1, I didn't see the OoD performance of v-supervised and v-MAE as better than the others if they learn better time-variant information. Most of the time Table 1 shows mixed results in linear probing and fine-tuning (e.g. v-MAE performs better in OoD with FT but worse in Lin.).
3. Maybe this paper can consider citing [1, 2] in lines 219 - 222.
4. In Table 5, why the trend of fine-tuning and linear probing is very different?

[1] Lei, Jie, Tamara L. Berg and Mohit Bansal. “Revealing Single Frame Bias for Video-and-Language Learning.”
[2] Buch, S., Cristobal Eyzaguirre, Adrien Gaidon, Jiajun Wu, Li Fei-Fei and Juan Carlos Niebles. “Revisiting the “Video” in Video-Language Understanding.”

---
**Post-rebuttal**
Thank you for the authors' response. I have read it and it addressed my questions.

**Limitations:**

The limitation is adequate.

---

> ### Author Rebuttal · Authors · 2023-08-08
>
> We thank the reviewer for sharing such thoughtful reviews. We are happy to find the overall positive feedback provided by the reviewer.
>
>
> > It's good to have a hyperparameter table in the appendix; if the authors have swept hyperparameters for each method?
>
>
> We would like to point out that we have indeed swept a broad range of parameters for each method to ensure that the hyperparameters are thoroughly tuned for each method.
>
>
> Below we provide a description for some of the key hyperparameter setups:
>
>
> - **Augmentation**: The optimal setup for augmentation varies amongst contrastive, non-contrastive, and masked autoencoder. Based on empirical findings, cropping ratios of [0.08, 1], [0.2, 0.766], and [0.5, 1] are used for contrastive, non-contrastive, and v-MAE respectively. Additionally, color-jittering, blur, or horizontal flip are not applied for v-MAE, but they are applied to other VSSL methods.
> - **Epoch**: Following [1, 6], we pretrain these methods *up to* 800 epochs and track their performance using InD validation sets (UCF101, Kinetics400) and notice that all the methods reach their optimal somewhere between 600 to 800 epochs. We use the best checkpoints of each method in the downstream tasks.
> - **Learning Rate (LR)**: We individually tune LR for all the methods through a grid search between 5e-5 to 1e-3. We find the optimal LR for v-MoCo, v-BYOL, and v-DINO as 3e-4, v-SimCLR as 2e-4, and v-SimSiam as 1e-4. Additionally, the predictor heads are trained with 10 times higher LR with respect to the base LR, to achieve optimal performance.
> - **Projector head**: The optimal configuration for the projector head also varies amongst the VSSL methods, e.g., an MLP head of 4 layers is optimal for v-SimCLR and v-SimSiam, while an MLP head of 3 layers works best for v-MoCo, v-BYOL, and v-DINO.
> - **Predictor head**: the configuration of the predictor head is also adjusted, i.e., v-MoCo and v-SimSiam work best with a predictor head of just 1 layer while a 2-layer predictor head works best for v-BYOL.
> - **Others**: we also tune other hyperparameters like **weight-decay**, and **EMA** coefficient, among others to find the optimal configuration for each method.
>
>
> We present the hyperparameters related to each method in **Tables S1 and S3** in **Appendix B**.
>
>
> > Guidelines for future work
>
>
> We thank the reviewer for this question.
> Following, we share some of our thoughts that can be used for future work. We will also include this in the final version of the paper.
>
>
> A general guideline could be to train video SSL frameworks to learn *local time-invariant representations* and *global time-variant representations*.
>
>
> Our intuition is that such models would be
> - robust against viewpoint shifts as it learns *view-invariance* through local time-invariant representations, similar to v-MoCo and v-SimCLR
> - robust against mere temporal perturbations as they learn locally time-invariant representations, similar to v-BYOL, v-DINO
> - robust to context shifts as it understands the global temporal dynamics well, similar to v-MAE.
>
>
> The objective function can be designed as a combination of masked reconstruction and contrastive/non-contrastive methods. We will further investigate such approaches in future.
>
>
> > Summary table to compare all approaches
>
>
> We thank the reviewer for sharing this great idea. Please see **Figure R1 (attached pdf)**, we will also add it in the final version.
>
>
> > How is the v-Supervised trained? Does it use kinetics' labeled data for pre-training?
>
>
> Yes, the v-Supervised model is pretrained using the labels of the Kinetics dataset. We follow a similar recipe to [76] ViViT: A Video Vision Transformer.
>
>
> > In Table 1, v-MAE and v-Supervised show mixed results; v-MAE performs better in OoD with FT. but worse in Lin.
>
>
> - Yes, the reviewer’s observation is correct. However, our key takeaway from Table 1 is that *v-MAE consistently outperforms in both OoD setups when finetuned and v-Supervised shows the best performance in 3 out of 4 OoD setups*. It should be noted that the poor performance of v-MAE in Lin. is not specific to context shift, v-MAE shows poor Lin. performance in almost all the setups. Therefore, we conjecture v-MAE and v-Supervised are strong temporal learners based on their overall superior performance.
> - However, as the trend in real-world evaluation is noisy as correctly noted by the reviewer, we carry out tests in a controlled setup on a toy dataset. In particular, we aim to disentangle the spatial and temporal representations to accurately evaluate who learns the temporal dynamics better irrespective of their spatial representation learning capability. As presented in Figure 2a, both v-MAE and v-Supervised show superiority over other methods by a very large margin, confirming that they learn better temporal dynamics.
>
>
> > Suggested reference
>
>
> We thank the reviewer for suggesting these refs, we will add them in the final version. They are indeed relevant to our study,
> - the work by Lei et. al. is related to context shift as it studies static appearance bias;
> - Buch et. al. introduces atemporal probe model which is relevant to our work, as we also discuss the ability of the video models in understanding temporal dynamics.
>
>
>
>
>
>
> > In Table 5, why is the trend of fine-tuning and linear probing very different?
>
>
> The superiority of v-SimCLR and v-MoCo in open-set recognition when finetuned may be attributed to their better generalizability, which, while advantageous in linear probing, becomes a pitfall in open-set scenarios. These highly generalizable models, driven by their overconfidence, often misinterpret unknowns as known classes, leading to incorrect predictions. Conversely, *weak frozen encoders* avoid such misclassifications due to their limited generalizability and perform better in open-set conditions, as observed in models like v-DINO and v-Supervised. Interestingly, such a trade-off is only noticed in linear evaluation and not in finetuning.

---

> > ### Comment · Area_Chair_hcRo · 2023-08-18
> > **Reviewer jxy6**
> >
> > The author has posted their rebuttal, but you have not yet posted your response. Please post your thoughts after reading the rebuttal and other reviews as soon as possible. All reviewers are requested to post this after-rebuttal-response.

---

### Official Review · Reviewer_D3xd · 2023-07-07

**Soundness:** 2 fair
**Presentation:** 3 good
**Contribution:** 3 good
**Rating:** 5
**Confidence:** 3

**Summary:**

The paper proposes a set of benchmarks to assess different robustness properties of video representation learning models, including contrastive, non-contrastive, and generative models. The paper trains multiple of these models using a common training protocol and reports multiple empirical findings on different forms of distribution shift.

**Strengths:**

The paper outlines a very clear training and evaluation protocol and nicely outlines the results. I like the "highlights" section at the end of each addressed question section (up to comments below). The considered datasets and models are extensive. Benchmark and models could become a very valuable resource for further robustness benchmarking on video datasets.

**Weaknesses:**

- One major weakness of the paper seems to be the missing control for the baseline performance of the proposed methods. For instance, in Table 2, several claims are made with respect to the performance of contrastive vs. other models under viewpoint shifts (l. 246 etc). However, in Table 2 it is unclear whether these improvements are originating from the improved OoD, or in fact the difference in InD performance. Given that the authors computed error bars for all results, I propose to equip every statement in the paper with a suitable hypothesis test capable to test the influence of the different factors.
- Another major weakness is the missing link to published results. All models in the paper are trained from scratch by the authors, and no numbers are reported by applying existing model checkpoints from the literature and confirming that the trained models reach a comparable performance level. What is the rationale for not verifying model performance e.g. against the released best models trained on image datasets?


Other weaknesses:

- In the training protocol, parameters like the batch size were kept constant across methods. However, it is known that methods like SimCLR depend on availability of large batch sizes, while models like MoCo have mechanisms build in to more effectively leverage data in small batchsize training setups. Hence, I disagree with the author's statement that the considered setting is "fair". Happy to discuss the rationale for these choices (vs. for example using the best available model configurations). Minimally, it would be good to discuss this point more in the paper, e.g. in the limitation section.
- For a purely empirical paper, I would recommend to back up the fairly general statements at the end of each section with statistical tests.
- The plots are unreadable without heavily zooming in, e.g. Figure 4 and Figure 6. Figure 5 is borderline.

**Questions:**

- If I read the methods correctly, you trained all models from scratch on the respective datasets (Kinetics400 and -700). Did the video models  you trained readily outperform the best "static image" models available for the respective methods?
- Will you open source the model checkpoints and source code for all models you trained in this study? The paper only makes a statement regarding the code.
- The results in Table 2, Viewpoint (ego) seem to be very bad (around 11%). Did you investigate possible causes for this performance drop? How do you justify to infer conclusions from this part of the table (e.g. in l. 246 etc) given the bad InD performance to begin with?

**Limitations:**

Yes, limitations have been addressed (up to the additional comments I made above).

However, given the amount of datasets used in the paper, I am missing license and copyright statements for these datasets, e.g. in the appendix, that go beyond the citations provided in section 5 in the main paper. This could e.g. be included into Appendix C. If the benchmark is intended for later release as outlined, I think it would be very useful to have such an overview directly in the paper for future reference by users of the benchmark.

---

> ### Author Rebuttal · Authors · 2023-08-07
>
> We thank the reviewer for providing such valuable feedback and a thorough review. We are glad to note that the reviewer finds the 'Highlights' at the end of each section useful and finds our work extensive and valuable.
>
> > Do the improvements originate from the improved OoD or from the difference in InD performance, e.g., Table 2?
>
> We thank the reviewer for raising the question. In **Table R4 (attached pdf)**,  we perform statistical tests (Pearson corr.) investigating if the improvements in OoD are due to the *OoD robustness* or from improved InD performance.
>
> - Overall, InD vs. OoD performance in linear eval. show a higher corr. compared to when finetuned, i.e., 0.71 vs. 0.55.
> - In the case of viewpoint shift (in Table 2), we notice a strong corr. in *linear eval. of egocentric view* and *finetuned eval. of surveillance camera view*. In all the other viewpoint shift setups, the corr. is fairly low.
>
> We further study the relative OoD robustness (i.e., measuring the performance drop w.r.t InD) to compensate the effect of varying InD performance using the corr. coefficient between OoD and InD performance, and find that our overall conclusions regarding models' robustness are still valid. We note that this is not a perfect solution to compensate the effect but nevertheless, the best we could think of. We'd be happy to try any other suggestions the reviewer might have in mind.
>
> > Confirming that the VSSL implementations reach a comparable performance level to prior works
>
> In **Table R5 (attached pdf)**, we compare the performance of v-SimCLR with prior works which are also based on (or inspired by) SimCLR, confirming that our implementation achieves comparable performance w.r.t the prior works. Please note that while these methods use a similar objective function to SimCLR, their implementations largely vary amongst each other. Considering such variations amongst the prior works, we choose to implement and train all the VSSL methods by ourselves and pretrain them in identical experiment setups with necessary hyperparameter tuning for a fair comparison.
>
> > What is the rationale for not using models trained on image datasets? Did the video models you trained readily outperform the best "static image" models available for the respective methods?
>
> In **Table R2 (attached pdf)**, we compare image vs. video pretraining based on 3 SSL methods (MoCo-v3, DINO, MAE) on a variety of OoD setups. We particularly choose these 3 methods as they are originally proposed with ViT similar to our setup. The results presented in Table R3 exhibit up to 9.8% improvements when using video models compared to their image variants.
>
> > Batch size is kept constant; SimCLR works better w/ large batch but MoCo works better w/ small batch.
>
> - We follow a similar *video* pretraining setup of [72] and use the same batch size for all the variants.
> - We would also like to clarify that our v-MoCo is based on MoCo-v3 [23] (not *MoCo by K. He et al., CVPR'20*), and as discussed in [23] MoCo-v3 performs best with a similar batch size (2048) to SimCLR when pretrained with unlabelled *images* from ImageNet.
> - From [3] we observe that the performance of *video contrastive methods* shows a very stable performance when using a batch size between 512 to 1024, and below or above that range performance degrades. We believe the batch size of 768 is not a detrimental factor here as it is within a standard range and the LR is adjusted accordingly.
>
> > Hypothesis and statistical test to back up the summary statements at the end of each section
>
> The Highlights mentioned in section 6 are a summary of the key findings based on our empirical study. To ensure our observations are statistically significant, we run these experiments 3 times with different seeds and report the average and standard deviation. Additionally, to strengthen our arguments observed from the evaluation on the real-world datasets, we also conduct a series of toy experiments in a controlled setup to further verify some of these hypotheses.
>
> > The plots are unreadable without zooming.
>
> We thank the reviewer for pointing it out. We will enlarge the size of the figures in the final version as it allows for an additional page.
>
> > Will you open source the model checkpoints and source code?
>
> Yes, we will open-source the model checkpoints and source codes upon publication.
>
> > Possible cause for poor results in egocentric viewpoint shift experiments and additional justifications
>
> This is likely considering the challenging nature of the Charades-Ego dataset, which is comprised of videos from 157 fine-grained household activities, moreover, each video contains multiple labels. However, we would like to highlight that our results are in a similar range to prior works, e.g., [36] reports InD mAP of 23.3 vs. ours 21.4, and OoD mAP of 19.5 vs. ours 16.1. The 2-3% drop is likely since [36] uses  RGB *+ optical flow*, whereas our method only uses RGB frames. As for the linear eval. performance being around 11%, we would like to point out that to our knowledge, no prior works have performed linear eval. on this dataset.
>
> > How to make conclusions about OoD when InD performance is not strong, e.g., Table 2 Viewpoint (ego.)?
>
> We acknowledge that in such a case, a strong claim about OoD could not be made. However, our conclusion about viewpoint shift is not only based on the performance on egocentric viewpoint shift but rather based on a trend noticed across all 3 viewpoint shifts in both linear and finetuning. Moreover, our claims are also validated through toy experiments for additional confirmation.
>
> > Dataset license
>
> We thank the reviewer for this suggestion. Following, we provide the license statements for each dataset and will also add this in Appendix C.
>
> - CharadesEgo, MiT-v2: License for Non-Commercial Use
> - Kinetics, HMDB51, ToyBox: CC BY 4.0
> - Mimetics, UCF101, TinyVirat-v2, COIL100, STL-10: Open access
> - ActorShift, Sims4Action: MIT
> - RareAct: Apache

---

> > ### Comment · Area_Chair_hcRo · 2023-08-18
> > **Reviewer D3xd**
> >
> > The author has posted their rebuttal, but you have not yet posted your response. Please post your thoughts after reading the rebuttal and other reviews as soon as possible. All reviewers are requested to post this after-rebuttal-response.

---

> > ### Comment · Reviewer_D3xd · 2023-08-18
> > **Re: Rebuttal**
> >
> > Thank you for the comprehensive rebuttal. I especially appreciate the efforts to perform experiments comparing image and video models, as well as the statistical analysis performed. I wanted to follow up on a few points:
> >
> > - **Statistical analysis**, I have follow up questions on Table R4:
> >   - Could you provide the details on how you compute the correlation coefficients, i.e. for which table in the paper? It would help to read a few more details on the analysis (e.g., like you would also put it into the methods/supplement of the paper), like which model was employed, which test was performed, etc. In general, thanks for going in this direction, I think this will greatly strenghten the claims in the paper. The way how I read it right now is that you find a correlation between IID and OOD performance across all methods, which was one of my concerns --- I would like to better understand the effect of model (e.g. contrastive vs. non-contrastive) vs. the confounder that the IID performances between the models vary. (maybe I am also not fully understanding your analysis, hence, please expand)
> >   - To make it more concrete: Could you again make a very clear example how e.g. the statement "contrastive methods (v-SimCLR and v-MOCO) are robust to viewpoint shifts as they consistently achieve better performance in all three setups in both linear and finetuning schemes." (l. 223 in the paper) maps to Table 2 and is supported by your statistical analysis? There are a few more of these kind of strong statements in the paper that are not fully clear to me yet, I might follow up with a few additional examples.
> > - **Source code release**: The additional info to open source model checkpoints and evaluation code are very useful. Regarding the code, is this already in a state ready to release? If so, I think there is an opportunity to send the AC a link to the codebase which they can pass along to me. I would be interested to have a look, if this is well setup it would strengthen the contribution.
> > - **Figure R1**: The color somehow needs to map to a metric/number. I think this overview is great and could e.g. go into the supplement, but you should work on making this quantitative (vs. "low" to "high")
> >
> > I apologize for the late response just before the weekend --- please feel free to post multiple replies as they get ready in case this speeds up the further discussion.

---

> > > ### Comment · Reviewer_D3xd · 2023-08-19
> > > **Re: Statistical Analysis**
> > >
> > > To make my previous comment more clear, here is a quick re-analysis of Table 2.
> > >
> > > The simplest control for IID performance is to consider the statistic "OOD - IID" (lower = more robust), for which we get with linear evaluation (bold = lowest degradation):
> > >
> > > |    | method    | method_type     |   ego |   surv |   view+act |
> > > |---:|:----------|:----------------|------:|-------:|-----------:|
> > > |  0 | v-MOCO    | contrastive     |  **-1.7** |  -15.2 |      -26.8 |
> > > |  1 | v-SimCLR  | contrastive     |  -2   |  -13   |      -25.4 |
> > > |  2 | v-BYOL    | non-contrastive |  -2.4 |  -15.1 |      -28.3 |
> > > |  3 | v-DINO    | non-contrastive |  -2.4 |  -13   |      -27.6 |
> > > |  4 | v-MAE     | non-contrastive |  -2.8 |   **-8.5** |      **-20.2** |
> > > |  5 | v-SimSiam | non-contrastive |  -2.5 |  -11   |      -25.5 |
> > >
> > > and for fine-tuning, we get:
> > >
> > > |    | method    | method_type     |   ego |   surv |   view+act |
> > > |---:|:----------|:----------------|------:|-------:|-----------:|
> > > |  0 | v-MOCO    | contrastive     |  **-3.3** |  -21   |      -17.2 |
> > > |  1 | v-SimCLR  | contrastive     |  -4   |  **-19.5** |      -15.7 |
> > > |  2 | v-BYOL    | non-contrastive |  -3.6 |  -22.2 |      -22   |
> > > |  3 | v-DINO    | non-contrastive |  -3.7 |  -20.9 |      -27.2 |
> > > |  4 | v-MAE     | non-contrastive |  -7.2 |  **-19.5** |      **-15.1** |
> > > |  5 | v-SimSiam | non-contrastive |  -3.9 |  -19.6 |      -22.4 |
> > >
> > > I did not run a statistical test here as I only had access to the mean values and standard deviations for either ood or iid. It would be insightful if the authors could run a full statistical analysis or share all results that were used to compute the standard deviation. But even without an analysis, I question that the statement "contrastive methods (v-SimCLR and v-MOCO) are robust to viewpoint shifts as they consistently achieve better performance in all three setups in both linear and finetuning schemes." is conclusivly backed by the experimental data.
> > >
> > > Another example is the statement "Overall, the results of zero-shot recognition presented in Table 4 show that no single method dominates in all three benchmarks of zero-shot recognition.", where again absolute performance values are compared in Table 4. However, the performance difference in InD performance on K400 varies as much as 58.4 - 35.9 = 22.5% points on linear eval, and 69.8 - 64.3 = 5.5% points for finetuning, which makes the statement almost trivial (the difference in in domain performance seems to be so drastic that it is impossible to make an informed statement about OOD performance).
> > >
> > > I hope these examples make my concerns even clearer. Finally, to reference the author rebuttal,
> > >
> > > > To ensure our observations are statistically significant, we run these experiments 3 times with different seeds and report the average and standard deviation.
> > >
> > > **This is not sufficient to claim statistical significance.**
> > >
> > > What is missing in multiple sections in the paper is the link from numbers in tables and plots to (minimally) a summary statistic that is ideally free of confounding information (e.g., difference between ood and iid performance in the example above, and other statistics in other cases). And/or ideally, there needs to be a suitable test to make these statements really convincing.
> > >
> > > The alternative is: (1) Remove the word "significant" from the abstract and text when no test is performed (2) Make it clearer on how you back up each statement in the "Hightlights" with numbers, that goes beyond comparing the OOD performances on the different datasets (as these are confounded by choices like fine-tuning vs. linear tuning, the IID performance on the source data, and so forth).
> > >
> > > I am insisting on this point from my original review as I fear that in case the paper gets published as-is, a lot of statements will be referenced in the literature that are not backed by experimental data due to this missing link between "raw" numbers and the respective statements in the text.
> > >
> > > Happy to discuss further.

---

> > > > ### Author Response · Authors · 2023-08-19
> > > >
> > > > We would like to sincerely thank you for all your time and efforts in engaging with our paper and rebuttal, providing constructive suggestions, and allowing us to respond again.
> > > > - We have created an anonymized repository that contains the codes for self-supervised pretraining and finetuning. The repo also contains a link to an anonymized google drive that contains all the necessary checkpoints for the reproducibility of the results. The link to the repo is now provided to the AC, so that they’d share it with you. Please kindly refer to the README.md files in the main folder along with the two main subfolders (`vssl-train` and `vssl-eval`) for details on how to navigate the codebase. We appreciate your patience while we were putting the cleanup touches on the code.
> > > > - The repo also contains the code (`table_r4.ipynb`) used in creating Table R4, per your request.
> > > > - The repo also contains an updated Figure R1 (`figure_r1.pdf`) where we have added all the numbers in the different columns, per your suggestion.
> > > > - We are currently working on providing all the statistical tests in a unified and comprehensible manner, along with a detailed response to your comments about the claims, so that we can share them with you.
> > > >
> > > > Thank you.

---

> > > > ### Author Response · Authors · 2023-08-21
> > > > **Re: Rebuttal & Statistical analysis**
> > > >
> > > > We would like to thank you again for engaging with our work and your feedback on our paper. We have now provided an anonymous google sheet document to the AC to be passed along to you. We have also added a link to this gsheet in the README file of the anonymous repository. The provided gsheet contains all the statistical analyses for every major claim in the paper. Following we provide a point-by-point response to your questions.
> > > >
> > > > > Comparing OOD performance after compensating InD performance.
> > > >
> > > > As you suggested, we use OoD - m * InD as the basis for compensating for InD performance, where $m \in [0, 1]$. Further, to consider statistical significance, we modified the following equation, and `accept` **method1 is more robust than method2** if **Adjusted $\Delta_{ood}$>0**:
> > > >
> > > > Adjusted $\Delta_{ood}$ =
> > > > (method1_ood - method2_ood) - m$\times$(method1_ind - method2_ind) - 1.96 $\times$ method1_ood_stderr
> > > >
> > > > - The first term **(method1_ood - method2_ood)** measures the difference between the OOD performance of two methods.
> > > >
> > > > - The second term **m $\times$ (method1_ind - method2_ind)** compensates for InD performance, where *m* refers to the slope of a linear fit between InD acc and OoD acc.
> > > >
> > > > - The third term **1.96 $\times$ method1_ood_stderr** considers the standard error to ensure statistical significance at 95% confidence.
> > > >
> > > > The first two terms of the formula correspond to the plots in **Figure S12** in the **Appendix**. In **Figure S12**, the method that lies above the linear fit with the highest margin is considered to be the most robust to OoD shift. While the statistical confidence is eliminated from **Figure S12** to prevent creating a clutter, we now present those statistical analyses in the shared gsheet.
> > > >
> > > > The shared google sheet contains the results using this equation for the claims in the paper. Overall, the sheet confirms that all our claims hold. While in the paper, we had originally given a higher-level overview of the claims, as per your suggestion, we will add further details to each claim to make them more clear as per our discussions below.
> > > >
> > > > > Could you again make a very clear example how e.g. the statement "contrastive methods (v-SimCLR and v-MOCO) are robust to viewpoint shifts as they consistently achieve better performance in all three setups in both linear and finetuning schemes." (l. 223 in the paper) maps to Table 2 and is supported by your statistical analysis?
> > > >
> > > > Based on our statistical analyses, v-SimCLR is more robust to all three viewpoint shifts and v-MOCO is robust to egocentric and top-down viewpoint shifts. Additionally, our claim that v-SimCLR and v-MOCO learn better viewpoint invariance is also supported by statistical analysis. The statistical results are presented in `tab: viewpoint`. This is also corroborated in **Figure S12 (c, d, and e)** in the **Appendix**. As can be seen, v-SimCLR consistently clears a positive margin from the linear fit suggesting its relative robustness to viewpoint shifts.
> > > > To add more clarity and detail, we will add further details to our claim in L. 223 to read as
> > > >
> > > > ```
> > > > v-SimCLR exhibits robustness to all three viewpoint shifts. v-MOCO shows robustness to two out of three viewpoint shift setups, i.e., egocentric (viewpoint (ego.)) and top-down (View+Act (t-down+syn.)).
> > > > ```
> > > >
> > > > > "Overall, the results of zero-shot recognition presented in Table 4 show that no single method dominates in all three benchmarks of zero-shot recognition.", where again absolute performance values are compared in Table 4. However, high-performance difference in InD makes the statement almost trivial.
> > > >
> > > > We test robustness between all possible pairs after compensating for InD performance and the results confirm our claim that there is no single method that exhibits superiority in all three setups of zero-shot recognition either in linear or finetuned setup.
> > > > The statistical results are presented in the google sheet `tab: zeroshot`.
> > > >
> > > >  > It would be insightful if the authors could run a full statistical analysis or share all results that were used to compute the standard deviation.
> > > >
> > > > We provide our statistical tests to support all our major claims in the paper in a separate comment.
> > > >
> > > > > (1) Remove the word ''significant'' from the abstract and text when no test is performed.
> > > >
> > > > We will remove the word "significant'' from the paper where appropriate, and replace with the actual numbers for better clarity. Kindly note that in some cases, e.g., the abstract, the term "significant" is used in the sentence: ''Video self-supervised learning (VSSL) has made significant progress in recent years''.
> > > >
> > > > > (2) Make it clearer on how you back up each statement in the "Hightlights" with numbers
> > > >
> > > > Thanks for this suggestion! We will summarize the gist of the google sheet (also discussed in our separate comment) with all the statistical tests in a new table and add it to the supplementary material of the paper. We will also provide the detailed results of the tests with the code.

---

> > > > > ### Author Response · Authors · 2023-08-21
> > > > > **Statistical analysis of all major claims in the paper.**
> > > > >
> > > > > Below, we provide our statistical tests to support all our major claims in the paper.
> > > > >
> > > > >
> > > > > - L.223:
> > > > >
> > > > > *The results on viewpoint shifts, presented in Table 2, reveal that contrastive methods (v-SimCLR and v-MOCO) are robust to viewpoint shifts as they consistently achieve better performance
> > > > > in all three setups in both linear and finetuning schemes.*
> > > > >
> > > > > Our statistical test confirms that v-SimCLR is robust to all three viewpoint shifts. Additionally, v-MOCO shows robustness to two out of three viewpoint shift setups, i.e., egocentric and top-down viewpoint shifts.  Please see the results in `tab: viewpoint` and accordingly we will add further details to our claim:
> > > > >
> > > > > ```
> > > > > The results on viewpoint shifts, presented in Table 2, reveal that contrastive method v-SimCLR is robust to viewpoint shifts as they consistently achieve better performance in all three setups in both linear and finetuning schemes. Additionally, v-MOCO exhibits robustness to egocentric and top-down viewpoint shifts.
> > > > > ```
> > > > >
> > > > > - L.259:
> > > > >
> > > > > *(b) Contrastive methods (v-SimCLR, v-MOCO) exhibit robustness to viewpoint shifts.*
> > > > >
> > > > > Our statistical test confirms that v-SimCLR is robust to all three viewpoint shifts. Additionally, v-MOCO shows robustness to two out of three viewpoint shift setups, i.e., egocentric and top-down viewpoint shifts.  Please see the results in `tab: viewpoint` and accordingly we will add further details to our claim:
> > > > >
> > > > > ```
> > > > > v-SimCLR exhibits robustness to all three viewpoint shifts. v-MOCO shows robustness to two out of three viewpoint shift setups, i.e., egocentric (viewpoint (ego.)) and top-down (View+Act (t-down+syn.)).
> > > > > ```
> > > > >
> > > > > - L.259:
> > > > >
> > > > > *(a): Video models generally struggle in out-of-context generalization, while v-Supervised and v-MAE exhibit more robustness as they are strong temporal learners.*
> > > > >
> > > > > Our statistical test confirms the robustness of v-MAE and v-Supervised in Context  (10 class) in both linear and finetune setups. Additionally, In Context (50 class) setup, v-Supervised is robust in linear evaluation and v-MAE is robust when finetuned. We will clarify this in the paper, please see the detailed results in `tab: context`, and accordingly we will add further details to our claim:
> > > > >
> > > > > ```
> > > > > (a): Video models generally struggle in out-of-context generalization, whereas
> > > > > v-Supervised and v-MAE exhibit higher robustness to **Context (10 class)** in both linear and finetuned setups. This could be explained by the fact that they are strong temporal learners.
> > > > > ```
> > > > >
> > > > > - L.244:
> > > > >
> > > > > ```
> > > > > While single-stream networks learn better temporal dynamics, Siamese frameworks learn better viewpoint invariance.
> > > > > ```
> > > > > Our experiments support this claim. Please see the results in `tab: single_and_siamese`.
> > > > >
> > > > > - L.229-233:
> > > > >
> > > > > ```
> > > > > It is worth noting that while v-Supervised exhibits comparable performance to VSSL methods in egocentric and surveillance camera viewpoint shifts, its performance decreases significantly when multiple shifts are applied concurrently, as evident in the case of synthetic top-down viewpoint shift.
> > > > > ```
> > > > >
> > > > > Our experiments support this claim. Please see the results in `tab: multiple_shift_supervised`.
> > > > >
> > > > >
> > > > > - L.327:
> > > > >
> > > > > ```
> > > > > (a) Contrastive methods demonstrate superior performance in open-set recognition when finetuned.
> > > > > ```
> > > > > Our experiments support this claim. Please see the results in `tab: openset`.
> > > > >
> > > > > - L.319:
> > > > >
> > > > > ```
> > > > > However, the pretrained encoders, v-SimSiam, v-DINO, and v-Supervised, which are considered weaker in closed-set, perform better in open-set recognition.
> > > > > ```
> > > > >
> > > > > Our experiments support this claim.  Please see the results in `tab: openset`.
> > > > >
> > > > > - L.273
> > > > >
> > > > > ```
> > > > > For example, as shown in Figure 4 (a), while finetuning is significantly beneficial under source shifts of UCF to HMDB, it interestingly hurts the performance when this shift is reversed (HMDB to UCF).
> > > > > ```
> > > > >
> > > > > Our experiments support this claim.  Please see the results in `tab: source_shift`.

---

### Official Review · Reviewer_iDRZ · 2023-07-07

**Soundness:** 3 good
**Presentation:** 3 good
**Contribution:** 3 good
**Rating:** 7
**Confidence:** 4

**Summary:**

This works studies various video SSL methods under distribution shifts. 6 different video SSL methods (SiMCLR, MoCO, BYOL, SimSiam, DINO, MAE) are trained on Kinetics-400 under the same experiment settings (e.g. ViT-B, fixed number of epochs, etc) and then evaluated on various distribution shifts (e.g. viewpoint shift, context shift) and report the findings (e.g. contrastive methods more robust to viewpoint shifts, while MAE is a stronger temporal learner).

**Strengths:**

1. Writing and presentation is clear.
2. Same experimental and test bed setup for all the baselines. The test bed and evaluation code could be useful to the community.
3. Useful empirical observations characterizing the strengths and weakness of prominent approaches.


**Weaknesses:**

No experiments or analyses on model scaling or different architectures.

Some findings are pretty much expected (e.g. contrastive methods not being good temporal learners happens by design), but it's nice have an easy test bed to quantify them.

Some of the settings controlled for can be strong caveats, e.g. fixing the number of epochs can bias evaluations towards methods that converge faster. It is unclear how much of the results hold when models used their respective optimal settings - which is an important setting for evaluations.

Some of the differences could stem not from core differences in pretext task, but in use of a teacher. e.g. MAE can also use an EMA teacher and the representations would converge much faster.

**Questions:**

Could the authors please discuss some of the limitations and caveats to their findings (e.g. some are listed above)?

**Limitations:**

Yes.

---

> ### Author Rebuttal · Authors · 2023-08-08
>
> We thank the reviewer for providing such valuable feedback. We are happy to note that the reviewer finds our work useful to the community.
>
>
> > Experiments on different architectures or model scaling
>
> We have now conducted experiments on video ResNet-50 given its strong performance in video SSL [72]. We conduct these experiments in a variety of OoD setups including context shift 10 and 50 classes, source shift in both *UCF to HMDB* and *HMDB to UCF*, and animal domain actor shift. The results are presented in **Table R1 (attached pdf)**. To conduct these experiments, we perform linear evaluations using pretrained weights released by [72] and compare them to the ViT-B. These results confirm that our general findings based on ViT are also applicable when tested using ResNet-based architecture. We anticipate similar trends to hold when scaling the backbone.
>
>
> > Some findings are pretty much expected, but it's nice to have an easy test bed to quantify them
>
> We agree with the reviewer that some of these findings might be intuitive and as the reviewer pointed out there is no work that studies the robustness of video SSL models under real-world distribution shifts, in a unified setup. We would also like to point out that some of the findings are in fact quite counterintuitive. For example, we were surprised to find that:
>
>
> - *frozen contrastive encoders* perform poorly in open-set recognition, while achieving the best open-set performance when finetuned;
> - superior performance of v-BYOL frozen encoder compared to finetuned under source shift (HMDB/UCF);
> - consistently poor performance of v-MAE in linear evaluations.
>
>
> However, our additional investigation allowed us to further explore these findings and pinpoint the root causes of such intriguing behaviours.
>
>
>
>
> > VSSL pretraining hyperparameters and convergence
>
>
> We would like to point out that we have swept a broad range of parameters for each method to ensure that the hyperparameters are thoroughly tuned for each method.
>
>
> Below we provide a description for some of the key hyperparameter setups:
>
>
> - **Augmentation**: The optimal setup for augmentation varies amongst contrastive, non-contrastive, and masked autoencoder. Based on empirical findings, cropping ratios of [0.08, 1], [0.2, 0.766], and [0.5, 1] are used for contrastive, non-contrastive, and v-MAE respectively. Additionally, color-jittering, blur, or horizontal flip are not applied for v-MAE, but they are applied to other VSSL methods.
> - **Epoch**: Following [1, 6], We pretrain these methods *up to* 800 epochs and track their performance using InD validation sets (UCF101, Kinetics400) and notice that all the methods reach their optimal somewhere between 600 to 800 epochs. We use the best checkpoints of each method in the downstream tasks.
> - **Learning Rate (LR)**: We individually tune LR for all the methods through a grid search between 5e-5 to 1e-3. We find the optimal LR for v-MoCo, v-BYOL, and v-DINO as 3e-4, v-SimCLR as 2e-4, and v-SimSiam as 1e-4. Additionally, the predictor heads are trained with 10 times higher LR with respect to the base LR, to achieve optimal performance.
> - **Projector head**: The optimal configuration for the projector head also varies amongst VSSL methods, e.g., an MLP head of 4 layers is optimal for v-SimCLR and v-SimSiam, while an MLP head of 3 layers works best for v-MoCo, v-BYOL, and v-DINO.
> - **Predictor head**: the configuration of the predictor head is also adjusted, i.e., v-MoCo and v-SimSiam work best with a predictor head of just 1 layer while a 2-layer predictor head works best for v-BYOL.
> - **Others**: we also tune other hyperparameters like **weight-decay**, and **EMA**, among others to find the optimal configuration for each method.
>
>
> We present the hyperparameters related to each method in **Tables S1 and S3** in **Appendix B**.
>
>
>
>
>
>
> > Discuss limitations/caveats
>
>
> As discussed in Sec.7 of the paper, it would be of interest to further investigate VSSL methods under distribution shifts with larger Transformer or convolutional architectures, which we could not perform due to resource constraints. However, we believe our findings serve as a foundation for future works to study the behaviour of OoD robustness with model scalability.
>
>
> References:
>
> - [1] VideoMAE: Masked Autoencoders are Data-Efficient Learners for Self-Supervised Video Pre-Training, Z. Tong et al., NeurIPS 2022
> - [6] XKD: Cross-modal Knowledge Distillation with Domain Alignment for Video Representation Learning, P. Sarkar et al., 2022
> - [72] A Large-Scale Study on Unsupervised Spatiotemporal Representation Learning, C Feichtenhofer et al, CVPR 2021

---

> > ### Comment · Reviewer_iDRZ · 2023-08-13
> >
> > Thanks for your response. I've updated my score to reflect the additional experiments and discussion.
> >
> > I should add that some of these findings are not really surprising, e.g. v-MAE's poor performance in linear evaluations is reflective of the mis-match between the pretext task and linear eval, while this is less so for contrastive learning. So my view is that an easy test bed to quantify these is the more valuable contribution here and hope the authors release the code and make it easy to use - which will also be strongly impactful to the paper.

---

> > > ### Author Response · Authors · 2023-08-14
> > >
> > > Thanks again for all the valuable feedback, going over our experiments and discussions, and updating your score accordingly!
> > >
> > > We will release all the code and all model checkpoints and will strive to make them an easy test bed for understanding the robustness of VSSL methods.

---

### Author Rebuttal · Authors · 2023-08-08

We sincerely thank the review committee for their time and for providing constructive feedback. We are happy to see the overall engaging comments given by all the reviewers and glad to note that all reviewers find our work valuable to the community. We have carefully addressed all the concerns raised by the reviewers under the individual response section. Following, we provide a summary of our response.




- **VSSL pretraining hyperparameter search**: In response to individual reviewers, we clarified that we have strived to tune all SSL methods studied in this work to achieve their best performance,  involving best practices and/or grid search for augmentation setups, learning rates, batch size, weight decay, EMA, mask ratio, temperature, and the configuration of the projector heads, predictor heads, and decoder, among others. Please see the detailed response under **Reviewers iDRZ or jxy6** and a summary of pretraining setups is provided in Tables S1 and S3 in Appendix B. Additionally, all the models and codes will be released upon publication for reproducibility.


- **Experiment on existing model checkpoints with a different architecture**: To analyze the generalizability of our findings beyond ViTs, we have conducted experiments to study the OoD robustness of the video SSL methods using existing checkpoints of Video ResNet-50 (Slow only). The results presented in **Table R1 (attached pdf)** show that our findings based on ViT are aligned when evaluated on a video ResNet.


- **Verifying image vs. video pretraining**: In order to ascertain the benefits of video pretraining over models pretrained on static images, we have conducted additional experiments comparing the performance of image SSL vs. video SSL pretraining. Our comparison setups include 3 SSL methods MoCo-v3, DINO, and MAE across a variety of distribution shifts. The results presented in **Table R2 (attached pdf)** exhibit significant improvements when using video models compared to their image variants.


- **Relation between InD vs. OoD performance**: We have now conducted additional statistical analysis investigating the relation between the models' performance in InD vs. OoD. The results presented in **Table R4 (attached pdf)** show a higher corr. when directly using the pretrained frozen encoders (linear probing) compared to finetuned, i.e., average corr. of 0.71 vs. 0.55. This indicates that finetuning is less beneficial under distribution shifts compared to InD. Please see the detailed response under **Reviewer D3xd**.


- **Guidelines for future work**: To further help future research in developing robust and reliable video learning models, we shared some of our thoughts on future work based on the findings from this paper. In short, we find that a robust video representation framework should learn *local time-invariant* and *global time-variant* representations. Our intuition is that such representations are robust to context shift, viewpoint shift, and temporal perturbations among others. Moreover, such models can be designed by exploring a joint objective of masked reconstruction and contrastive/non-contrastive methods. Please see individual responses under **Reviewers jxy6 or E4vF** for more details.


- **Summary table**: In **Figure R1 (attached pdf)** we provide a high-level overview of different VSSL models depicting their robustness and vulnerability under different evaluation setups in both InD and OoD.


- **Pretraining with other large datasets**: We have now added results of VSSL methods pretrained on another large-scale dataset AudioSet consisting of 2M video. The results presented in **Table R3 (attached pdf)** exhibit a very similar trend to the pretraining of Kinetics700. Such strong relation confirms that our findings are likely to be aligned even when pretrained on other large datasets. Please see the detailed response under **Reviewer E4vF**.


We hope that our responses adequately address all the points raised by the reviewers. We would be more than happy to address any additional comments the reviewers may have during the discussion phase.

---

### Decision · Program_Chairs · 2023-09-21

**Decision:**

Accept (spotlight)

**Comment:**

The paper presents a thorough study of video self-supervised learning that uncovers a series of interesting findings that uncover the hidden dynamics of VSSL methods across a variety of distribution shifts.  There was an effective rebuttal and Four knowledgeable reviewers recommended accepting the paper: Accept, borderline Accept, Weak Accept, Weak Accept. Based on the clear positive consensus, it is our recommendation to accept the paper. No basis to overturn the reviews.   Authors should attend to the main points in the reviews and integrate the rebuttal material when preparing a final version.